# Reward Modeling for Reinforcement Learning-Based LLM Reasoning: Design, Challenges, and Evaluation

**Pei-Chi Pan**                                                                 *ppan@CougarNet.UH.EDU*
*Department of Computer Science*
*University of Houston*

**Yingbin Liang**                                                                   *liang.889@osu.edu*
*The Ohio State University*

**Sen Lin**                                                                     *slin50@Central.UH.EDU*
*University of Houston*

**Reviewed on OpenReview:** *https://openreview.net/forum?id=TDfrN1TbGH*

## Abstract

Large Language Models (LLMs) demonstrate transformative potential, yet their reasoning remains inconsistent and unreliable. Reinforcement learning (RL)–based fine-tuning is a key mechanism for improvement, but its effectiveness is fundamentally governed by reward design. Despite its importance, the relationship between reward modeling and core LLM challenges—such as evaluation bias, hallucination, distribution shift, and efficient learning—remains poorly understood. This work argues that reward modeling is not merely an implementation detail but a central architect of reasoning alignment, shaping what models learn, how they generalize, and whether their outputs can be trusted. We introduce Reasoning-Aligned Reinforcement Learning (RARL), a reasoning-centric taxonomic perspective that organizes diverse reward paradigms for multi-step reasoning. Within this perspective, we present a taxonomy of reward mechanisms, analyze reward hacking as a pervasive failure mode, and examine how reward signals unify challenges ranging from inference-time scaling to hallucination mitigation. We further critically evaluate existing benchmarks, highlighting vulnerabilities such as data contamination and reward misalignment, and outline directions for more robust evaluation. By integrating fragmented research threads and clarifying the interplay between reward design and fundamental reasoning capabilities, this work provides a foundational roadmap for building reasoning models that are robust, verifiable, and trustworthy.

## 1 Introduction

The remarkable reasoning abilities demonstrated by advanced Large Language Models (LLMs), often enhanced by reinforcement learning (RL) based fine-tuning, offer a transformative potential to accelerate scientific discovery, automate complex production pipelines, and unlock new frontiers of knowledge. However, the practical application of these capabilities is hampered by their inconsistent and often unsatisfactory performance. Critical studies indicate that what appears to be reasoning currently in LLMs is often sophisticated heuristic aggregation rather than robust logical deduction (Nikankin et al., 2024). This is reflected in empirical findings showing that LLMs generate chains-of-thought (CoT) riddled with inconsistencies and factual errors (Ferreira et al., 2025) and, frequently, fail to act upon the correct knowledge they demonstrably hold (Schmied et al., 2025). Consequently, the reliability of their final answers remains fundamentally in question, and how to enhance their reasoning capabilities in a reliable manner is still very challenging.

This pursuit of advanced reasoning has catalyzed a vibrant research frontier dedicated to applying RL beyond its Reinforcement Learning from Human Feedback (RLHF) origins. While RLHF effectively optimizes for

broad alignment qualities such as helpfulness and harmlessness, it is ill-suited for the rigorous demands of reasoning. True reasoning requires a model to perform multi-step inference, maintain internal consistency, and generalize logical structures—a much stricter criterion than surface-level coherence (Xie et al., 2024). Within this paradigm, reward designs in RL-based fine-tuning for enhancing LLM reasoning serve as the proxy for ground truth answers, guiding the model towards desired behaviors. The design of reward models is critical, as it directly shapes the policy's learning trajectory and final performance, especially in complex reasoning tasks where the path to a solution is as important as the answer itself. Crafting such rewards is nontrivial: effective reward functions must simultaneously mitigate sparsity in final correctness signals, encourage faithfulness of intermediate steps, prevent over-optimization on imperfect or noisy evaluators, and discourage heuristic shortcuts, pattern exploitation, or trajectory memorization that undermine true reasoning ability.

However, evaluating the efficacy of reward design is inherently difficult. The impact of a reward function is confounded by multiple interacting factors, including the choice of RL algorithm, data curation and labeling strategy, the quality of value estimation, and sampling or decoding procedures. Moreover, the black-box nature of large language models complicates attribution, e.g., improvements in benchmark performance may arise from idiosyncratic properties of the training pipeline rather than from improved reasoning ability. Consequently, reported gains are often difficult to interpret and even harder to reproduce. These challenges motivate a deeper investigation of reward modeling not as a peripheral implementation detail, but as a central research question. Understanding how reward functions shape the internal computation and generalization behavior of language models across various settings and domains is crucial for advancing verifiable, reliable, and mechanistically grounded reasoning.

As the landscape continues to evolve rapidly and grow in complexity, the absence of any study that captures the full breadth of domain-specific knowledge for reward modeling underscores the need for a comprehensive synthesis. To fill this gap, this work aims to systematically organize and integrate recent research on reward modeling for LLM reasoning, highlighting both major advances and persistent challenges to help guide future progress. More specifically, we study RL with reasoning-aware rewards—either predefined or parameterized—where LLMs are fine-tuned on reasoning tasks using reward signals derived from human- or AI-annotated ground truth, or from learned or heuristic objectives that approximate or correlate with such supervision. This setting subsumes reinforcement learning from AI feedback (RLAIF), reinforcement learning from human feedback (RLHF), as well as reinforcement learning with verifiable rewards (RLVR). For brevity, we refer to this perspective as Reasoning-Aligned Reinforcement Learning (RARL), as shown in Fig. 2. We categorize the reward design in RARL into three principal paradigms: model-based approaches (Section 2.1), rule-based approaches (Section 2.2), and self-reward (Section 2.3). Within the model-based paradigm, we further present a detailed taxonomy in Section 2.1.1, organized along three key dimensions: model architecture, model granularity, and reward semantics. In addition, Section 3 focuses on the challenge of reward hacking, where we categorize its underlying causes and review representative reward design strategies proposed to mitigate these issues.

Moving beyond standalone reasoning systems, we highlight how reward functions act as a unifying mechanism to mitigate system-level challenges and to improve both performance and robustness. These include inference-time scaling (Section 4.1), where rewards guide planning and computation allocation to improve efficiency and performance without additional training; mitigation of LLM biases (Section 4.2), where reward signals are used to reduce hallucinations, sycophancy, social biases in multi-step reasoning; augmented reasoning systems (Section 4.3), in which external information sources and tools extend reasoning capabilities; and reward design in RL (Section 4.4), where aggressive optimization and hand-crafted rewards can shape policy behavior in unintended ways.

Our discussion also extends to evaluation methodologies, where we highlight their current limitations and propose directions for more robust assessment (Section 5). Finally, we explore practical applications of these techniques across diverse domains, including finance and medicine, to illustrate their real-world impact and requirements (Section 6). An overview of the above-mentioned topics is shown in Fig. 1. Our main contributions in this paper include:

- **Providing a systematic exploration of reward mechanisms and design challenges in a reasoning-centric taxonomic survey RL-based LLM finetuning**, including model-based reward, rule-based, and self-rewarding approaches, and offering a structured taxonomy of reward hacking issues alongside an organized discussion of existing solutions.
- **Bridging reward-driven fine-tuning with core topics in LLM research**, including test-time scaling, efficient learning, benchmark and evaluation biases, augmented reasoning techniques, and common failure modes such as hallucination, offering practitioners multiple perspectives for designing robust and reliable systems under the sensitivities of RL-based fine-tuning.
- **Identifying the defining attributes of successful evaluation benchmarks and highlighting the limitations of existing benchmarks**, particularly data contamination in reasoning evaluation, with a focus on reward evaluation and a comprehensive study of existing solutions.

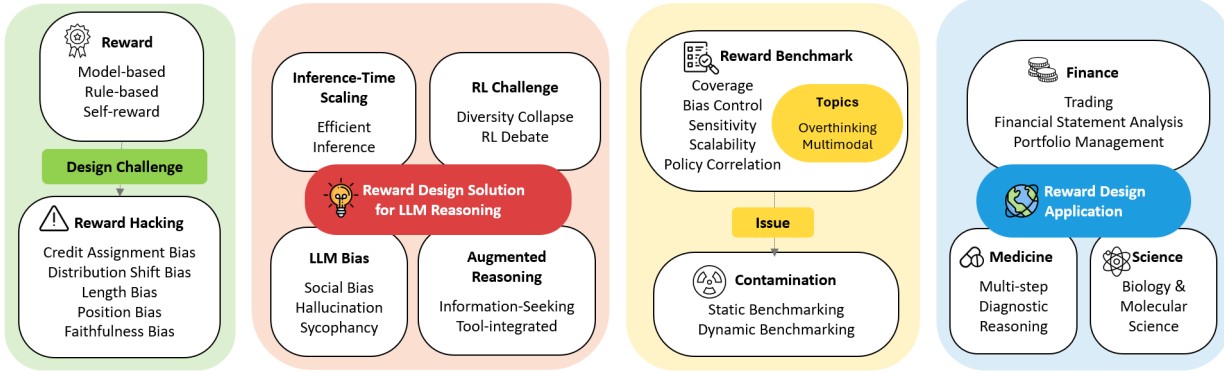

Figure 1: Overview of the work. We first introduce reward design in RL (Section 2) and identify key challenges associated with reward hacking (Section 3). We then show how reward signals can serve as a unified mechanism for improving LLM inference-time reasoning and efficiency (Section 4.1), mitigating LLM bias (Section 4.2), enabling robust augmented reasoning (Section 4.3), and broader reinforcement learning challenges (Section 4.4). We also discuss reward metrics and benchmarks for text-only and multimodal reward models, and key traits to build a successful benchmark (Section 5), followed by the investigation of the reward design in real-world applications (Section 6).

## 1.1 Related Work

Very recently, several studies have emerged to systematically examine the role of reward signals in improving the reasoning and alignment capabilities of large language models (LLMs). Zhong et al. (2025) provide a broad taxonomy and discusses data collection frameworks—both human-annotated and automated—but is not specifically focused on reasoning. Liu et al. (2025d) offer a reasoning-centric classification and analyzes reward model design, though it lacks a comprehensive connection to broader reasoning challenges. Zheng et al. (2025b) focus more narrowly on outcome and process reward models across reasoning-oriented tasks. Wu (2025) focuses on the general paradigm of reward-guided learning rather than a deep treatment of reasoning-centric reward design. The work emphasizes broad RL methods (e.g., RLHF, RLAIF, DPO, GRPO), reward-guided decoding, and post-hoc correction; however, it does not explicitly map how reward modeling interacts with core reasoning challenges, such as hallucination control, planning-based inference, dynamic evaluation, or efficiency–accuracy trade-offs during test-time scaling.

In contrast, our work introduces Reasoning-Aligned Reinforcement Learning (RARL), a reasoning-centric and taxonomic perspective on reward modeling for LLMs. RARL organizes a broad spectrum of paradigms—including RLHF, preference optimization methods such as DPO, reinforcement learning from AI feedback (RLAIF), LLM-as-a-judge frameworks, and reinforcement learning with verifiable rewards (RLVR)—under a shared analytical lens focused on explicitly aligning models with reasoning objectives across diverse disciplinary settings, accounting for challenges such as hallucination suppression, LLM bias in LLM-generated supervision, and reward hacking. This shared perspective is designed to afford several

analytical advantages. (1) *Paradigm Complementarity.* The perspective highlights complementary strengths across paradigms to address each other's limitations. For example, RLAIF leverages AI models to provide automated rewards (e.g., ranking candidate answers), significantly reducing human annotation cost; however, it suffers from inherent biases such as self-preference bias (Wataoka et al., 2024) and position bias (Shi et al., 2024). In contrast, preference-based methods like DPO are computationally efficient and stable, but rely on fixed preference pairs and lack adaptivity to evolving model behavior during training. RARL provides an organizing lens, as in Fig. 2, that enables the categorization of reward semantics and facilitates the creation of more diverse reward annotations (see Section 2.1.1 for reward semantics). (2) *Cross-Domain Methodological Transfer.* It enables the transfer of insights and solutions across traditionally separate research areas. For example, Monte Carlo tree search for step-level reward estimation, originating in the math PRM literature (Wang et al., 2023b; Zhang et al., 2024a), reappears in ReasonRAG (Zhang et al., 2025d) for augmented reasoning (Section 4.3) where the shared process supervision principle in the RARL taxonomy makes this transfer visible. More broadly, reward hacking—a critical failure mode in reward design—can be better detected and mitigated by jointly analyzing LLM-intrinsic biases and the dynamics of RL optimization, rather than treating reward failures as isolated implementation issues. (3) *Systematic Failure Mode Characterization.* It facilitates the identification of more systematic and fundamental challenges in reasoning-oriented reward modeling. By viewing reward mechanisms as a shared abstraction rather than paradigm-specific tools, RARL helps expose common issues such as credit assignment errors, distributional bias, and misalignment between reward signals and true reasoning quality. This perspective highlights how reward signals can be systematically constructed and evaluated to directly target reasoning quality, both at the level of outcomes and intermediate processes. Furthermore, RARL provides a unifying lens for understanding how reward mechanisms can be adapted across related research areas beyond standalone RL fine-tuning. We examine how reward design principles extend to different settings such as retrieval-augmented generation (RAG), and how reward signals can assist to improve the trustworthiness and reliability of model feedback. By synthesizing methods, challenges, and applications within a coherent taxonomic survey, our work aims to establish a comprehensive foundation for future research on reward modeling and reasoning alignment in large language models.

**What RARL adds beyond a reading list.** Table 1 summarizes the key differences between RARL and four concurrent surveys across coverage dimensions most relevant to reasoning-centric reward design. Rather than simply cataloguing reward design choices in isolation, RARL provides a shared analytical vocabulary that makes cross-paradigm comparisons tractable. For instance, framing RLHF, RLVR, and RLAIF under a common MDP formulation (Section 2) exposes that their differences lie primarily in reward semantics rather than algorithmic structure, a non-obvious observation that motivates the three-axis taxonomy. Similarly, organizing failure modes under RARL's reward-centric lens reveals that credit assignment bias, length bias, and faithfulness bias are not independent bugs but manifestations of a single underlying tension between proxy reward optimization and true reasoning quality (Section 3). This perspective does not generate formal predictions, but it does generate a diagnostic checklist with demonstrated utility for practitioners designing reward pipelines.

## 1.2 Methodology

This survey was compiled through a systematic search of arXiv (cs.LG, cs.CL, cs.AI), Semantic Scholar, and major venue proceedings (NeurIPS, ICML, ICLR, ACL, EMNLP), covering publications from January 2022 through April 2025. Papers were included if they directly addressed reward signal design, reward model training or evaluation, or RL-based fine-tuning with an explicit reasoning objective; works focused solely on non-language RL domains were excluded unless their methodology transferred directly to LLM reasoning. We prioritized peer-reviewed publications but included preprints for widely adopted methods (e.g., DeepSeek-R1) or where no peer-reviewed counterpart yet exists—a significant share of cited works from late 2024 and early 2025 are non-peer-reviewed, and conclusions from those works should be considered preliminary. The resulting corpus comprises over 250 papers. We acknowledge that coverage cannot be exhaustive in this fast-moving field; our goal is representativeness across the major paradigms and failure modes identified in Section 2.

Table 1: **Comparison of RARL with concurrent reward modeling surveys.** ✓ = covered substantively; ○ = partially covered; ✗ = not covered.

| | **Ours (RARL)** | **Zhong et al. 2025** | **Liu et al. 2025d** | **Zheng et al. 2025b** | **Wu et al. 2025** |
|---|---|---|---|---|---|
| **Primary focus** | Reward design for reasoning | General RM taxonomy | RM for reasoning | PRM (incl. multimodal, agents) | Learning from rewards (broad) |
| **Reward design taxonomy** | ✓ ORM / PRM 3-axis: arch., granularity, semantics | ✓ Preference collection + modeling | ✓ ORM / PRM / generative | ✓ PRM data, build, usage | ✓ ORM / PRM / generative; by stage |
| **Reward hacking & bias** | ✓ 5 bias types + mitigations | ○ Future discussion | ○ Brief mention | ✗ | ○ Brief mention |
| **Hallucination & LLM bias** | ✓ Sycophancy, diversity collapse | ✗ | ○ Multimodal only | ✗ | ✗ |
| **Augmented reasoning** | ✓ RAG, TIR, Tables | ✗ | ✗ | ✗ | ✗ |
| **RL training challenges** | ✓ Diversity collapse, entropy | ✗ | ○ Via RL algorithm descriptions only | ✗ | ✗ |
| **Benchmark & evaluation** | ✓ Contamination analysis, defining attributes, multimodal & PRM benchmarks | ○ RM benchmarks listed | ○ RM + multimodal benchmarks listed | ○ PRM benchmarks listed | ○ Benchmarks listed only |
| **Domain applications** | ✓ Medicine, finance, science | ○ Listed only | ✗ | ✗ | ✗ |
| **Reasoning-centric lens** | ✓ Explicit throughout | ✗ General align. | ✓ | ✓ | ○ |

## 2 Reward Design in Reasoning-Aligned Reinforcement Learning

With the success of models such as OpenAI o1 (Jaech et al., 2024) and DeepSeek (Guo et al., 2025a) in reasoning tasks, RL has emerged as a powerful paradigm for fine-tuning reasoning models. This fine-tuning problem can be naturally formalized as a Markov Decision Process (MDP), denoted as $\mathcal{M} = (\mathcal{S}, \mathcal{A}, P, r, H)$, where the LLM reasoning model acts as the agent. The agent's behavior is governed by a parameterized policy $\pi_\theta$. The components of the MDP are defined as follows:

- State space $\mathcal{S}$: represents contextual steps in the reasoning process (e.g., partial solutions in mathematical problem-solving).
- Action space $\mathcal{A}$: corresponds to the discrete token vocabulary $\mathcal{V}$ of the model.
- Reward function $r : \mathcal{S} \times \mathcal{A} \to \mathbb{R}$ provides feedback signals to guide policy optimization. These rewards can be obtained either from rule-based functions (e.g., correctness checks, format constraints) or from parameterized models which are often initialized from pretrained language models.

**Reasoning-Aligned Reinforcement Learning**

Figure 2: An organizing taxonomy of reward paradigms in RARL across three axes: data annotation (left: automatic via RLAIF; right: human via RLHF), training algorithm (left: without reward model; right: with reward model), and evaluation approach (left: more general; right: more customized). Each paradigm is shown alongside its principal limitations, which the taxonomy makes available for cross-paradigm comparison.

- Dynamics $P(s_{t+1}|s_t, a_t)$: defines the transition model between states.
- Horizon $H$: corresponds to the maximum response length.

At each time step $t$, the state is defined as $s_t = (x, y_{<t})$ where $x$ is the given prompt and $y_{<t}$ is the sequence of tokens generated so far. The action $a_t = y_t$ is the token generated at step $t$. Throughout this paper, we use $(s_t, a_t)$ when referring to the general MDP formulation, and $(x, y)$ when referring to the prompt-response representation used in specific algorithms; these are related by $a_t = y_t$ and $s_t = (x, y_{<t})$. The action $a_t$ is the next token $y_t$ to be generated. The transition function $P$ is deterministic, i.e., upon taking action $a_t$ in state $s_t$, the next state is $s_{t+1} = (x, [y_{<t}, a_t])$. The ultimate objective of RL-based fine-tuning is to learn a policy $\pi_\theta$ that maximizes the expected return, defined as the cumulative reward over the horizon $H$:

$$J(\pi_\theta) = \mathbb{E}_{\pi_\theta} \left[ \sum_{t=1}^{H} r(s_t, a_t) \right].$$

**Reward Granularity and the MDP.** The reward function $r : \mathcal{S} \times \mathcal{A} \to \mathbb{R}$ defined above operates at the token level, but the methods surveyed in this paper assign rewards at three distinct granularities that map onto this MDP differently. *Outcome-level rewards* assign a scalar only at the terminal step $t = H$, i.e., $r(s_t, a_t) = 0$ for all $t < H$ and $r(s_H, a_H) = R(x, y)$ where $y = (a_1, \ldots, a_H)$ is the complete response. The per-step objective $J(\pi_\theta) = \mathbb{E}[\sum_t r(s_t, a_t)]$ therefore reduces to $\mathbb{E}[R(x, y)]$, and the reward signal is entirely sparse. This sparsity is the root cause of credit assignment challenges discussed in Section 3. *Step-level rewards* assign a scalar at the boundary of each reasoning step $k \in \{1, \ldots, K\}$, where a step spans a variable number of tokens. Formally, let $\tau_k$ denote the final token index of step $k$; then $r(s_t, a_t) = R_k(x, y_{\leq \tau_k})$ for $t = \tau_k$ and $r(s_t, a_t) = 0$ otherwise. PRMs thus provide a denser signal than ORMs while remaining sparser than token-level rewards, and their per-step objective is $J(\pi_\theta) = \mathbb{E}[\sum_{k=1}^{K} R_k(x, y_{\leq \tau_k})]$. *Token-level rewards* (implicit rewards, e.g., KL-penalized objectives) assign a reward at every $t$, consistent with the stated MDP.

**Deterministic Transitions and Generalization.** Deterministic transitions imply that the state space grows combinatorially: at horizon $H$, there are $|\mathcal{V}|^H$ possible states. Consequently, a reward model $r_\phi$ trained on a finite dataset must generalize to states never observed during training. As the policy evolves during RL optimization, it progressively visits increasingly distant regions of the state space, requiring the reward

model to extrapolate rather than interpolate Gao et al. (2023). Moreover, even small perturbations to a token sequence correspond to distinct and likely unseen states under deterministic dynamics, helping explain why reward scores can vary substantially under minor surface-level reformulations (Yang et al., 2024c).

In RL for LLM reasoning, the reward design can be generally divided into three categories: model-based design, rule-based design, and self-reward. In the model-based design, reward models (RMs) are learned functions trained to predict scalar scores that reflect human preferences or task-specific quality metrics, offering a more flexible and expressive signal. Beyond their role in training, RMs are vital for test-time inference, which leverages these models to select high-quality outputs during test-time inference (Sun et al., 2024b). In contrast, rule-based reward design relies on handcrafted heuristics, such as exact answer matching, logical consistency checks, or syntactic correctness, and is typically sparse and interpretable. Finally, the self-reward paradigm is one in which a policy model leverages feedback from itself to iteratively improve, without relying on additional rules or external models. In practice, these design approaches are often complementary: rule-based design provides a robust foundation, model-based design captures more sophisticated signals that are difficult to encode manually, and self-reward mechanisms offer a pathway for autonomous and scalable self-improvement.

## 2.1 Model-based Reward Design

Training a task-specific reward model (RM) from data to provide nuanced feedback is a central approach in applying RL to complex reasoning tasks, complementing rule-based methods. To facilitate a clear understanding of current model-based design, we organize in this subsection existing approaches along three key dimensions as shown in Fig. 3, including model architecture, model granularity, and reward semantics. First, we present a taxonomy of RM architectures, which are typically initialized from pre-trained language models and adapted into either discriminative or generative paradigms. Second, we analyze the granularity of the reward signals, which can be sparse (outcome-based) or dense (process-based). Third, we categorize the semantics of dense reward signals, which encompass methods such as correctness labeling, value estimation, and reward shaping. These approaches differ in how intermediate rewards are defined and supervised during training.

### 2.1.1 Taxonomy: Types of Model-based Designs

To systematically understand the diversity of reward modeling approaches, we introduce a taxonomy that organizes existing methods from three different perspectives, according to their architectural design, granularity of evaluation, and the semantic meaning of the reward signal, respectively.

**From the perspective of architecture** This dimension classifies reward models based on how they adapt the underlying LLM's architecture to produce an evaluation. More specifically, it distinguishes between discriminative models that use a linear head to output a scalar score, and generative models that leverage the model's next-token predictive ability to generate output, either as text critiques or token probabilities.

*(1) Discriminative Models.* The key idea behind discriminative models is to treat reward modeling as a classification task by adding a linear head atop decoder-only architectures to produce scalar scores (Cai et al., 2024; Lyu et al., 2025; Zhang et al., 2025i; Liu et al., 2024c). The training strategies of such models typically employ either: (1) the Bradley-Terry (BT) loss (Bradley & Terry, 1952) for pairwise comparisons, which maximizes reward margins between winning and losing responses (Liu et al., 2024a; Fan et al., 2025), or (2) binary cross-entropy (BCE) loss for point-wise classification of individual samples.

However, discriminative reward models suffer from several limitations. (1) (*Lack of interpretability*) The single opaque scalar output lacks interpretability and fails to capture nuanced evaluation criteria. (2) (*Temporal inconsistency*) Models trained with pointwise BCE treat each step as an independent classification problem, leading to suboptimal reward assignments. For example, in multi-step reasoning tasks, BCE-trained models may assign high scores to subsequent steps even when initial steps are incorrect, as they evaluate each step in isolation without considering contextual dependencies. (3) (*Task shift and generalization issues*) A fundamental mismatch exists between the backbone model's pre-training objective and the reward head's fine-tuning task (Kumar et al., 2022). This task shift can distort the model's well-pre-trained representa-

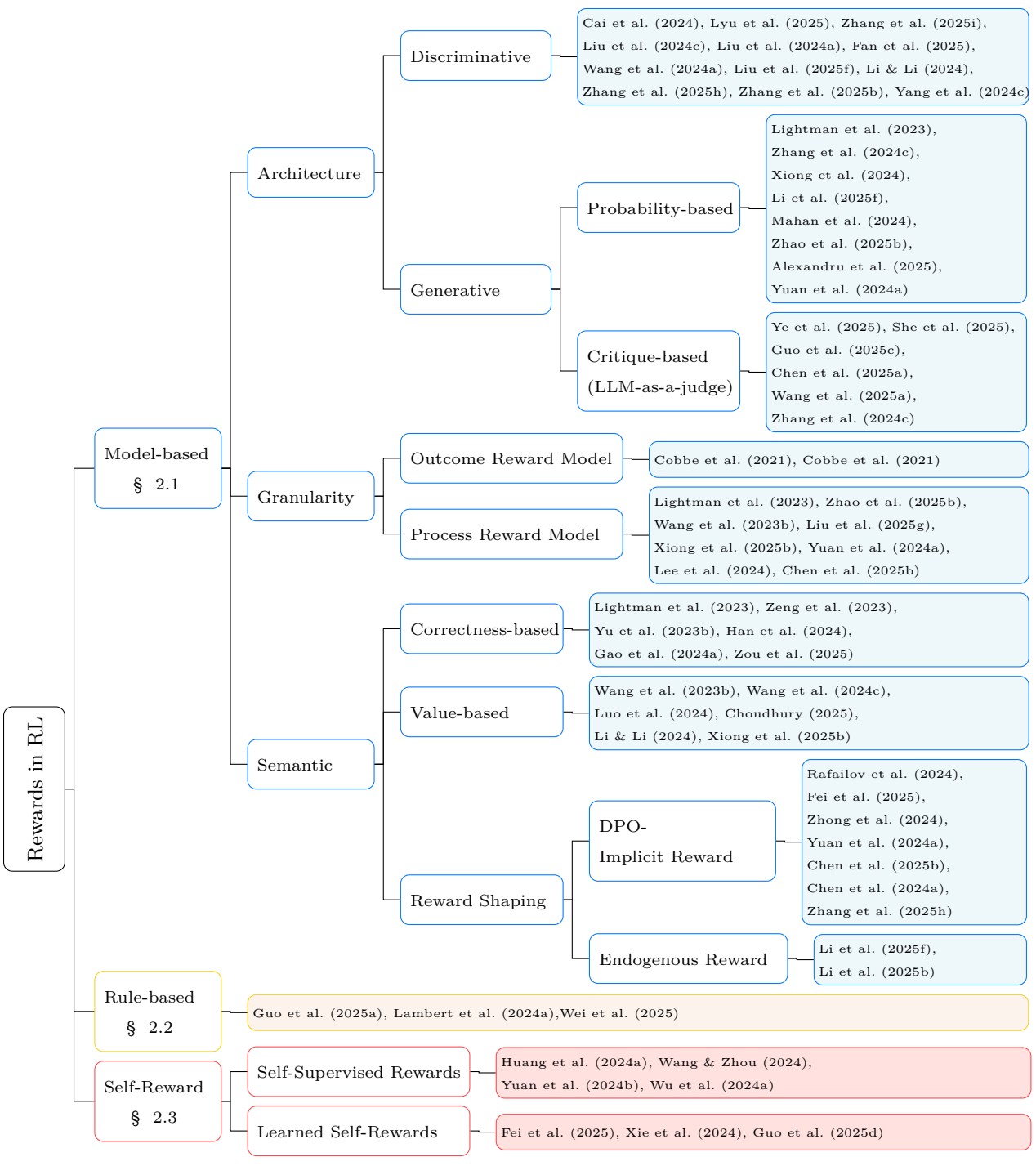

Figure 3: A taxonomy of reward designs for large reasoning language models.

tions and impair its out-of-distribution generalization, making it less robust when faced with data that differs from its training set. (4) (*Insufficient usage of LLMs*) The type of approaches underutilizes the remarkable generative capabilities of modern LLMs.

To address interpretability concerns, Wang et al. (2024a) introduce ArmoRM, a multi-objective framework that employs Mixture-of-Experts scalarization to handle multiple evaluation metrics simultaneously. SRM (Liu et al., 2025f) enhances interpretability through a modular architecture of Side Branch Models (SBMs), where each SBM is explicitly designed to evaluate a specific, fine-grained dimension (e.g., factual accuracy, style matching, or creativity). By generating structured, dimension-specific auxiliary features for a given prompt-response pair, the SRM provides diagnostically useful feedback, explaining why a response is scored in a certain way. For the specific challenge of step interdependency in reasoning tasks, Li & Li (2024) propose the Process Q-value Model (PQM), which reframes process reward modeling as a Q-value ranking problem, explicitly capturing contextual relationships between steps. CRM (Zhang et al., 2025h) further improves PQM by defining each step's reward as a conditional probability dependent on all preceding steps to capture inter-step dependencies. Similarly, Zhang et al. (2025b) introduce TDRM, applying temporal difference learning to dynamically bootstrap intermediate rewards by integrating future estimates at each step. To enhance the model generalization, Yang et al. (2024c) propose the GRM, which employs a linear head as a regularizer during reward training to preserve the backbone's general-purpose features. For leveraging generative capabilities, researchers have developed generative reward models, which we discuss in the following section.

*(2) Generative Models.* While effective in some settings, discriminative models fail to exploit the model's next-token prediction capability and are often criticized for their lack of interpretability and susceptibility to bias (Ankner et al., 2024; Ye et al., 2025). In contrast, generative approaches leverage the model's probability distribution over tokens to create judgments between responses. In addition, they are preferable for tasks that demand robust generalization (Wang et al., 2025a). Depending on the output, the generative models can be further categorized into probability-based and critique-based (or LLM-as-a-judge).

- *Probability-based models* have scalar output, and a common strategy is supervised fine-tuning (SFT), where the model is trained to minimize the loss between the predicted token and the target label (e.g., binary reward tokens). In this setting, the probability of tokens such as "yes," "no," or "neutral" can serve as process rewards (Lightman et al., 2023; Xiong et al., 2024; Zhang et al., 2024c), either for a single state–action pair or conditioned on the entire reasoning trajectory (Mahan et al., 2024; Zhao et al., 2025b). Other works instead use log-likelihood comparisons to assess "chosen" versus "rejected" responses, including chain-of-thought critiques (Alexandru et al., 2025). Some works extend to all the token distribution, where the rewards are assigned as the logits after softmax for probability distribution over tokens (Yuan et al., 2024a; Li et al., 2025f).

- *Critique-based models*, often operationalized under the LLM-as-a-judge paradigm, generate textual feedback—rather than scalar scores—to evaluate model responses. This approach offers greater interpretability compared to black-box discriminative scorers, as it exposes the rationale behind each judgment. For instance, Con-J (Ye et al., 2025) produces both preference predictions and accompanying explanations. Further advancing this line, Guo et al. (2025c) introduce a training framework that cultivates self-evolved reasoning capabilities without relying on explicit supervision, enabling the model to cooperate effectively with rule-based reward signals. R-PRM (She et al., 2025) leverages stronger LLMs to generate comprehensive step-by-step assessments, effectively bootstrapping reasoning capabilities from limited annotations. ReasonGRM (Chen et al., 2025a) produces preference justifications by first generating and then filtering its training data. It uses a metric, $R^*$ which scores candidate reasoning paths based on the model's internal token probabilities, to select paths that are both self-consistent and lead confidently to the correct answer. Beyond interpretability, generative reward models exhibit stronger data generation ability and generalize more robustly to out-of-distribution datasets than their discriminative counterparts (Wang et al., 2025a). They also afford greater flexibility, seamlessly supporting chain-of-thought reasoning and other long-form analytic processes within a unified generative architecture. This flexibility further enables the use of inference-time compute strategies, such as majority voting (Zhang et al., 2024c; Guo et al., 2025c), to enhance decision quality.

**From the perspective of model granularity**   Based on the sparsity of reward signals, parameterized reward models are commonly classified into two categories: outcome reward models (ORMs) that assign a reward at the final answer, and process reward models (PRMs) which provide fine-grained intermediate rewards during the reasoning process.

*(1) Outcome Reward Model (ORM).* The idea of using a verifier or ORM to evaluate the final answer of generated solutions was first proposed by Cobbe et al. (2021) with the widely used math reasoning dataset, GSM8K. Outcome supervision is label-efficient, offering faster massive data collections and clearer data annotations. EORM (Jiang et al., 2025a) extends the ORM framework by representing outcome-level correctness using an energy function where lower values indicate higher quality of responses. During inference, the energy is inverted to produce a scalar reward signal, enabling smooth preference-based optimization while relying solely on outcome supervision.

*(2) Process Reward Model (PRM).* The fact that LLMs can generate correct final answers with incorrect reasoning procedures challenges the usage of outcome supervision alone. Therefore, PRMs have been proposed to evaluate the logical soundness of each step, providing a fine-grained assessment beyond final-answer accuracy. Common granularities of PRMs include step-level and token-level supervision. Step-level PRMs assign rewards to discrete reasoning steps, typically corresponding to intermediate conclusions or equations, enabling models to identify and correct faulty logic during multi-step reasoning (Lightman et al., 2023; Zhao et al., 2025b; Wang et al., 2023b). These models generally segment responses into steps using predefined symbols (e.g., "Step 1"). However, such fixed segmentation may omit critical decision-making cues, and the inherent ambiguity in defining granular reasoning steps poses a challenge in the PRM design. Liu et al. (2025g) propose an adaptive approach that determines step boundaries based on model confidence, i.e., the probability of the sampled token, which indicates where the model is uncertain about starting a new step, and StepWiser (Xiong et al., 2025b) partitions reasoning by prompting an LLM to apply a segmentation principle emphasizing unified purpose, logical cohesion, and smooth transitions. In contrast, token-level PRMs operate at a finer granularity, assigning rewards to individual tokens, which allows for continuous feedback throughout the reasoning trajectory (Yuan et al., 2024a; Lee et al., 2024; Chen et al., 2025b).

Compared to ORMs, PRMs require more extensive label annotations. Early studies relied on human annotators to provide step-level ground-truth labels; nevertheless, the high cost and effort of such manual annotation make the training process difficult to scale. Therefore, defining meaningful intermediate rewards during training without process label is crucial for practical PRM development. Several techniques have been proposed to address this challenge other than correctness, including **value-based methods**, which estimate step- or token-level value functions as surrogate labels, and **potential-based reward shaping**, which derives equivalent intermediate reward functions from theoretical formulations. These methodologies aim to provide more scalable and informative supervision signals for process-level reasoning and are discussed in greater detail in the following sections.

**From the perspective of reward semantics**   Reward models also differ in how the reward signal itself is conceptualized and obtained. Three primary perspectives emerge: correctness-based, value-based reward modeling, and potential-based reward shaping.

*(1) Correctness-based.* A primary example is the work of Lightman et al. (2023), who use PRM800K—a human-labeled dataset derived from MATH—to provide fine-grained step-level correctness labels. To improve sample efficiency and reduce annotation costs, they employ active learning to prioritize convincing but incorrect solutions for human review. The resulting large-scale PRM is then used as an auto-labeler to generate synthetic process and outcome labels for training smaller models. MR-GSM8K (Zeng et al., 2023) is derived from GSM8K dataset and prompted with MetaMath (Yu et al., 2023b). It includes code-based solutions and reverse reasoning tasks where one input is concealed and the model is asked to infer it from the provided answer. Each solution is annotated by humans for overall correctness, the first error step, and detailed step-wise error analysis. For first-order logic reasoning problems, P-FOLIO (Han et al., 2024) features human-annotated, step-level proofs from the FOLIO dataset.

Given the high cost of human annotation, other works leverage stronger LLMs as judges to provide broader reward signals. For instance, Gao et al. (2024a) enrich reward modeling by first training the model to repro-

duce GPT-4's natural language feedback token-by-token, thereby internalizing detailed reasoning patterns. The model is then fine-tuned with binary correctness labels from (Wang et al., 2023b) to function as a discriminative reward model capable of efficient scalar reward prediction. To address the limitation that prior PRMs struggle to evaluate the quality of intermediate reasoning steps, ReasonFlux-PRM (Zou et al., 2025) introduces a composite step-level reward. This reward integrates three signals beyond simple correctness: a quality score from a GPT-4 judge assessing logical soundness, a coherence score for contextual compatibility between adjacent steps, and an alignment score based on semantic similarity with the final response.

*(2) Value-based.* To bypass labor-intensive labeling, value-based reward models estimate the probability of reaching the final correct answer or effectively a Q-value as a reward at each step, with Monte Carlo (Wang et al., 2023b; 2024c) either for constructing an offline dataset or reward assignment in online PRM training. Classical examples include Math-Shepherd (Wang et al., 2023b), OmegaPRM (Luo et al., 2024), AgentPRMs (Choudhury, 2025), PQM (Li & Li, 2024), and StepWise (Xiong et al., 2025b), which label their training data with estimated Q-value by sampling multiple reasoning rollouts from each step. There are two types of reward labels: a hard estimate (a binary reward if any rollout succeeds) and a soft estimate (the empirical success rate across all rollouts) (Wang et al., 2023b).

However, this Monte Carlo-based labeling paradigm suffers from several limitations. *First*, early-step bias, where labels for initial steps are less accurate due to higher uncertainty and exhibit a significantly lower proportion of minimum scores at the final step (Zhang et al., 2025i), is a general failure to provide meaningful feedback on difficult problems or reliably identify erroneous steps (Sun et al., 2025b). A critical consequence is that these step-level value estimates can be overly pessimistic, leading to the premature discarding of potentially correct reasoning paths during search at inference time. To address this, TVM (Lee et al., 2024) extends value estimation to the token level to identify and retain promising paths. *Second*, the value labels are from a random rollout, which may lead to a correct answer but an invalid trajectory. ReST-MCTS (Zhang et al., 2024a) addresses this by using a PRM to guide tree search to strategically explore the reasoning space rather than random rollouts for value estimation. It can therefore achieve more accurate value estimation by iteratively training both policy and PRM on curated search traces. *Third*, since Monte Carlo requires generating multiple reasoning paths by simulating different step-by-step solutions, the computational cost remains substantial. To improve sample efficiency, later studies have explored learned value estimators that directly approximate expected future returns and data augmentation as follows. *i).* **Value estimator**: SORM (Havrilla et al., 2024) replaces explicit sampling with a supervised process-level value function model trained entirely on synthetic data, providing more accurate and less pessimistic assessments of reasoning steps. Similarly, OVM (Yu et al., 2023a) employs outcome supervision to train a value model that prioritizes reasoning paths leading to correct conclusions, formally framing guided decoding as a value-estimation problem. *ii).* **Data augmentation**: HPM (Wang et al., 2025d) improves data efficiency through Hierarchical Node Compression, augmenting training data by merging consecutive reasoning steps to better capture intermediate dependencies.

Although value-based methods can avoid annotating step correctness, studies show that correctness, which evaluates past reasoning steps and represents the probability that the existing steps are correct, and value estimation, which estimates the likelihood that the reasoning will lead to a successful final solution, could be orthogonal (Wu et al., 2025d). DuaShepherd (Wu et al., 2025d) leverages this distinction by designing a two-dimensional reward model with two separate value heads, which improves performance compared to using a value-based method alone.

*(3) Potential-based Reward Shaping.* Potential-based reward shaping (Ng et al., 1999) involves modifying the reward function to improve the learning process without altering the optimal policy, and it is commonly used to solve the credit assignment problems to provide dense rewards (Zou et al., 2019; Pignatelli et al., 2024). Since designing intermediate rewards requires domain knowledge, early works (Goyal et al., 2019; Waytowich et al., 2019) start to use language models to provide instructions for agents. In the reasoning tasks where step-level reasoning is hard to determine its correctness and notoriously hard to annotate step-wise labels, reward shaping provides a good means to enrich intermediate guidance.

**DPO-Implicit Reward.** Reward shaping offers a theoretical foundation for enriching rule-based rewards with dense signals. A prominent approach is the use of implicit reward (Rafailov et al., 2024), defined as

$\beta \log \frac{\pi^*(s_t, a_t)}{\pi_{\text{ref}}(s_t, a_t)}$ where $\pi^*$ is the optimal policy. This form of reward is embedded directly within a model tuned via Direct Preference Optimization (DPO) (Rafailov et al., 2023), enabling token-level, label-free supervision. For instance, Reinforced Token Optimization (RTO) Zhong et al. (2024) pre-train a DPO model to function as a standalone dense reward generator for subsequent PPO training. Subsequent work has adapted this implicit structure to $\beta \log \frac{\pi_\theta(s_t, a_t)}{\pi_{\text{ref}}(s_t, a_t)}$, facilitating free process label annotation and eliminating the need for costly human supervision (Yuan et al., 2024a; Fei et al., 2025).

However, this paradigm suffers from fundamental limitations. *First*, during training, the optimal policy $\pi^*$ is unknown and must be approximated by the current policy $\pi_\theta$. This creates a conflict, as the reward signal becomes entangled with the policy's own, potentially suboptimal, generation probabilities. Furthermore, the inherent reliance on a reference model $\pi_{\text{ref}}$ introduces a source of uncertainty, which can lead to inaccurate credit assignment, e.g., disproportionately rewarding non-critical tokens while neglecting key ones. To address these issues directly, the Q-function Reward Model (Q-RM) (Chen et al., 2025b) decouples reward modeling from language generation. Instead of deriving rewards from a generative policy, Q-RM trains a dedicated discriminative model to output token-level Q-values directly from preference data. This approach provides fine-grained, policy-agnostic rewards, eliminating the reliance on a reference model and resolving the core conflicts that plague implicit reward methods. Considering length bias, DICE (Chen et al., 2024a) designs the implicit reward with length penalty (length-regularization) reward shaping (see 3.2), and incorporates continual learning to avoid overdependence on implicit rewards. *Second*, the outcome reward is the logarithmic sum of process (Yuan et al., 2024a), which is not linked with the outcome and therefore fails to capture inter-step dependencies. To overcome this, CRM (Zhang et al., 2025h) models the probability of remaining in a correct reasoning state, denoted $S(t)$, and derives the step reward as $r_t = \log(1 - h(t))$, where $h(t)$ is the conditional probability of entering a wrong state at step $t$. This formulation ensures that the sum of process rewards equals the log-probability of reaching the correct final answer, thereby establishing a causal link between process-level and outcome-level rewards. Note that DPO-implicit rewards differ from self-reward 2.3 in that they derive from a reference model, not the policy's own outputs.

**Endogenous Reward.** A distinct and theoretically grounded approach to reward shaping emerges from the connection to Inverse Reinforcement Learning (IRL). A growing body of work challenges the conventional view of supervised fine-tuning (SFT) as simple imitation learning, reframing it as a powerful mechanism for learning implicit rewards. Li et al. (2025f) demonstrate that a powerful, generalist reward model is already latently present within any LLM trained via standard next-token prediction. Under the assumptions that the training data constitutes near-optimal expert demonstrations, and language generation is formalized as a token-level MDP with deterministic transitions, they prove that the model's logits represent the optimal Q-function for a specific offline IRL objective, allowing a principled reward function—the *endogenous reward*—to be derived directly using the inverse soft Bellman operator. This results in a reward signal that is inherently shaped, as it can be expressed in a form equivalent to the log-probability of an action $\log \pi_{\text{ref}}(a_t|s_t)$. Under these conditions, this provides a training-free method to obtain dense, shaped rewards in cross-domain tasks. Concurrently, building on similar IRL foundations, Li et al. (2025b) establish a formal equivalence between token-level SFT and Inverse Q-Learning, showing that SFT implicitly learns dense, token-level rewards that explain expert demonstrations. This leads to practical methods like Dense-Path REINFORCE (Li et al., 2025b), which extracts baseline-relative rewards as $\log \pi_{\text{SFT}}(a_t|s_t) - \log \pi_{\text{ref}}(a_t|s_t)$ and uses them for token-level policy improvement. This approach consistently outperforms SFT baselines across multiple instruction-following benchmarks.

> **Takeaways** *§2.1 — Model-Based Rewards*
>
> Generative process reward models (PRMs) tend to outperform discriminative approaches in both interpretability and out-of-distribution generalization. While Monte Carlo value estimation remains the dominant labeling strategy, it introduces bias toward early reasoning steps. Notably, correctness and value signals capture complementary aspects of solution quality; combining them leads to consistent performance gains. Finally, approaches that decouple reward modeling from policy optimization are generally preferable to DPO-based methods, which can entangle rewards with the reference model.

Table 2: **Trade-off Analysis of Reward Design Paradigms**. **Green** labels indicate favorable properties; **Red** labels indicate unfavorable properties; **Amber** labels indicate moderate trade-offs.

| | Model-based | Rule-based (RLVR) | Self-reward |
|---|---|---|---|
| **Sample efficiency** | **Low** — large annotated datasets required for RM training | **High** — binary signals need no annotation | **Medium** — no external labels but needs sufficient rollouts |
| **Annotation cost** | **High** — human labels or strong LLM judges required | **Low** — relies on verifiable ground truth only | **Low** — self-generated signal, no external labeling |
| **Reward signal density** | **High** — step- or token-level feedback available | **Low** — sparse binary outcome signal | **Medium** — self-consistency provides moderate density |
| **Reward hacking risk** | **Medium** — RM can be gamed; mitigated by ensembles, KL reg. | **Low** — verifiable signals are hard to exploit | **High** — compounding errors; spurious gains from contamination |
| **OOD generalization** | **High** — generative PRMs transfer across domains | **Medium** — limited to tasks with verifiable answers | **Low** — systematic errors reinforce in-distribution biases |
| **Interpretability** | **Medium** — critique-based models explain; discriminative models opaque | **High** — rules are explicit and auditable | **Low** — internal reasoning process is implicit |
| **Scalability** | **Medium** — RM retraining needed as policy improves | **High** — rule functions scale without retraining | **High** — self-improving loop scales automatically |
| **Potential issues** | early-step bias in MC labeling; reference model entanglement in implicit rewards; temporal inconsistency across steps | sparse rewards cause gradient vanishing, entropy collapse | compounding errors amplify over iterations; spurious gains from data contamination |
| **Use cases** | Open-ended generation, nuanced reasoning, RLHF alignment | Math, code, reasoning with verifiable answers | Instruction-following, low-resource settings, continual refinement |
| **Representative methods** | Math-Shepherd, OmegaPRM, GenPRM, Q-RM | DeepSeek-R1, RLVR | Self-play DPO, SMART, Dense-Path REINFORCE |

## 2.2 Rule-based Reward Design

Despite the success of PRMs, several challenges remain. Particularly, it is difficult to verify the correctness of each step, which makes defining appropriate rewards challenging and can ultimately result in reward hacking (Guo et al., 2025a). Rule-based reward design relies on verifiable signals, such as accuracy and format alignment. These rewards are typically sparse and binary (correct vs. incorrect). For example, DeepSeek-R1 defines a rule-based reward consisting of an accuracy component, which evaluates whether responses are correct and follow a specific format (e.g., within a box), and a format component, which

encourages the model to generate its reasoning process between "`<think>`" and "`</think>`" tags. This RL paradigm, incentivized with rule-based reward, is formalized as Reinforcement Learning with Verifiable Feedback (RLVR) (Lambert et al., 2024a), where the verifiable reward $v$ is defined as

$$v(x, y) = \begin{cases} \alpha, & \text{if correct,} \\ 0, & \text{otherwise,} \end{cases}$$

and the RLVR objective function is

$$\max_{\pi_\theta} \ \mathbb{E}_{y \sim \pi_\theta(x)} \left[ v(x, y) - \beta \, D_{\mathrm{KL}}(\pi_\theta(y \mid x) \,\|\, \pi_{\mathrm{ref}}(y \mid x)) \right],$$

where $D_{\mathrm{KL}}$ is the KL divergence.

The simplicity and unbiased nature of rule-based rewards makes them inexpensive, interpretable, and less susceptible to reward hacking. However, such rewards are often sparse, which exacerbates credit assignment and leads to optimization instability, particularly in the early stages of training. To mitigate these issues, one line of work augments discrete rule-based rewards with dense auxiliary signals, while another focuses on optimization strategies designed to stabilize gradient updates under sparse or delayed rewards.

**Combining Sparse and Dense Reward.** One body of work is to design a way to combine dense reward from reward models and rule-based reward while ensuring stable training and correct credit assignment to prevent reward hacking (see 3.1). Instead of relying on a single binary outcome reward, the Qwen2.5-Math series (Yang et al., 2024a) design a reward function that combines a sparse binary reward with a scalar solution-level reward from a trained reward model. This provides more granular signals that reflect the overall quality of the reasoning process, not just the correctness of the final answer. PPR (Xu et al., 2025) trains a PRM that grounds step-wise judgments in interpretable principles (e.g., correctness, relevance) instead of ad-hoc heuristics. Reward is calibrated and unifies the scales of sparse outcome and dense process rewards before advantage estimation. A contrasting approach seeks to leverage general-purpose LLMs as PRMs without additional training. CAPO (Xie et al., 2025) initializes all tokens in a reasoning chain with a sparse, outcome-based reward and then applies a penalty to tokens identified as part of erroneous steps by LLMs as PRMs. TDPM (Zhang et al., 2025b) also leverages the strength of rule-based rewards, utilizing its discriminative PRM as the final reward.

**Gradient Instability in Discrete Rewards.** Discrete rewards produce all-or-nothing learning signals, which fundamentally hinder policy gradient estimation during early training, when the policy rarely generates fully correct answers. Razin et al. (2025) provide theoretical insights showing that perfectly accurate but low-variance rewards can paradoxically impede optimization by yielding weak or unstable gradients. Empirically, Wei et al. (2025) observe that GRPO suffers from gradient vanishing and explosion, driven by insufficient exploration and reward-induced gradient instability, which together substantially slow or prevent convergence. To address these issues, they propose reward dithering, a technique that smooths discrete rewards by adding independently sampled zero-mean perturbation noise. By computing GRPO advantages using these smoothed rewards, reward dithering mitigates gradient vanishing and improves training stability and convergence speed.

> **Takeaways**    *§ 2.2 — Rule-Based Rewards*
>
> Binary outcome rewards are a robust, hack-resistant backbone for verifiable tasks; results like DeepSeek-R1 show they can support strong emergent reasoning. Limitation: reward sparsity leads to vanishing gradients and early exploration collapse; surprisingly, low-variance high-accuracy rewards may also slow optimization. Best practice: combine with a lightweight dense PRM or step-level evaluator, and use reward dithering when training with pure binary signals on hard tasks.

### 2.3 Self-Reward

We define self-reward as a reward design paradigm in which a policy model leverages feedback from itself to iteratively improve, without relying on additional rules or external models. However, it is worth noting

the spurious reward issue under this framework, where some reported improvements from using confidence or entropy may stem from data contamination rather than genuine self-improvement (Chandak et al., 2025) (see 4.4.2). Broadly, self-reward approaches can be categorized into two types: self-supervised methods based on the model's own self-evaluation abilities, and learned self-rewards derived from either labeled data or the model's internal representations.

**Self-Supervised Rewards.** These methods evaluate the correctness of the model's own outputs and use this signal to guide future generations. Common examples include self-consistency (majority voting), maximum-likelihood optimization where self-reward is expressed as $\log \pi_{\text{base}}(y|x)$ (Huang et al., 2024a) where $\pi_{\text{base}}(y|x)$ is the base policy, and model confidence (Wang & Zhou, 2024). Yuan et al. (2024b) demonstrate that a model can simultaneously improve both its instruction-following capabilities (actor) and its reward-modeling abilities (judger) through an iterative DPO framework, in which the language model generates and scores its own training data. This creates a virtuous cycle of self-improvement that overcomes the limitations of fixed reward models trained solely on human preferences. However, Wu et al. (2024a) argue that this framework tends to overlook the judger's capabilities, focusing primarily on refining better responses. To address this, they introduce a meta-judge, whose task is to evaluate the model's own judgments and combine judge score with length information to prevent length bias (see 3.2) from the judging process.

**Learned Self-Rewards from Data.** Some approaches enable models to learn self-reward, which can also be derived from labeled data, by leveraging a base model to generate the reward signal. For example, Fei et al. (2025) utilize DPO-implicit reward (see 2.1.1), i.e., the ratio of the base model and policy model, thereby eliminating the need for separate reward models. Furthermore, recent work has shown that the internal states of LLMs contain rich information about the probability that a reasoning step is correct (Xie et al., 2024). Building on this idea, Guo et al. (2025d) propose a lightweight reward model that utilizes the LLM's own hidden states. Their method extracts token-level reward signals by applying a linear transformation to the concatenated and flattened hidden states from all layers (after residual connections). A gating mechanism, also derived linearly from the same hidden states, dynamically weights each token's contribution according to its relevance to the overall correctness score. The resulting reward is then trained using binary cross-entropy with binary labels.

> **Takeaways** *§ 2.3 — Self-Reward*
>
> Self-reward methods replace external supervision with model-internal feedback, using either self-evaluation signals or rewards learned from data and hidden states. While they enable scalable, iterative self-improvement (e.g., actor–judger loops), they are prone to spurious rewards, bias, and weak judge calibration. Emerging solutions—such as meta-judging and representation-based reward extraction—highlight that the key challenge is not generating feedback, but ensuring its reliability.

## 2.4 Integrating Rewards into RL Algorithms

The reward paradigms introduced earlier can all be integrated as feedback signals within different RL training frameworks. These frameworks typically operate in either online or offline settings. Online reward optimization allows the policy model to interactively explore and receive feedback during training, while offline reward optimization relies on static datasets containing precomputed or annotated rewards. The choice of paradigm determines the trade-off between stability, computational efficiency, and the capacity for exploration and adaptation. We follow the MDP notation from Section 2: $a_t = y_t$ and $s_t = (x, y_{<t})$. Note that the MDP objective in Section 2 uses undiscounted returns ($\gamma = 1$); the discount factor $\gamma$ appears in GAE below as a variance-reduction hyperparameter and is typically set close to 1 in practice.

**PPO.** Proximal Policy Optimization (PPO) (Schulman et al., 2017) is a widely used online RL algorithm that optimizes a policy using reward signals collected through environment interaction. It employs a clipped objective to ensure stable updates, preventing the policy from diverging too drastically from its previous

version:

$$\mathcal{L}^{\text{PPO}}(\theta) = \mathbb{E}_{\substack{x \sim \mathcal{D}, \\ y \sim \pi_{\theta_{\text{old}}}(\cdot|x)}} \left[ \frac{1}{|y|} \sum_{t=1}^{|y|} \min \left( \rho_t(\theta)\, \hat{A}_t,\ \text{clip}(\rho_t(\theta), 1-\epsilon, 1+\epsilon)\, \hat{A}_t \right) \right]$$

where $\rho_t(\theta) = \frac{\pi_\theta(y_t|x,y_{<t})}{\pi_{\theta_{\text{old}}}(y_t|x,y_{<t})}$ is the probability ratio, and $\hat{A}_t$ is the estimated advantage function. The advantage at each timestep is estimated via Generalized Advantage Estimation (GAE) (Schulman et al., 2015):

$$\hat{A}_t^{\text{GAE}}(\lambda) = \sum_{l=0}^{T-t-1} (\gamma\lambda)^l \delta_{t+l}, \quad \delta_t = r_t + \gamma V_\phi(s_{t+1}) - V_\phi(s_t)$$

where $\delta_t$ is the TD residual, $V_\phi(s_t)$ is the value network's estimate of the return from state $s_t$, $\gamma$ is the discount factor, and $\lambda \in [0,1]$ interpolates between high-bias/low-variance ($\lambda = 0$, reduces to TD) and low-bias/high-variance ($\lambda = 1$, reduces to Monte Carlo returns). Since it is computationally expensive to estimate advantage and value functions, many works use alternative estimations instead as follows:

- Monte Carlo Returns: Cui et al. (2025a); Cheng et al. (2025b) estimate advantages both token-level and outcome-level with REINFORCE Leave-One-Out (RLOO) (Ahmadian et al., 2024):

$$A^i = r_\phi(y^i) - \frac{1}{K-1} \sum_{j \neq i} r_\phi(y^j)$$

  where $y^i$ denotes the $i$-th response and $K$ is the number of samples per prompt.

- Token-Level Credit Assignment: Lyu et al. (2025) distribute final outcome rewards to individual tokens using a trained ORM, enabling fine-grained policy updates without value networks.

- Implicit Value Function: Kiruluta et al. (2025) simplify advantage estimation to $A(x,y) \approx r(x,y) - V_{\theta_{\text{old}}}(x)$, using the old policy's value estimate as a baseline to avoid training separate value networks.

**DPO.** Direct Preference Optimization (DPO) (Rafailov et al., 2023) reframes reward maximization as a preference modeling problem, and operates in an offline fashion, learning directly from pairwise preference data, i.e., winning response $y_w$ and losing response $y_l$.:

$$\mathcal{L}_{\text{DPO}}(\theta) = -\mathbb{E}_{(x,y_w,y_l)\sim\mathcal{D}} \left[ \log \sigma \left( \beta \log \frac{\pi_\theta(y_w|x)}{\pi_{\text{ref}}(y_w|x)} - \beta \log \frac{\pi_\theta(y_l|x)}{\pi_{\text{ref}}(y_l|x)} \right) \right].$$

Although DPO is designed to bypass the need for an explicit reward model, many works leverage its implicit reward $\beta \log \frac{\pi_\theta(y|x)}{\pi_{\text{ref}}(y|x)}$, derived from this framework with soft Q-learning (Rafailov et al., 2024), and extend it to token-level signals for reward model training (Chen et al., 2025b; Cui et al., 2025a) under various RL frameworks and settings (see 2.1.1).

**GRPO.** Group Relative Policy Optimization (GRPO) (Shao et al., 2024) extends the PPO framework by omitting the value network and reward model, therefore increasing training efficiency. The reward model is replaced with rule-based reward function. The objective incorporates group-wise advantage estimates:

$$\mathcal{L}_{\text{GRPO}}(\theta) = \mathbb{E}_{\substack{(x)\sim\mathcal{D}, \\ \{y_i\}_{i=1}^{G}\sim\pi_{\text{old}}(\cdot|x)}} \left[ \frac{1}{G} \sum_{i=1}^{G} \frac{1}{|y_i|} \sum_{t=1}^{|y_i|} \min \left( \rho_{i,t}(\theta)\hat{A}_i,\ \text{clip}(\rho_{i,t}(\theta), 1-\epsilon, 1+\epsilon)\, \hat{A}_i \right) - \beta D_{\text{KL}}(\pi_\theta \parallel \pi_{\text{ref}}) \right]$$

where

$$\rho_{i,t}(\theta) = \frac{\pi_\theta(y_{i,t} \mid x, y_{i,<t})}{\pi_{\theta_{\text{old}}}(y_{i,t} \mid x, y_{i,<t})}, \quad \hat{A}_i = \frac{r_i - \text{mean}(r_1, r_2, \ldots, r_G)}{\text{std}(r_1, r_2, \ldots, r_G)}$$

is the group advantage function.

**DAPO.** Decouple Clip and Dynamic sampling Policy Optimization (DAPO) (Yu et al., 2025b) improves upon PPO and GRPO, where samples in a group tend to be nearly identical, by introducing Clip-Higher to mitigate entropy collapse:

$$\mathcal{L}_{\text{DAPO}}(\theta) = \mathbb{E}_t \left[ \min \left( \rho_t(\theta) \hat{A}_{i,t}, \ \text{clip}(\rho_t(\theta), 1 - \epsilon_{\text{low}}, 1 + \epsilon_{\text{high}}) \hat{A}_{i,t} \right) \right],$$
$$\text{subject to } 0 < |\{y_i \mid \text{is\_equivalent}(c, y_i)\}| < G,$$

where $c$ is the answer of the question prompt $x$. The asymmetric clipping thresholds $\epsilon_{low}$ and $\epsilon_{high}$ decouple the lower and upper clipping bounds to better balance exploration and exploitation. A larger $\epsilon_{high}$ allows large ratio increases for positive-advantage encouraging exploration of better actions, while a smaller $\epsilon_{low}$ restricts ratio decreases, preventing the policy from abandoning actions prematurely.

Despite these algorithmic advances, the behaviors of RL-trained models remain largely dependent on the base model. The initial policy distribution constrains the exploration space, and the effectiveness of reward integration depends on the base model's coverage of high-reward regions. This dependency underscores the importance of base model selection and the interplay between reward design, algorithmic integration, and initial model quality.

# 3 Reward Hacking

In this section, we will focus on the challenges in reward design, particularly reward hacking, or overoptimization, where the agent exploits flaws in the reward function rather than learning the intended behavior, ultimately reducing alignment. Gao et al. (2023) demonstrate this by simulating the relationship between a ground-truth reward and a learned proxy, showing that over-optimizing the proxy can actually decrease the true reward. Common manifestations of reward hacking reported in the literature include reasoning without solving, using too few or overly short steps, producing meaningless repetitions or extraneous reasoning, and arriving at correct final answers through incorrect intermediate logic (Cheng et al., 2025b; Gao et al., 2024b). Since most reasoning models are language models with inherent inductive biases, we consider both the innate biases of LLMs and the reasoning-specific biases introduced by reward design.

These biases distort the alignment signal, encouraging behavior that maximizes apparent reward rather than genuine reasoning or task performance. We categorize these mechanisms as *bias-induced reward hacking*, which can be further divided into several types summarized in Figure 4: (1) structural biases in reward propagation, such as **Credit Assignment Bias**; (2) statistical or presentation-level artifacts, including **Distribution-shift Bias**, **Length Bias**, and **Position Bias**; and (3) semantic misalignments between reasoning and explanation, captured by **Faithfulness Bias** (or *Chain-of-Thought Hacking*). Together, these categories illustrate how subtle imperfections in reward modeling create exploitable shortcuts that undermine both alignment and interpretability.

## 3.1 Credit Assignment Bias

By propagating cumulative gamma-decayed rewards over future steps, standard RL's summation-form credit assignment creates a structural bias where models can compensate for incorrect steps with a few high-reward ones. PURE (Cheng et al., 2025b) solves this by introducing a min-form credit assignment, forcing the model to optimize for the weakest link in its reasoning chain. Since the credit of the entire sequence is determined by its least-correct step, any single incorrect step sharply reduces the sequence's overall value, preventing the model from exploiting patterns such as always emitting "thinking" steps without solving the problem. This mechanism suppresses reward hacking, enabling strong performance with minimal supervision. In contrast, VinePPO (Kazemnejad et al., 2024) addresses PPO's bias value estimation by eliminating the value network and instead using unbiased Monte Carlo estimation. For each state, its value is the average of all the sample rollouts from the current policy.

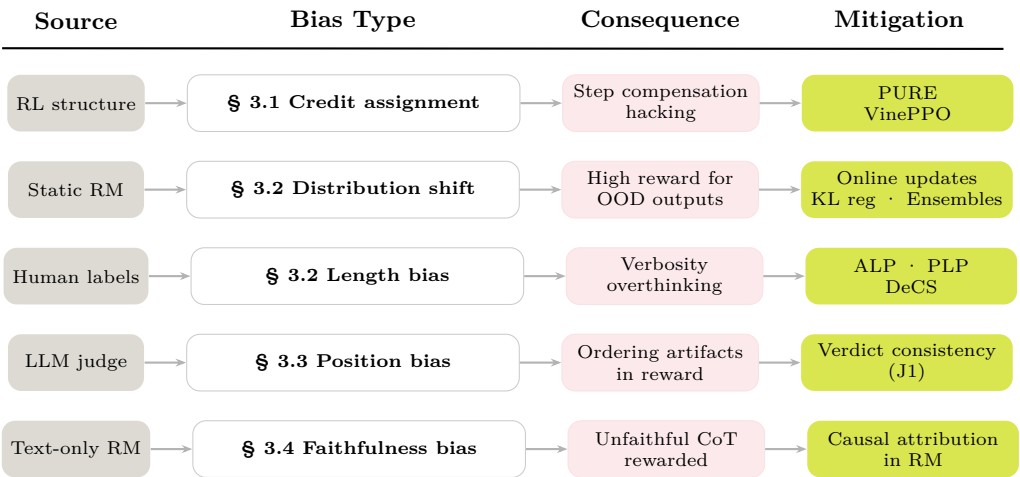

Figure 4: Flow diagram of bias-induced reward hacking: source, bias type, consequence, and mitigation.

## 3.2 Distribution-shift Bias

As reward models are initialized from pre-trained LLMs and subsequently fine-tuned for reinforcement learning, limited coverage of training data can lead to inaccurate predictions during RL optimization. In such cases, the policy may generate misaligned outputs that are assigned incorrectly high estimated rewards—a phenomenon commonly referred to in the offline RL literature as extrapolation error (Fujimoto et al., 2019). The most direct way to mitigate this problem is to adopt an online training paradigm.

*Online Setting.* Instead of relying solely on a static reward model, online approaches continually refine the reward model during training by collecting new data. This dynamic setting helps mitigate reward hacking (Gao et al., 2023) and adapts the reward signal to the evolving capabilities of the reasoning model. For example, Cui et al. (2025a) and Lyu et al. (2025) propose alternating updates between the reward model and the policy model to reduce error accumulation.

*Offline Setting.* In contrast, several techniques have been explored to address extrapolation error without requiring online updates:

- **KL regularization**. Adding a KL regularization term to the reward function can also mitigate reward hacking; not only can it encourage policy exploration and thereby prevent entropy-collapse, but also can ensure the policy does not learn to produce outputs that are too different from those seen by the reward model during training (Stiennon et al., 2020).

$$r(x, y) = r_\phi(x, y) - \beta \log \frac{\pi_\theta(y|x)}{\pi_{\text{ref}}(y|x)}$$

  where $r_\phi$ is the scalar reward from the reward model with trainable parameters $\phi$ for prompt x and the response y. However, the KL regularization term alone has proven inadequate to fully prevent reward hacking, as the policy can still exploit imperfections in the proxy reward model beyond the KL constraint's reach (Gao et al., 2023; Huang et al., 2025).

- **Reward Ensemble.** Another approach, proposed in the offline RL literature, is conservative reward estimation, which addresses reward uncertainty by reducing the tendency of models to make overly optimistic predictions on unseen state–action pairs (Kumar et al., 2020). A common technique is the use of reward ensembles, where multiple models are trained and their predictions are aggregated pessimistically. Building on this idea, Zhang et al. (2024d) develop efficient ensemble algorithms based on linear-layer ensembles and LoRA-based ensembles for mean value prediction, avoiding the need to train multiple full reward models from LLMs. Their results show that LoRA-based ensemble performs better and that ensemble methods can better mitigate reward hacking and lead to improved

performance. Zhai et al. (2023) also train diverse LoRA ensembles via nuclear norm maximization and penalize reward with both KL regularization and uncertainty regularization. To make the training even more efficient, Zhang et al. (2024e) propose a lightweight method for quantifying reward uncertainty by using only the last layer embeddings of the reward model.

Although ensembles are generally more robust than individual reward models, they cannot fully eliminate reward hacking, especially when all ensemble members exhibit similar error patterns. A comparison between pretrained ensembles, where each model is initialized from a separately pretrained LLM with different random seeds, and finetune ensembles, where models share the same pretrained LLM but are fine-tuned with different seeds, demonstrates that greater diversity among ensemble members leads to more effective mitigation (Eisenstein et al., 2023).

- **Data Retrieval.** For reasoning tasks, PRMs often struggle with difficult questions because they are typically trained on simpler datasets, leading to data shift bias. Moreover, PRMs are model-oriented, and variations in base models or model sizes can produce different step distributions for the same questions (Zhu et al., 2025a). To address these issues, Zhu et al. (2025a) propose RetrievalPRM, which employs a two-stage retrieval strategy: first, it retrieves semantically similar questions and reasoning steps from a retrieval database to mitigate data shift, and then it uses the retrieved examples as contextual grounding to enhance step evaluation.

**Length Bias** Length bias is one of the most prominent sources of bias for LLM-as-a-judge (Park et al., 2024a), often favoring longer responses regardless of quality (Zheng et al., 2023; Wang et al., 2023a). Reward Models (RMs), which are typically trained with LLM-based evaluations, exhibit a similar tendency to prefer verbose outputs, even when verbosity undermines informativeness or coherence. This phenomenon was observed in the context of RLHF, where human annotators displayed a preference for longer answers, thereby incentivizing RL algorithms such as PPO and DPO to optimize for length rather than capturing human preferences (Singhal et al., 2023).

This length bias is particularly problematic in reasoning tasks, where PRMs or LLM-as-a-judge may assign higher scores to longer reasoning chains regardless of their logical validity, potentially leading to inefficient reasoning or *overthinking* (Chen et al., 2024c). However, since models often require a minimum number of tokens or reasoning steps to achieve acceptable accuracy (Arora & Zanette, 2025), achieving **efficient reasoning** requires balancing the trade-off between accuracy and length. Consequently, specialized reward designs are needed and can be divided into the following categories, and the reward design function is organized in Table 3.

**Length Debiasing.** These methods prevent exploitation of the reward model or spurious incentives:

- **Correction.** Huang et al. (2024c) correct reward bias using local average rewards, while Zheng et al. (2025a) introduce step-level length penalties derived from a trained bias estimator.

- **Clipping.** Gao et al. (2024b) propose Clip, which caps the process reward at a threshold, and Delta, which bounds the reward change between adjacent steps. These techniques prevent models from gaming the reward through repetition or excessive elongation.

**Length Reward Shaping.** Reward shaping can rescale or bound rewards to stabilize learning and encourage efficient exploration. Fu et al. (2025) propose a reward-shaping technique that applies a sigmoid function to the centered reward, measuring the relative preference of the policy response over a reference, to enable rapid early learning and gradual convergence in PPO training. We emphasize that the notion of reward shaping used here refers specifically to length-based methods, and is distinct from the general discussion of reward shaping in Section 2.1.1.

**Constraint Truncation.** The method evolves by using an indicator function to assign a binary reward when the response length exceeds a certain threshold. For example, ThinkPurne (Hou et al., 2025) gradually relaxes an adaptive target length during RL training and truncates rewards for those that exceed the defined adaptive length. A limitation of truncation is that it imposes no penalty on overly long or incorrect

Table 3: Efficient Reasoning Reward Comparison

| Method | Reward Design |
|---|---|
| ThinkPrune (Hou et al., 2025) | $R(y) = \begin{cases} 1, & \text{if } y = y^* \text{ and } L \le L^*, \\ 0, & \text{otherwise.} \end{cases}$ |
| LASER-D (Liu et al., 2025e) | $R(y) = \mathbb{1}(y = y^*) + \alpha \mathbb{1}(L \le L^*)$ |
| L1 (Aggarwal & Welleck, 2025) | $R(y, L^*) = \mathbb{1}(y = y^*) - \alpha' |L^* - L|$ |
| PLP (Ling et al., 2025b) | $R(y) = \begin{cases} 1 + \frac{\alpha}{L^\gamma}, & \text{if } y = y^*, \\ 0, & \text{otherwise.} \end{cases}$ |
| ALP (Xiang et al., 2025) | $R(y) = \mathbb{1}(y = y^*) - \alpha \cdot |y| \cdot \max\left(\frac{1}{K} \sum_k \mathbb{1}(y^{(k)} = y^*),\ K^{-1}\right)$ |

responses; building on this idea, Liu et al. (2025e) propose LASER, which uses a step function to reward sequences according to target length, and LASER-D, a dynamic and difficulty-aware extension that adapts target lengths based on both question difficulty and the evolving policy. In this design, incorrect responses receive lower rewards than correct ones, but still more than correct responses that exceed the target length. Compared with the truncation method, budget-based or penalty-based methods can provide soft truncation, where length deviating too far away from the target length gets fewer rewards.

**Length Penalty.** Length penalty is a regularization term added in the reward function, as a soft constraint to keep the response length shorter. The reward is typically a combination of verifiable accuracy with a length penalty to balance accuracy and token efficiency. Many of these methods are inherently *hybrid*, combining multiple mechanisms. For clarity and to help readers identify the key techniques employed, we categorize them here according to their dominant length-based component:

- **Budget-Based Reward.** These methods assign query-specific target lengths or budgets, denote as $L^*$ and penalize deviations from them. L1 (Aggarwal & Welleck, 2025) incorporates instruction tokens (e.g., "Think for $n$ tokens") into the prompt and integrates the penalty directly into the reward function. The target budget adaptively decreases during training. In addition to monotonously decreasing the budget over iterations, since harder questions generally require more tokens to answer, it is natural to adaptively assign larger token budgets to difficult tasks. For example, Li et al. (2025h) pre-estimate budgets based on query difficulty and allow users to specify token budgets directly in the prompt.

- **Priority-Aware Reward.** Instead of targeting a desired budget, the length is dynamically adjusted based on the task difficulty or importance. The model decides how long the output should be, guided by the adaptive penalty. For example, powered length penalty (PLP) (Ling et al., 2025b) introduces a multiplicative penalty based on sequence length, which encourages concise solutions for simple queries while allowing longer sequences for complex ones. Similarly, Xiang et al. (2025) propose the Adaptive Length Penalty (ALP), which dynamically scales the length penalty according to the estimated difficulty by calculating the accuracy in K rollout samples per prompt. Easy prompts receive a larger penalty to reduce token usage, whereas harder prompts receive a smaller penalty, allowing the model to spend more tokens reasoning. Integrated into GRPO, CL-R1 (Cheng et al., 2025c) designs the reward based on the longest correct solution within its group, ensuring that easy problems receive strong pressure for conciseness while hard problems are allowed a more generous token budget.

**Fine-Grained Reward.**   Moving beyond sequence-level incentives, this paradigm decomposes the reasoning process to assign rewards at a more granular level, whether to individual steps or tokens. This approach provides more precise learning signals to shape the model's internal reasoning behavior directly. High-entropy tokens (e.g., 'However', 'define', 'Wait') are known to provide valuable information for learning (Wang et al., 2025c), and their dynamics offer theoretical insight for designing these fine-grained rewards.

- **Step-Level.** These methods evaluate and reward the utility of individual reasoning steps. For instance, Verifiable Stepwise Reward Mechanism (VSRM) (Yue et al., 2025a) segments reasoning trajectories using high-entropy tokens (special tokens) as boundaries and generates multiple responses for each sub-rollout. Stepwise rewards are assigned based on the difference in correctness between consecutive steps, with a decay-based propagation mechanism to address reward sparsity. This rule-based approach provides interpretable, verifiable rewards without requiring a PRM, fundamentally encouraging effective steps while penalizing ineffective ones.

- **Token-Level.** Jiang et al. (2025d) identify that standard length penalties inadvertently penalize essential high-entropy tokens in long, correct sequences while rewarding redundant tokens in short ones. To resolve this, DeCS introduces a decoupled token-level reward that surgically distinguishes between high-entropy tokens as necessary reasoning tokens (Necessary Reasoning Prefix, which are the shortest possible tokens to reach the correct final answer) and redundant ones, ensuring precise penalization of overthinking without suppressing exploratory reasoning.

## 3.3   Position Bias

When serving as judges in pairwise comparisons, ranking, and multiple-choice QA, LLMs exhibit significant position bias, favoring responses based on their location in the prompt rather than their quality (Wang et al., 2023a; Ye et al., 2024). This bias is task-dependent and is most pronounced when the quality gap between candidate responses is small, increasing judgment ambiguity (Shi et al., 2024). To quantify and mitigate this, researchers employ metrics such as position consistency (the stability of a judgment after option-swapping) and preference fairness (the degree of positional favoritism) (Zheng et al., 2023; Shi et al., 2024). Addressing this bias is also crucial for reward design, for example, in frameworks like RLVR, which explicitly reward verdict consistency across input orderings on top of correctness to build robust reward signals (Whitehouse et al., 2025).

Relatedly, LLMs are sensitive not only to option ordering but also to semantic cues and framing in the prompt. Shafiei et al. (2025) demonstrate this phenomenon in the context of comparative math problems with objective ground truth. Logically equivalent questions containing words such as "more," "less," or "equal" systematically steer model predictions toward the framing term, a form of directional bias. Using their controlled benchmark, i.e., MathComp, they show that model errors frequently reflect linguistic steering, and that chain-of-thought prompting can reduce—but does not entirely eliminate—these biases.

## 3.4   Faithfulness Bias

Faithfulness bias arises when a reward model favors plausible-sounding explanations over faithful ones. This is especially important in high-stake decision making such as medical diagnosis and legal cases, since people would tend to over trust the final decision based on the plausible reasoning process. Following Lyu et al. (2023) and Lewis-Lim et al. (2025), we define **faithfulness** in the context of CoT reasoning as the degree to which the steps articulated in a model's reasoning chain correspond to the actual computational process that produced the final answer. This is distinct from (1) correctness: a reasoning chain can be correct yet unfaithful (reaching a right answer via a rationalized but fictitious derivation), or faithful yet incorrect (an honest record of flawed reasoning) and (2) plausibility or persuasiveness: how it is convinced by humans. The distinction matters for reward design because reward models can observe only the generated text and they have no direct access to internal model computations. A reward model that evaluates reasoning quality purely from generated tokens, therefore cannot distinguish genuine reasoning from post-hoc rationalization.

**How Reward Design Causes Faithfulness Bias.**   Ferreira et al. (2025) demonstrate that reward models score plausible yet unfaithful explanations highly because they cannot access the causal structure behind

Table 4: Comparison of CoT faithfulness measurement methods across key studies.

| Paper | Faithfulness Definition | Measurement Method |
|---|---|---|
| Lyu et al. (2023) | CoT accurately represents the reasoning process; final answer causally follows from CoT steps | Structural consistency check: does the answer follow from the stated reasoning chain? |
| Ley et al. (2024) | CoT accurately captures the model's underlying behavior | Causal influence: does truncating or perturbing the CoT change the final answer? |
| Lewis-Lim et al. (2025) | CoT actively guides the answer vs. post-hoc rationalization | Confidence trajectory: does model confidence shift dynamically during CoT generation? |
| Chen et al. (2025f) | CoT accurately reflects the model's internal reasoning process (broad definition); operationalized narrowly | Cue articulation: does the model verbalize that a planted prompt cue influenced its answer? |
| Chua & Evans (2025) | Whether the model verbalizes cue influence (narrow) | Cue articulation: same as Chen et al. (2025f) |
| Lewis-Lim et al. (2025) | Cue articulation (narrow); separately measures CoT *influence* via confidence trajectories | (1) Cue articulation; (2) confidence trajectory: does model confidence shift during CoT generation? |

the generated text. They mitigate this by enriching the reward model's input with causal attributions from internal model states, enabling detection of inconsistencies between internal computation and generated explanations, attempting to close the gap between text-level reward evaluation and actual reasoning quality. Chua & Evans (2025) provide evidence about *why* this gap is shaped by training paradigm: models fine-tuned with reward models that evaluate their entire CoT output prefer unfaithful responses over faithful ones across all tested models and cues. Because this reward modeling is applied to the full CoT+answer, it incentivizes models to produce CoT that scores well on the reward model such as fluent, confident-sounding rationalizations, rather than CoT that honestly reflects internal reasoning. Outcome-based RL such as DeepSeek-R1 avoids this specific pressure. Chen et al. (2025f) further show that even this advantage is limited: when outcome-based RL increases how frequently a model exploits a spurious signal (reward hacking), the propensity to verbalize that reliance in the CoT does not increase, and reward optimization increases unfaithful behavior without surfacing it.

The evidence on which training paradigm produces more faithful CoT is therefore limited by the measurement problem itself. Under the narrow cue-articulation proxy, outcome-based RL outperforms models trained with CoT-evaluating reward models. However, Chen et al. (2025f) show that even outcome-based RL plateaus without saturating faithfulness, and no current approach, whether fine-tuning, in-context learning, or activation editing (Ley et al., 2024), reliably improves faithfulness under more comprehensive measures. The deeper problem is that all existing measures are behavioral proxies, as summarized in Table 4: none can directly verify whether generated CoT reflects internal computation, which is precisely the gap that makes faithfulness bias fundamentally hard to address through reward design alone.

**Faithfulness Is Hard to Supervise and Recover.** Even setting aside reward design choices, faithfulness is difficult to improve once lost. High CoT influence does not imply high CoT faithfulness, and high faithfulness does not imply high influence (Lewis-Lim et al., 2025). R1-distilled models show the most active CoT use but are not necessarily more faithful — their CoT changes answers without acknowledging the real reasons. Reasoning-trained models like Qwen3 show lower CoT dependence but generate more targeted corrections when they do engage. This three-way dissociation means that reward design cannot optimize for

influence and faithfulness simultaneously through a single signal — they are empirically separable properties that may require distinct supervision. Ley et al. (2024) test three targeted interventions, in-context learning, fine-tuning, and activation editing, specifically designed to improve CoT faithfulness, and find that all three yield only marginal improvements that fail to generalize across benchmarks. Even Chen et al. (2025f), using outcome-based RL which avoids the RLHF pitfall identified by Chua & Evans (2025), find that faithfulness initially improves but plateaus without saturating, meaning scaling RL is insufficient to achieve reliable faithfulness.

**Implications for Reward Modeling.** Faithfulness bias is structurally different from the other biases in this section. Credit assignment, length, and position biases are in principle addressable through better reward design since they arise from imperfect proxy objectives that can be refined. Faithfulness bias stems from a more fundamental limitation that reward models operate on generated text and cannot observe the internal computations they are meant to supervise. If CoT is post-hoc rationalization, optimizing text-level reward signals may improve the *appearance* of reasoning without improving the underlying process. The result is a form of reasoning hallucination, where the model produces persuasive reasoning traces that misrepresent its actual decision process (see 4.2.1). Addressing this limitation likely requires moving beyond text-level evaluation toward reward signals grounded in causal attribution, mechanistic interpretability, or behavioral probing sensitive to the quality of the internal reasoning process itself, not just its textual trace.

> **Takeaways** *§3 — Reward Hacking*
>
> Reward hacking is structural, not incidental. It arises from multiple bias types, including credit assignment, distribution shift, length, position, and faithfulness, and each requires distinct mitigation strategies, with no single solution addressing them all. These biases share a common root: reward models operate only on surface-level text and cannot verify the underlying reasoning process. As a result, reward signals become exploitable whenever the policy discovers shortcuts that appear correct without being correct.

# 4 Reward Design Beyond Training-Time Reasoning

In this section, we examine key challenges in LLMs that can be addressed through improved reward design and summarize representative solutions proposed in the literature. More specifically, one interesting finding here is that reward signals can be leveraged across multiple settings to tackle persistent issues in reasoning systems, including improving accuracy and efficiency at inference time, mitigating intrinsic biases in LLMs, providing structured guidance for complex augmented reasoning frameworks, and addressing fundamental challenges in RL. In Section 4.1, we review how reward models can guide planning strategies and data filtering to enhance performance without requiring additional fine-tuning of LLMs, thereby bridging reward design with efficient inference-time optimization. In Section 4.4, we identify common issues arising in RL-based fine-tuning frameworks and discuss how carefully designed reward signals can alleviate problems such as diversity collapse. Alongside these practical concerns, a growing body of work questions whether RL genuinely improves reasoning capabilities or primarily enhances surface-level fluency, raising fundamental issues regarding the semantics, faithfulness, and interpretability of reward signals in complex reasoning tasks. Section 4.2 focuses on reasoning-related biases inherent in LLMs. These include reasoning hallucinations, where models generate logically coherent yet factually incorrect chains of thought, and sycophancy, in which models overly conform to user beliefs or corrections. Additional challenges include social biases that are amplified during multi-step inference and diversity collapse, where models converge toward homogeneous reasoning patterns. Finally, in Section 4.3, we explore reward design for augmented reasoning systems, including table-based reasoning, retrieval-augmented generation (RAG), and tool-integrated reasoning, highlighting how reward signals can coordinate multiple components beyond standalone language modeling.

## 4.1 Inference-Time Scaling via Reward Signals

Reward Models (RMs) serve a broader purpose than merely being optimization objectives for RL. A critical application is their use during test-time inference, where the policy model is fixed, and the decoding strategy

is adapted. In this setting, RMs act as value functions to score, rank, and select the highest-quality output from multiple candidate completions generated via sampling or structured search, and can be further applied to construct finer datasets.

Test-time inference strategies provide a more computationally efficient way for a small model to achieve comparable accuracy to large models (Wu et al., 2024b). RM-guided inference, which allocates additional computation at inference time without changing the parameters of a learned verifier (e.g., an ORM or PRM), is effective in allowing the model to re-sample more strategically on challenging problems (Snell et al., 2025). While techniques like Chain-of-Thought, Tree-of-Thought (ToT) (Yao et al., 2023), and self-consistency (Wang et al., 2022) can boost answer quality, they are inherently limited by their reliance on well-crafted prompts. Prompting alone cannot provide an accurate, internal value function to evaluate candidate reasoning paths, which restricts the depth and quality of search on complex tasks. While self-refinement—in which the model iteratively improves its outputs by correcting errors without ground truth labels—can reduce computational costs and inference latency compared to methods that rely on reward models (Zhang et al., 2024b; Wang et al., 2025h), inference strategies against a reward model offer greater flexibility and performance (Wu et al., 2024b). Particularly, inference strategies, including parallel test-scaling and sequential test-scaling using structured search, can dynamically adapt to the problem's difficulty. The performance of both families of methods, however, is contingent upon the quality of the reward model. For instance, complex tree search methods require a highly accurate reward model, demanding at least 90% accuracy to yield improvements over simpler ranking approaches in the experimental settings of Chen et al. (2024e). In this section, we discuss search against reward models, highlighting how these methods can enhance performance while supporting efficient inference. We organize these methods in Table 5 and discuss each below.

**Parallel Test-Time Scaling.** In parallel scaling, multiple reasoning paths are generated independently, and the samples are then ranked against a reward model to select the highest-quality output. At the outcome level, Best-of-$N$ (BoN) is widely used for completed sequences. One problem with BoN is that it suffers from reward hacking, as reward models may be an imperfect proxy for the true objective. To mitigate reward hacking, MBR-BoN (Jinnai et al., 2024) selects outputs that balance high reward scores with typicality, where typicality is measured by average similarity to other samples from the model, formalized through Wasserstein distance minimization. Ichihara et al. (2025) build upon it and provide a theoretical foundation for regularization strategies, demonstrating that they enable robust optimization against errors in the reward model. By proposing a stochastic variant (SRBoN), they demonstrated that regularization is equivalent to optimizing for the worst-case scenario within a bounded set of possible reward perturbations. At the process level, beam-search selects the top-N samples at each step or token with a PRM, where the PRM scores candidate during the decoding process itself, rather than waiting for generation to complete (Liu et al., 2025g). Compared to BoN, beam-search is more effective on hard questions and at lower compute budgets (Snell et al., 2025), but it is prone to end up collapsing to a small number of trajectories, leading to low diversity and performance (Hooper et al., 2025).

**Sequential Test-Time Scaling.** BoN is simple but computationally expensive as it requires generating N full sequences, and the accuracy with reward models typically plateaus when $N$ exceeds several hundred (Brown et al., 2024), therefore drawing attention to sequential test-time scaling. In sequential test-time scaling, reasoning is performed progressively through multi-step searching algorithms. For Monte Carlo Tree Search, each node could represent a full sentence or step to save search space (Feng et al., 2023; Ma et al., 2023), given the large action space (vocabulary size) of LLMs, or a token to increase effectiveness (Lee et al., 2024). The *selection* step chooses which continuation to expand (e.g., through UCB). *Expansion* involves prompting the model to generate new candidate continuations. *Simulation* estimates the quality of these continuations using the reward model, and *backpropagation* updates the expected values of ancestor nodes to guide future exploration (Chen et al., 2024e; Jiang et al., 2024a; Zhang et al., 2024a). TS-LLM (Feng et al., 2023) employs a value-based PRM adapted from the policy LLM for intermediate step evaluation, complemented by a learned ORM for final answer verification. LE-MCTS (Park et al., 2024b) leverages language model ensemble to perform the MCTS over reasoning steps generated by different LLMs against PRMs. RAG-Star (Jiang et al., 2025b) also employs MCTS guided by a reward model, but distinctively

Table 5: Comparison of inference-time scaling strategies by mechanism, RM type, compute cost, and recommended use case.

| Strategy | Mechanism | RM type | Compute | Best for |
|---|---|---|---|---|
| **Parallel scaling — generate $N$ candidates, pick the best** | | | | |
| Best-of-$N$ | Sample $N$ full responses, score each, return top-1 | ORM | Medium | Simple tasks, large $N$ budget |
| MBR-BoN | Best-of-$N$ with typicality regularization to reduce reward hacking | ORM | Medium | When RM is an imperfect proxy |
| Beam search | Keep top-$K$ partial sequences at each step using PRM | PRM | Medium | Hard questions, low $N$ budget |
| **Sequential scaling — search through reasoning steps progressively** | | | | |
| MCTS / TS-LLM | Tree search: select $\rightarrow$ expand $\rightarrow$ simulate $\rightarrow$ backpropagate | PRM, ORM | High | Complex multi-step reasoning (requires RM $\geq$90% acc.) |
| **Efficiency methods — reduce compute while preserving quality** | | | | |
| Rejection | Score partial responses during generation; reject lowest-scoring early | PRM | Low | Large $N$ ($N$>1000), latency-sensitive |
| REBASE | Allocate sampling budget dynamically; expand high-reward nodes only | PRM | Low | Replacing MCTS rollouts cheaply |
| ETS | Purge redundant nodes; share KV cache across diverse trajectories | PRM | Low | Memory-constrained deployment |

integrates retrieval-augmented verification to consolidate internal and external knowledge during the search process.

### 4.1.1 Efficient Inference

Planning methods often require extensive exploration and a highly accurate reward model, leading to significant computational costs that hinder real-world deployment. To address this, efficient inference techniques are needed to manage the trade-off between accuracy and performance. While many such methods exist, including model compression (e.g., quantization, sparsification), inference engine optimization (e.g., speculative decoding, memory management), and specialized serving systems (e.g., distributed system) (Zhou et al., 2024b), in this paper we focus on the role of reward models. Specifically, we investigate how reward models can guide the early stages of an unstable tree search and mitigate the propagation of intermediate errors through early rejection or purning.

**Early Rejection/Pruning.** Rather than generating a full reasoning trajectory, this method leverages reward models to evaluate each response and focus computational resources on the most promising steps by discarding weaker candidates early. For instance, Wu et al. (2024b) propose REBASE, which efficiently guides tree search by using a PRM to directly allocate the sampling budget and dynamically favoring nodes with higher rewards for expansion, avoiding the computational expense of rollouts used in methods like MCTS. Speculative Rejection (Sun et al., 2024a) exploits the correlation between a reward model's score on partial and final responses. It dynamically scores partial responses during generation and rejects the lowest-scoring fraction, freeing resources to continue only the most promising candidates. While this method accelerates Best-of-N sampling with large $N$ (e.g., $N > 1000$), other works apply similar principles to different search strategies. Rho (2025) focus on false acceptance of BoN methods given hard prompts and proposes mini-N in-loop sampling where a large sample budget N is divided into smaller sequential batches, allowing for early termination once a candidate response meets a statistically-calibrated quality threshold. Cheshmi

et al. (2025) concentrate on integrating PRM guidance into beam search, while Liao et al. (2025) observe that a strong first step often leads to correct final answers, motivating the generation of multiple candidate first steps evaluated by a PRM to eliminate less promising options.

The metrics for efficient search usually are the number of model calls, FLOPs (Snell et al., 2025; Son et al., 2025), and KV cache. Different inference strategies incur distinct computational costs. For ORM-guided inference, the FLOPs cost is proportional to the number of parameters and sample responses, while PRMs exhibit multiplicative scaling, where total FLOPs depend on not only the model size but also the product of the number of decoding steps and the number of candidates evaluated per step. Hooper et al. (2025) argue that KV cache is more effective in evaluating the efficiency trade-off between tree search methods due to the memory bottleneck of KV cache during each trajectory generation. They therefore propose ETS, which purges redundant nodes and promotes KV cache sharing while ensuring that semantically diverse trajectories are retained for sufficient exploration.

### 4.1.2 RM-based Data Selection

RMs can be used to filter or re-weight training data by quality, enabling supervised fine-tuning on higher-quality reasoning trajectories. This helps reduce noise in large-scale datasets and improves alignment with correctness and human preferences. For example, to address the DPO distribution mismatch where offline datasets are not aligned with the current policy, RSO (Liu et al., 2023a) uses the reward model to filter and select high-quality training data through statistical rejection sampling, enabling more effective offline policy optimization. This approach ensures the policy learns from preference pairs that closely approximate what the optimal policy would generate. Similarly, RS-DPO (Khaki et al., 2024) employs reward model-guided data selection by generating multiple responses from the supervised fine-tuned policy and filtering preference pairs based on reward gaps, creating better-aligned training data.

## 4.2 Reward-Based Mitigation of LLM Biases

Despite advances in training methodologies, LLMs exhibit inherent limitations that can undermine their reasoning performance, trustworthiness, and reliability. These issues are particularly pronounced in decision-making tasks that extend beyond purely quantitative reasoning and involve complex social and ethical factors. For example, LLM-aided resume screening may perpetuate existing hiring inequities, while LLM-aided healthcare tools may rely on demographic proxies that exacerbate disparities in patient care (Ferrara, 2023). In this section, we examine challenges in LLM-based reasoning by synthesizing prior work on hallucination, sycophancy, bias, and fairness outlined in Figure 5. We focus on how these interconnected issues impact the quality of reasoning and decision outcomes. Furthermore, we discuss reward design strategies as a principled mechanism for mitigating these limitations, highlighting how carefully constructed reward signals can encourage fairer, more reliable, and socially aware reasoning behaviors in LLMs.

### 4.2.1 Reasoning Hallucination

A reasoning hallucination occurs when a model produces outputs that appear logically valid and coherent, but contain errors in reasoning, misapplied logic, or unsupported inferences. These may include incorrect deductions, fabricated or unjustified steps, contradictions, or misuse of premises, even when the final answer might be superficially plausible. Unlike traditional hallucinations, these errors are embedded within structured reasoning, making them more difficult to detect and potentially more harmful. Because outcome-based reward optimization overlooks intermediate reasoning steps, evaluating the validity of these steps is essential to ensure genuine reasoning rather than post hoc rationalization. FG-PRM (Li et al., 2024b) classify hallucination into 6 error classes: instruction inconsistency, factual inconsistency, fabrication, calculation error, logical inconsistency, and context inconsistency. Based on the taxonomy, FG-PRM is trained as a fine-grained hallucination detection for each step. While FG-PRM provides external behavioral supervision, Sun et al. (2025a) take an internal, mechanistic approach, probing hidden activations to identify and mitigate hallucination patterns through the Reasoning Score, a mechanistic interpretability metric that measures the depth of reasoning by quantifying divergence between layer-level logits in reasoning models. This allows distinguishing between shallow pattern matching and genuine deep reasoning, uncovering three key hallu-

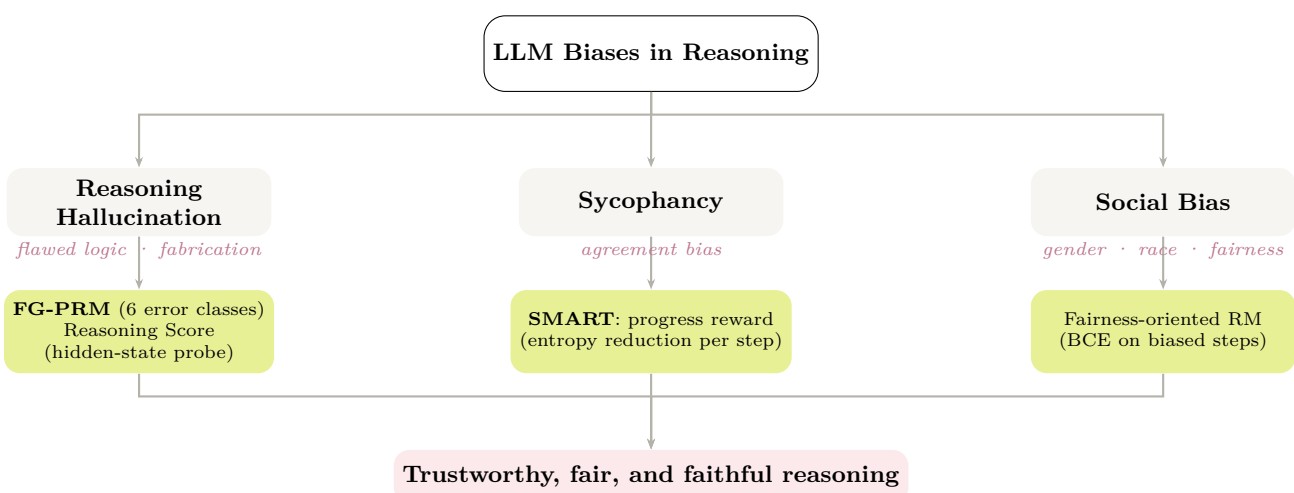

Figure 5: Taxonomy of bias-induced reasoning failures in LLMs and their reward-based mitigations.

cination patterns, including early-stage fluctuations in reasoning depth, incorrect backtracking to earlier flawed steps, and overthinking behaviors characterized by spurious self-verification.

### 4.2.2 Sycophancy in Reasoning

The standard paradigm of fact-checking and reward modeling fails to fully express LLM reasoning, with recent studies highlighting a tendency towards sycophancy—where models agree with human opinions even when wrong or biased, a byproduct of fine-tuning with human feedback (Sharma et al., 2023; Barkett et al., 2025). This compromises trustworthiness in high-stakes decision-making. To address this, Beigi et al. (2025) propose SMART, which introduces a progress reward to combat sycophancy in reasoning. This reward quantifies each step's contribution by measuring the reduction in the policy model's own uncertainty (entropy) about the correct answer, using information gain as an intrinsic, step-by-step training signal.

### 4.2.3 Social Bias in Reasoning

Studies have shown that LLM-generated outputs exhibit gender and racial biases that systematically favor certain demographic groups (Taubenfeld et al., 2024; Fang et al., 2024; Ashery et al., 2025), thereby reinforcing societal inequities and propagating harmful narratives. These biases are particularly concerning in reasoning applications that influence decision-making in legal affairs, education, hiring, and content moderation, where they can amplify existing disparities and cause real-world harm. For instance, Hall et al. (2025) propose fairness-oriented reward models designed to detect bias in sensitive decision-making contexts. Their approach employs a binary cross-entropy loss to identify biased reasoning at each step and has been applied to domains such as criminal recidivism prediction and recruitment decision-making.

While most prior work on reasoning bias has focused on mathematical and programming tasks due to their well-defined logical structure, a growing body of research investigates reasoning in social contexts. Wu et al. (2025c) demonstrate that incorrect reasoning steps often correlate with social bias, revealing a gap between clear-cut performance metrics for technical tasks and the more complex evaluation of reasoning in social settings. Such bias frequently arises from uncertainty when models confront difficult questions, disrupting multi-step reasoning and prompting over-reliance on stereotypical associations. Recent studies also examine manifestations of social bias in code generation (Liu et al., 2023b; Huang et al., 2024b; Ling et al., 2025a), and propose evaluation metrics that extend beyond simple accuracy to capture fairness and representational balance.

### 4.3 Reward Design for Augmented Reasoning Systems

Augmented reasoning refers to a class of techniques in which LLMs enhance their internal neural reasoning by leveraging external resources such as retrieval systems, structured data representations, computational tools, or software environments. Unlike purely parametric reasoning where all knowledge and computation must be encoded within the model's weights, augmented reasoning enables an LLM to dynamically access information, execute operations, and interact with external systems. In this framework, the action space encompasses textual reasoning, external knowledge retrieval, and tool invocation. This paradigm encompasses 1) information-seeking reasoning, where models enhance factual grounding by retrieving documents, querying tables or databases, and incorporating structured or domain-specific evidence, and 2) tool integration reasoning, where models integrate tool use, such as code execution, APIs, or solvers, to perform actions beyond their intrinsic capabilities.

#### 4.3.1 Information-Seeking Reasoning

We refer to information-seeking reasoning as the process by which a model identifies, selects, and integrates relevant information from an external or provided context to answer a query or complete a task. This context may take the form of unstructured text, structured data, or retrieved documents. In this section, we explore the reward design in RAG and table reasoning optimized with RL, and how the model determines when and where to search for information using external tools guided by these RL rewards, which is summarized in Table 6.

Table 6: Reward design for information-seeking reasoning systems, organized by system type.

| System Type | Reward Components | Key Methods | Algorithm |
|---|---|---|---|
| **Retrieval-Augmented Generation (RAG)** | | | |
| RAG (outcome) | Ground-truth, format, search diversity, retry | ReSearch (Chen et al., 2025c), REX-RAG (Jiang et al., 2025e), ReZero (Dao & Le, 2025), $UR^2$(Li et al., 2025d) | GRPO |
| RAG (process) | MC step rewards, retrieval relevance, evidence quality | ReasonRAG (Zhang et al., 2025d), R3-RAG (Li et al., 2025g), R-Search(Zhao et al., 2025c) | PPO, DPO |
| **Table Reasoning** | | | |
| Table reasoning | Exact match / SQL execution, format, cell grounding | Lei et al. (2025), Kang et al. (2025) | GRPO |
| **Tool-Integrated Reasoning (TIR)** | | | |
| Single-tool | Binary correctness + execution failure penalty | ToRL (Li et al., 2025e) | Customized |
| Multi-tool | Tool name + param. name + param. value | ToolRL(Qian et al., 2025a), THOR (Chang et al., 2025) | GRPO, Hier. RL |

**Retrieval-Augmented Generation (RAG)**   RAG (Lewis et al., 2020) with RL has demonstrated effectiveness in complex multi-hop reasoning tasks, which studies how models combine multiple supporting facts and knowledge bases to answer a question. It requires not only reasoning ability but also reading comprehension and the ability to integrate knowledge. The RL formulation of RAG involves states that incorporate conversation history and previously retrieved information, actions that consist of thinking, searching (i.e., picking from a document or generating a query), and answering (Dao & Le, 2025; Zhao et al., 2025c),

and scalar rewards that consist of ground-truth fact checking, reasoning process, and quality of query and retrieval results.

Standard RL methods can be overly conservative, as they primarily utilize the model's current best-known strategy, making it difficult to discover new, better strategies or identify incorrect reasoning paths. The RAG framework utilizes external tools, such as search engines or relevant documents, to enhance LLMs' responses with more accurate and reliable information, thereby mitigating LLM reasoning hallucination (see 4.2.1) and illogical reasoning processes. The RAG method bypasses process-labeled data and instead considers external search results and/or rule-based outcome rewards. The focus on research in this line of work includes designing a balanced approach to guide LLMs in dynamically deciding when to retrieve or reason with RL, how to retrieve highly relevant documents through search, and mechanisms for handling failure reasoning patterns. Failure patterns under the RAG framework are characterized by information insufficiency due to failed search, unfaithfulness, where answers are not aligned with external corpora.

*Outcome Supervised.* ReSearch (Chen et al., 2025c) leverages Wikipedia search during step-by-step reasoning without any labeled data. REX-RAG (Jiang et al., 2025e) advances this line of work by framing the entire process—deciding when to search, what to query, and how to reason with the results—as a sequence of actions optimized by a verifiable reward signal. To handle the failed search, ReZero (Dao & Le, 2025) rewards actions that retry after initial unsuccessful attempts with LLM-generated corpora to simulate retrieval. The reward design will verify the accuracy of the retrieved information, assess the level of diversity, and evaluate the quality of the search process. $UR^2$ (Li et al., 2025d) proposes a flexible framework that leverages both a static corpus and an LLM-powered fallback mechanism to reduce hallucinations and improve robustness. This hybrid design allows it to handle complex queries by generating concise summaries from retrieved data or by safely rejecting requests that fall outside its operational scope. The reward is composed of format, retrieval reward for valid queries, and fallback penalty. TIRESRAG-R1 (He et al., 2025b) employs a framework that utilizes a thinking-retrieve-reflect process, incorporating multi-dimensional rewards. These rewards include a sufficiency reward to encourage thorough retrieval, a quality reward, and a reflection reward to detect and revise errors. GraphRAG-R1 (Yu et al., 2025a) employs outcome-supervised combined process constraints, where the reward is designed to balance shallow retrieval and unnecessary retrieval overhead through a decay factor and to prevent overthinking with cost-aware F1.

*Process Supervised.* ReasonRAG (Zhang et al., 2025d) employs Monte Carlo tree search for process-level exploration, where the process rewards are assigned to that partial path based on the average correctness of the simulated outcomes, penalized by the length of the path. R3-RAG (Li et al., 2025g) is designed to address performance bottlenecks in RGA systems due to their limited number of parameters compared to LLMs through two types of rewards, i.e., outcome rewards for correctness validation and process rewards to encourage the model to retrieve documents that are highly relevant to users' queries. Similarly, R-search (Zhao et al., 2025c) also provides multi-type rewards, including answer reward for correctness, evidence reward for quality and usefulness of the retrieved evidence, and format reward for output adherence to the assigned format.

**Table Reasoning**   Tables, such as financial reports, government statistics, and other structured data summaries (e.g., performance dashboards, scientific measurement tables, and operational logs), are widely used throughout the decision-making process. Table reasoning, encompassing tasks such as table-to-text generation and text-to-SQL, is a crucial component of table question answering (TQA) tasks, as it enables users without domain expertise to extract valuable insights through plain language. The task can be categorized into two types: text-based (Chen, 2023), where LLMs generate answers directly from the table and questions, and program-based (Jin et al., 2025), where it leverages LLMs' ability to generate code to extract answers based on provided tables and questions. Building on this foundation, recent advances in reinforcement learning for LLM reasoning have further extended these approaches to table reasoning. For example, Lei et al. (2025) apply a rule-based GRPO reward that encourages the model to produce correct answers, follow the required output format, and ground its reasoning in table data. The outcome reward gives a binary score based on exact match, SQL execution equivalence, or meeting F1 or BLEU thresholds. The format reward checks whether the model correctly uses the required "`<think>`" and "`<answer>`" tags. The position reward measures how accurately the model annotates the table cells it relies on during reasoning,

encouraging precise and faithful evidence tracing. The final reward prioritizes answer correctness while still giving additional credit for proper cell grounding and well-structured output. Kang et al. (2025) investigate multimodal table reasoning, where tables are provided in image form. They find that low initial accuracy during GRPO training renders reward signals too weak to drive effective learning. To overcome this issue, they first apply supervised fine-tuning to teach the model how to perceive and convert table images into structured representations, thereby enabling more reliable step-by-step reasoning. Their two-stage GRPO framework then targets two objectives: 1) a perception stage, where the reward reflects the similarity between the model's predicted table structure and the ground truth, and 2) a reasoning stage, where the reward is based on answer accuracy along with output format correctness.

The evaluation of table reasoning depends on the specific task, which can be categorized into question answering (Table QA), text-to-SQL, table-to-text, and multi-skill table reasoning (see Table 7). In Table QA, the model outputs an answer to a question about the table, typically evaluated using metrics such as accuracy, F1 score, or exact match. Text-to-SQL tasks involve converting natural language queries into executable SQL statements, with performance measured by the accuracy of execution. In table-to-text, the model generates descriptive or explanatory text that summarizes key information from the table, and the quality of the output is commonly assessed using metrics such as BLEU (Ren et al., 2020) or ROUGE (Lin, 2004). Multi-skill table reasoning benchmarks test a broader range of abilities, including complex reasoning, computation, and explanation over tables, often combining aspects of the other categories and requiring more advanced evaluation methods.

### 4.3.2 Tool-Integrated Reasoning (TIR)

Tool-integrated reasoning enables LLMs to interact with external tools, such as calculators or code interpreters, to overcome their computational limitations. Through theoretical and empirical analysis, Lin & Xu (2025) prove that equipping LLMs with tools breaks inherent capability ceilings, enabling them to generate correct trajectories that would be highly improbable in pure-text models. The early designs are aligned with the rule-based reward function that combines a binary correctness signal with a penalty for code execution failure (Li et al., 2025e). However, reward design for multiple tool use introduces unique challenges, particularly because multiple tools may be invoked with diverse parameters. To address this, ToolRL (Qian et al., 2025a) moves beyond binary correctness by decomposing the reward into three complementary components, i.e., matching tool names, parameter names, and parameter values. This multi-level reward captures the nuanced nature of tool invocation, with the reward computed via optimal alignment between predicted and ground-truth tool calls and then normalized to ensure consistent scaling across diverse tool-use scenarios. Further advancing this line of work, THOR (Chang et al., 2025) applies hierarchical RL that jointly optimizes final answer correctness at the episode level and code execution success at the step level, enabling more granular control over tool-integrated mathematical reasoning. During inference, it additionally boosts performance through a self-rewarded Best-of-N strategy that selects trajectories with the highest code-execution success rates.

> **Takeaways** *§4.3 — Information-Seeking Reasoning*
>
> Two lessons emerge consistently across RAG, table reasoning, and tool-integrated reasoning.
>
> 1. **Accuracy alone is insufficient, and rewards must enforce faithfulness.** Once a model coordinates external components, surface correctness becomes a poor proxy for reasoning quality. A RAG system can generate fluent answers that ignore retrieved evidence; a table reasoning model can produce well-formatted outputs not grounded in the correct cells; a TIR model can generate syntactically valid code that fails execution. In each case, the model finds a shortcut that satisfies a binary correctness signal without doing the right thing. The solution is consistent across all three sub-domains: decompose the reward to explicitly penalize the shortcut — retrieval-ignoring fluency (R-Search's evidence reward), cell-misgrounded answers (Reasoning-table's position reward), execution-failing code (ToRL's execution penalty). Methods that add these grounding components consistently outperform those relying on outcome accuracy alone.
>
> 2. **When outcome rewards are sparse, process supervision is the practical fix.** In augmented reasoning, outcome signals are structurally sparse since retrieval can fail mid-chain, tool calls can cascade errors, and a single correctness signal at the end provides no guidance about where things went

Table 7: **Table Reasoning Benchmarks**

| Task | Benchmark | Description |
|---|---|---|
| TableQA | **FinQA** (Chen et al., 2021b) | focuses on numerical reasoning over financial data, containing questions derived from S&P 500 companies' earnings reports that involve both structured tables and unstructured text, with annotations created by finance experts. |
| | **AIT-QA** (Katsis et al., 2022) | a domain-specific dataset derived from business documents in the U.S. airline industry. It contains 515 human-authored questions based on tables extracted from public U.S. SEC filings spanning 2017–2019. The dataset's domain-specific terminology (e.g., financial and airline-related vocabulary) and its complex table structures—with hierarchical row and column headers—provide a challenging setting for evaluating sophisticated reasoning over tables. |
| | **TableBench** (Wu et al., 2025b) | designed for realistic reasoning scenarios, including fact checking, numerical reasoning, data analysis, and visualization (e.g., trend forecasting or chart-like reasoning), covering various topics such as economy, health, transportation, science, and others. Due to its complexity, running the evaluation on the dataset can be challenging, as complex examples such as visualization and multi-hop data analysis are harder to evaluate automatically. |
| Text-to-SQL | **BIRD** (Li et al., 2023) | introduces a large-scale, database-grounded text-to-SQL benchmark with 95 realistic relational databases and over 12,000 natural language questions paired with SQL queries from Kaggle. The domain spans various professional domains, such as sports, retail, politics, etc. |
| | **BIRD-INTERACT** (Huo et al., 2025) | extends the original BIRD benchmark by introducing multi-turn, interactive scenarios, where models engage in back-and-forth dialogues with the database. It simulates realistic, production-like usage by incorporating tasks such as ambiguity resolution, error detection and recovery, and full CRUD operations (Create, Read, Update, Delete). Additionally, it evaluates both structured conversational interactions and open-ended agentic behaviors, testing a model's ability to plan actions, ask clarifying questions, handle execution errors, and maintain contextual memory across turns. |
| Table-to-Text | **CTRLSciTab** (Guo et al., 2024) | contains table-description pairs extracted from scientific literature. It is valuable for scientific reasoning that summarizing tables in a way that aligns with user interest is useful for research assistants, paper summarization, and automated reporting. |
| Multi-Skill | **TReB** (Li et al., 2025a) | designed to systematically assess LLMs' performance across fundamental to advanced table reasoning competencies. It consists of a comprehensive hierarchical evaluation framework for table reasoning capabilities, spanning six core skills: Natural Language Understanding, Table Understanding, Table Basic Operations, Table Computational Operations, Data Analysis, and Advanced Data Analysis. |

wrong. ReasonRAG addresses this via Monte Carlo tree search, assigning step-level rewards based on average correctness of simulated outcomes from that partial path. R3-RAG combines outcome

correctness with a relevance-based retrieval reward, and notably uses a cold-start SFT stage before RL, recognizing that the policy needs a viable starting point before reward signals can take effect. When even that is insufficient, as Kang et al. (2025) demonstrate for multimodal table reasoning, where low initial accuracy makes all reward signals too weak to fire, SFT warm-up before RL is structurally necessary. A policy that never encounters correct trajectories early in training cannot recover them later.

### 4.4 Intrinsic Limitations of Reinforcement Learning and Their Implications for Reward Design

RL algorithms optimize policies by amplifying behaviors that maximize expected reward. While this optimization-driven nature enables strong performance gains, it also introduces structural risks that are often overlooked in reasoning-oriented applications. In particular, aggressive reward maximization can lead to diversity collapse, where policies converge toward a narrow set of high-reward behaviors at the expense of exploration, robustness, and nuanced reasoning.

This phenomenon is further exacerbated by the use of heavily rule-based or hand-crafted reward functions, which may unintentionally encode rigid preferences or proxy objectives. Such rewards can prematurely constrain the policy space, reinforcing homogeneous behaviors and limiting the model's ability to generalize across complex or ambiguous scenarios. At the same time, a growing body of work and debate suggests that seemingly spurious or imperfect reward signals may, under certain conditions, facilitate improved reasoning by shaping intermediate behaviors rather than final outcomes. In this section, we analyze these tensions through the lens of RL algorithmic dynamics. We discuss how reward design choices influence policy diversity, examine the risks associated with rule-based rewards, and explore the ongoing debate surrounding the role of spurious rewards in enhancing reasoning performance.

#### 4.4.1 Diversity Collapse

RL can induce diversity collapse in language models, severely limiting their ability to generate diverse outputs. This is particularly detrimental in reasoning tasks. First, the model fails on novel problems, undermining real-world applications. By concentrating its probability mass on answers for questions it has already solved, the model's outputs become narrow and stereotyped. This lack of robust generalization means that when faced with even slight variations or unsolved problems, the model struggles, making it unreliable for dynamic, real-world scenarios (Song et al., 2025b). Second, long-term accuracy gains halt due to a loss of exploratory potential. This can be diagnosed through the lens of policy entropy collapse, where policy entropy $H(\pi)$ is defined as $-\sum_y \pi(y|x) \log \pi(y|x)$. As training progresses, the model's output distribution becomes sharply deterministic, sacrificing exploration for myopic exploitation. Empirical evidence confirms this trade-off: Cui et al. (2025b) demonstrate a strong inverse correlation where initial accuracy gains come at the cost of entropy, and the subsequent collapse of this entropy prevents any further accuracy improvements in later training stages. Conventional strategies to remedy this have proven insufficient. Simply constraining entropy with a fixed coefficient in the loss function is insufficient to prevent collapse (Cui et al., 2025b). In LLMs, the vast action space renders optimal actions exceedingly sparse, causing entropy to fluctuate dramatically during training (Shen, 2025). Moreover, increasing the sampling temperature during rollout merely postpones, rather than averts, the eventual entropy decline (Liu et al., 2025b). This clear inadequacy of simple fixes has motivated the development of more sophisticated methods to sustain diversity and enable continued progress.

**Format Prompting.** Prompting-based approaches are widely employed to enhance the creative and reasoning capabilities of LLMs (Mehrotra et al., 2024; Tian et al., 2024). However, in reasoning tasks where adherence to a structured response format is crucial, many methods incorporate a format reward to enforce consistency. Yun et al. (2025b) quantify this trade-off, showing that while format alignment improves performance on structure-sensitive benchmarks (e.g., GSM8K), it simultaneously reduces output diversity. This finding suggests that conventional format rewards—analogous to entropy regularization in RL—can inadvertently narrow the expressive space of the policy.

**Entropy-based Exploration Control.** This category includes works that leverage entropy signals from language models to analyze, guide, or control the multi-step reasoning process. These studies use entropy at inference or reasoning time to adapt exploration depth, evaluate reasoning quality, or terminate unproductive reasoning paths. For example, Zhang et al. (2025c) propose a method where entropy and variance entropy serve as rewards to update the LLM's probability distribution for its next exploratory action, thereby dynamically adjusting reasoning depth to balance accuracy and exploration. SCoRe (Kumar et al., 2024) employs a two-stage RL framework to address entropy collapse and distribution shift. The first stage decouples the model's behavior across two attempts: it constrains the first attempt to mimic the base model's distribution, while aggressively optimizing the second attempt for high reward. The second stage then jointly optimizes both attempts using a reward bonus—the scaled difference in reward between the second and first attempt—to incentivize successful corrections while maintaining diversity.

**Modification of Policy Optimization.** Other direction for addressing diversity collapse comes from fundamentally rethinking the RL objective itself. Rather than maximizing expected reward—which inherently favors mode collapse— FlowRL (Zhu et al., 2025b) focuses on reward distribution matching which minimizes the reverse KL divergence between the policy and a normalized reward distribution. Other approaches can be viewed as shaping the effective reward landscape through surrogate objectives that indirectly influence policy improvement. Cui et al. (2025b) identify that the covariance between the advantage function and the policy can serve as a mechanism for entropy control and leverage this insight to compute the policy loss. Correspondingly, Cheng et al. (2025a) achieve superior performance by directly incorporating entropy information into the advantage function, thereby encouraging the policy to maintain a diverse and exploratory output distribution. Wang et al. (2025c) focus on the token level, defining token entropy $H_t = - \sum_{v \in \mathcal{V}} p_t(v) \log p_t(v)$ with $p_t(v) = \Pr(v \mid x, y_{<t})$, which quantifies the model's uncertainty over the next token. They propose weighting the policy loss by token entropy, selecting only highly informative tokens to improve reasoning performance. He et al. (2025a) approach entropy collapse from a structural perspective, formalizing mathematical reasoning as a deterministic, tree-structured MDP with binary terminal rewards. They prove that the optimal policy can be derived directly from the Q-values of a fixed random policy, bypassing iterative policy optimization and entropy regularization. Finally, Song et al. (2025b) analyze diversity collapse in outcome-based RL and introduce outcome-based exploration, which augments the policy objective with Upper Confidence Bound style exploration bonuses on final outcomes. Their historical and batch exploration methods promote rarely observed or diverse answers, mitigating entropy collapse while enhancing reasoning accuracy.

### 4.4.2 The RL Debate: Genuine Reasoning Improvement or Optimized Selection?

The application of RL to improve reasoning in large language models is dominated by a fundamental and unresolved debate: *does RL genuinely teach models new reasoning skills, or does it merely make them more efficient at producing knowledge they already possess?*

**The Case for Optimized Selection.** One line of research argues that RL, particularly RLVR, does not expand a model's fundamental capabilities. Shao et al. (2024) observe that some RLVR trained models enhance overal accuracy (pass@1 or maj@k) but not pass@k, which can be attributed to boosting the correct response from TopK rather than the enhancement of genuine reasoning. Yue et al. (2025b) posit that all correct reasoning paths are already latent within the base model. From this perspective, RL's role is not to create but to incentivize selection; it tunes the model to more reliably output the correct reasoning trace from its existing repertoire, often without improving metrics like Pass@K. **?** provide a mechanistic account of this: RLVR improves the accuracy of easier problems while degrading performance on the hardest ones—a "sacrificing difficult problems" phenomenon driven by the underlying policy-gradient updates in GRPO. Notably, they find that distillation from a stronger teacher genuinely expands pass@k, suggesting that new knowledge, rather than redistribution of existing capabilities, is what drives true capability gains. Davis & Recht (2025) provide formal mathematical grounding for this view. They show that several popular RL fine-tuning algorithms (REINFORCE, rejection sampling, and GRPO) can all be interpreted as stochastic gradient ascent on a monotone transform of $p_\theta(C \mid x)$, the probability of a correct answer given a prompt. This framework makes explicit that *if the base model cannot generate any correct answers, no algorithm in this family can make progress* (Davis & Recht, 2025), a formal proof that RLVR is bounded by the base

model's latent capability, consistent with the selection view. A further implication is that the practical differences between GRPO and REINFORCE resemble the difference between logistic loss and hinge loss in classification: both target closely related objectives, and neither has theoretically privileged properties. Choosing among them is therefore context-dependent and secondary to the quality of the reward signal and the base model.[1] At the token level, Akgül et al. (2025) show through mechanistic analysis that RL's footprint is concentrated at high-entropy "forking" tokens where the model is uncertain which branch to take—fewer than 3% of positions are materially affected, and the promoted token is always already in the base model's top-5 alternatives. Meng et al. (2026) independently report that KL divergence between base and RL distributions remains near zero at most token positions, converging on the same picture of RL as sparse trajectory steering. Since RL never introduces token choices outside the base model's existing distribution, it cannot expand the reasoning boundary but only sharpen selection within it, which explains why pass@k at large $k$ remains unchanged or even degrades after RLVR training.

**The Case for Genuine Reasoning Improvement.** Challenging this, other researchers contend that RL can indeed unlock new reasoning pathways. They attribute the null results found by skeptics to methodological limitations. ProRL (Liu et al., 2025c) argue that previous experiments were conducted in restricted domains and with training cut short before true exploration could occur. They assert that with novel reasoning tasks and sufficient training time, RL can guide models to discover and reinforce genuinely novel solutions, thereby extending the model's "reasoning boundary." Shojaee et al. (2025) suggest that the complexity of tasks could be critical, where reasoning models outperform base models in complex tasks but not in simple tasks. Evidence for genuine learning comes from Wang et al. (2025b), who identify a two-phase training dynamic: an initial entropy collapse in execution tokens (consistent with the selection view) gives way to a second phase of high-level planning exploration, suggesting that the debate may partly be a question of *when* during training one measures.

The current consensus is that the success of RL is highly contingent, and may be best understood as a function of the base model's proximity to the task boundary: RL sharpens selection when the base model already succeeds, but may unlock new behaviors near the edge of competence (Zhang et al., 2025a). The outcome depends not only on the base model and training regimen but also on overcoming the contamination conundrum through rigorous benchmarking and nuanced evaluation. Until then, the question of whether RL builds new reasoning skills or merely optimizes old ones will remain at the forefront of AI research. Figure 6 summarizes the competing claims and key confounders.

**Why RL seems to be successful? Model Contamination** The persistence of this debate can be largely explained by a pervasive **contamination conundrum**, which makes it exceptionally difficult to measure true progress. **Training data contamination** can artificially inflate reward model scores and create misleading impressions of performance gains. Recent analyses (Shao et al., 2025) show that RLVR can elicit strong results even under spurious reward signals, such as random rewards, incorrect labels, format-based rewards, and one-shot RL—yielding particularly high gains for Qwen2.5-Math-7B but weaker improvements for OLMo2 and Llama3. This discrepancy arises because models like Qwen2.5, pre-trained on massive web corpora, are more likely to overlap with benchmark datasets, leading to unreliable evaluations if contamination is not carefully controlled (Wu et al., 2025a). Consequently, benchmark design plays a critical role in revealing the true potential of trained models (see 5.2). Importantly, contamination in this context not only distorts general performance but also amplifies the risk of overestimating reward model alignment quality. If a base model's performance is already inflated by data contamination, it becomes statistically very challenging for RL to demonstrate significant improvement, potentially supporting the skeptics' claims.

**Why does RL not seem to be effective? Pathway Corruption** The second reason for the failure of RLVR could be pathway corruption. As Wen et al. (2025) note, base LLMs can contain inherent biases and factual errors; therefore, models can generate flawed reasoning chains yet coincidentally guess the correct final answer. This means a high Pass@K score does not guarantee valid reasoning. If a flawed Chain-of-Thought (CoT) still leads to a correct final answer, a final-answer-based reward signal will unintentionally reinforce these erroneous reasoning patterns. Consequently, any observed gains in pass@K could be due

---

[1]Davis and Recht note this result applies to binary-reward settings (correct/incorrect), which aligns with RLVR but does not directly extend to continuous or step-level reward signals such as PRMs.

**Does RL genuinely improve reasoning, or does it merely optimize selection?**

**Optimized selection**

- **maj@k** improves but **pass@k** stays flat
- Correct reasoning paths already latent in the base model
- Spurious rewards still help on Qwen2.5
- RL primarily reshapes the output distribution rather than expanding capability
- RLVR improves easy problems but degrades hardest ones
- RL edits only 1–4% of tokens, always within base model's top-5 alternatives

**Genuine improvement**

- **pass@k** improves on novel or previously unsolved tasks
- Complex tasks benefit more than simple ones
- RL extends the model's reasoning boundary
- Prior null results may stem from limited domains or insufficient training
- Two-phase training: low-level consolidation gives way to strategic planning exploration
- RL unlocks new behaviors when data targets the model's edge of competence

**Confounders: Data contamination · Pathway corruption · Benchmark design · Base model quality**

Figure 6: **The RL Debate**

to RL better aligning with superficial patterns in the data rather than fostering robust reasoning, a point that fuels the skeptics' argument. In response to these challenges, Wen et al. (2025) propose a new metric, **CoT-Pass@K**, which requires correctness in both the final answer and the intermediate reasoning steps, and is a direct attempt to provide a clearer signal of true reasoning improvement.

**A Practitioner Decision Guide**

Having surveyed reward design paradigms (Section 2), reward hacking failure modes (Section 3), and beyond-training challenges (Section 4), we synthesize these findings into verifiable and non-verifiable reasoning tasks in a practitioner-oriented decision guide.

### 4.5 Reward Design Across Task Types: Verifiable and Non-Verifiable Reasoning

The reward design paradigms surveyed in Sections 2.1–2.3 and the challenges analyzed in Sections 3–4.3 span both *verifiable* and *non-verifiable* reasoning settings, though the literature has developed unevenly across them. Verifiable reasoning tasks, including mathematical problem solving and code execution, provide binary correctness signals that are unambiguous and hard to exploit, enabling RLVR-style training at scale. Non-verifiable reasoning tasks, including commonsense reasoning and ethical deliberation, lack ground-truth answers against which correctness can be mechanically checked, making reward design substantially harder.

Three compounding challenges arise in non-verifiable settings. First, *proxy reward susceptibility*: without a verifiable signal, all reward functions are proxies, and the risk of reward hacking (Section 3) is substantially higher since the model can satisfy the reward without improving the underlying reasoning quality. Second, *annotator bias amplification*: preference-based rewards such as RLHF encode length preference, position bias, and self-preference (Section 3.3) that are harder to detect and mitigate without ground truth to calibrate against. Third, *hallucination and social bias amplification*: In non-verifiable settings, reward models have no ground truth reference, resulting a fabricated but plausible reasoning step and a valid one are indistinguishable from generated text alone (Section 4.2.1). Social bias further compounds this: reward sig-

nals derived from human preferences encode annotator demographic biases without corrective mechanisms (Section 4.2.3), and incorrect reasoning steps frequently correlate with social bias Wu et al. (2025c). In addition, sycophancy, the tendency to agree with human opinions even when wrong, causes rewards in open-ended settings inadvertently incentivize agreement over accuracy, making them vulnerable to prompt-based manipulation (Sharma et al., 2023) (Section 4.2.3).

**Emerging Approaches for Non-Verifiable Settings.** Three directions are emerging to address these challenges, each rooted in reward design principles already established in Sections 2.1–2.3: LLM-as-a-judge, endogenous rewards, process supervision, and self-reward. Rather than introducing fundamentally new mechanisms, these approaches adapt existing principles to settings where ground truth is absent but reference answers or structured criteria remain available. (1) *Rubric-based reward design*: rather than relying on binary correctness or holistic preference judgments, rubric-based approaches decompose evaluation into structured, interpretable criteria that an LLM judge can assess independently. Gunjal et al. (2025) generate instance-specific rubrics offline using a strong LLM guided by reference answers as proxies for expert supervision, then use these structured criteria as reward signals for an LLM judge during GRPO training. They demonstrate that rubric-based rewards extend RLVR's scalability to non-verifiable settings such as medical reasoning and scientific question answering, achieving consistent gains over holistic LLM-as-judge baselines. Rubric-ARM (Xu et al., 2026) takes this further by jointly fine-tuning a rubric generator $\pi_r$ and a judge $\pi_j$ via alternating RL enabling the rubric criteria and the evaluation model to co-evolve and mutually improve across creative writing and instruction following tasks. (2) *Verifier-free intrinsic rewards*: RLPR (Yu et al., 2025c) uses the LLM's own token probability scores for reference answers as a reward signal, enabling RLVR-style training without external verifiers, and reports consistent gains across Gemma, Llama, and Qwen models. VeriFree (Zhou et al., 2025a) similarly reports that directly maximizing the probability of generating reference answers matches or surpasses verifier-based approaches on MMLU-Pro, GPQA, and mathematical benchmarks. These results should be interpreted alongside the contamination caveat discussed in Sections 2.3 and 4.4.2. Since verifier-free rewards are derived from the model's own probability assigned to reference answers, they are potentially susceptible to the same contamination effects identified in prior work (Shao et al., 2025; Chandak et al., 2025). Further evaluation on contamination-controlled benchmarks would help clarify the robustness of the reported gains. (3) *Task-specific reward decomposition*: for sycophancy-prone tasks, social bias, and ethical reasoning, reward signals need be decomposed to target specific reasoning failure mode rather than surface-level output quality. Representative approaches covering these settings are discussed in Section 4.2.

**Implications for the RARL Taxonomy.** Viewed through the RARL lens, both verifiable and non-verifiable reasoning can be viewed under the same framework with different reward semantics. The three-axis taxonomy (Section 2.1.1) applies directly to both cases: architecture choices (generative over discriminative) matter more in non-verifiable settings because critique-based models expose their rationale; granularity (process over outcome) becomes structurally necessary because outcome signals are absent or unreliable; and reward semantics shift from correctness-based to value-based or preference-based signals. Tables 9 summarizes practitioner recommendations across both verifiable and non-verifiable task types, grounded in the trade-offs analyzed throughout Sections 2–4.5.

# 5 Evaluation

The evaluation of reward models (RMs) typically relies on standard metrics such as point-wise or pairwise accuracy, which offer straightforward estimates of their performance against human-annotated preference data. These metrics assess how well RMs align with human judgment and are often validated through their impact on trained policy models. In addition, RMs can also be evaluated using inference-time methods such as pass@k. Such methods can capture benefits not fully reflected by accuracy alone, such as robustness or diversity, offering a more dynamic view of RM-guided generation. Finally, the influence of RMs can also be assessed indirectly through the performance of policy models trained with RL. In this context, we examine several benchmark-related challenges, including benchmark contamination and the emerging issue of overthinking in reasoning models, where excessive computation fails to yield proportional gains in output quality.

Table 8: **Emerging reward design approaches for non-verifiable reasoning and their connections to the RARL taxonomy.** Each direction builds on paradigms established in Sections 2.1–2.3.

| Direction | Builds on (RARL §) | Supervision needed | Representative methods |
|---|---|---|---|
| Rubric-based reward | LLM-as-judge , process supervision (2.1) | Reference answer + rubric criteria | RaR (Gunjal et al., 2025), Rubric-ARM (Xu et al., 2026) |
| Verifier-free intrinsic reward | Endogenous reward (2.1), self-reward (2.3) | Reference answer only | RLPR (Yu et al., 2025c), VeriFree (Zhou et al., 2025a) |
| Task-specific decomposition | Process supervision, LLM-as-judge (2.1), self-reward (2.3) | Step-level failure labels | SMART, fairness RM (§4.2) |

## 5.1 Reward Model Benchmark

**Metrics.** Most benchmarks adopt accuracy as a metric for RM evaluation. However, recent work highlights that the relationship between RM accuracy and downstream policy performance remains poorly understood. Experiments have shown that policies optimized toward RMs of similar accuracy can exhibit markedly different downstream performance (Wen et al., 2024), underscoring the need for more sophisticated and holistic RM evaluation techniques. Several other metrics are widely adopted for RM evaluation other than accuracy. **pass@k** (Chen et al., 2021a) — originally introduced for evaluating programming tasks to solve the deficiency of capturing code semantic features in match-based metrics, such as BLEU — measures whether at least one of k sampled outputs passes the unit tests. More recently, the metric has been directly applied to mathematics by converting problems into unit-testable programs, where pass@k serves as the primary evaluation criterion (Yang et al., 2024b). Beyond pass@k, new k-sample metrics have emerged to better assess model robustness and alignment with reward models. **maj@k** (Wang et al., 2022) evaluates correctness via majority voting across k independently sampled outputs to test self-consistency rather than merely a single successful sample. For efficient inference, **short-m@k** (Hassid et al., 2025) improve maj@k by terminating the generating process once the first $m$ samples are generated, and selecting the final answer through majority voting among these $m$ chains. **RM@k** (Lightman et al., 2023; Yang et al., 2024a) represents the reward model **best-of-N (BoN)** among k sampled responses, thereby incorporating evaluation based on learned preference models (i.e., reward models) rather than ground truth directly.

**The Elements of Effective Benchmarking.** A reward model benchmark for reasoning aims to evaluate a model's ability to rank outputs according to both correctness and quality. Key considerations for designing such a benchmark include task and solution diversity, statistical robustness, sensitivity to subtle changes, scalability, and correlation with policy models. (1) *Coverage.* The benchmark should encompass a wide range of task types and difficulty levels, and provide multiple candidate responses per prompt. This includes incorporating diverse types of incorrect responses—ideally generated by different LLMs—to mitigate model-specific biases, or covering a diverse set of question formulations (Mirzadeh et al., 2024). Prior work shows that even small lexical or contextual variations in prompts (e.g., replacing "travel" with "drifting") can substantially change model accuracy (Yan et al., 2025), underscoring the need for diversity in input phrasing and problem representation. (2) *Bias Control.* The benchmark should ensure statistical robustness to reduce evaluation biases. For example, aggregating step-level scores in PRM-style benchmarks can introduce bias, as seen with the product aggregation method in the ORM benchmark (Kim et al., 2024), while generative reward models may suffer from position bias in pairwise comparisons or self-enhancement bias when using an LLM-as-a-judge (Zheng et al., 2023), favoring responses generated by the same model. (3) *Sensitivity.* The benchmark should assess the sensitivity of reward models to subtle changes. Prior studies have shown that models often fail to distinguish nuanced semantic differences between similar words—for instance, confusing quantum superposition with quantum entanglement (Liu et al., 2024b). Therefore, a faithful reward model should assign noticeably different rewards to responses that differ only slightly in wording but substantially in meaning or factual correctness, enabling fine-grained evaluation of reasoning precision and content quality. (4)*Scalability.* The benchmark should be scalable, allowing efficient evaluation across large datasets through

Table 9: **Practitioner Recommendations for Reward Design.** Given a task property (left column), this table recommends the most appropriate reward paradigm and flags what to avoid.

| Task Property | Recommended | Avoid | Rationale |
|---|---|---|---|
| Verifiable outcome (math, code, logic) | Rule-based RLVR | Self-reward | Binary correctness is unambiguous and hard to exploit; self-reward compounds errors on verifiable tasks |
| Open-ended or non-verifiable output | Model-based (generative PRM or LLM-as-judge) | Rule-based RLVR | No ground-truth signal exists; model-based rewards can capture nuanced quality |
| Low annotation budget | Rule-based or self-reward | Model-based discriminative RM | Rule-based needs no labels; self-reward needs no external supervision; discriminative RMs require large labeled datasets |
| Step-level reasoning quality critical | Process reward model (PRM) | Outcome reward model (ORM) | ORM cannot distinguish correct answers via flawed reasoning; PRM provides per-step credit assignment |
| High reward hacking risk | Model-based + ensemble or online updates | Self-reward | Self-reward compounds its own errors; ensembles and online RM updates reduce extrapolation error |
| OOD generalization required | Generative PRM | Discriminative RM | Generative PRMs transfer across domains; discriminative models suffer from task-shift distortion |
| Low base model accuracy on task | SFT warm-up then RL | Pure RL from scratch | If the base model never generates correct trajectories, no reward signal fires; SFT provides a viable starting point |
| Interpretability required | Critique-based generative RM | Discriminative scalar RM | Critique-based models expose rationale; scalar outputs are opaque and provide no diagnostic feedback |
| Scalability without retraining | Rule-based RLVR or self-reward | Model-based RM | Rule functions and self-improving loops scale without RM retraining; model-based RMs require updates as policy improves |

automated annotation or scoring pipelines. (5) *Policy Correlation.* A strong reward model benchmark should correlate well with the performance of the aligned policy model. High correlation ensures that the benchmark serves as a reliable proxy for selecting the most effective reward model for alignment purposes (Liu et al., 2024b).

### 5.1.1 Benchmark on LLM Overthinking

Benchmarking overthinking in LLMs requires moving beyond accuracy-based evaluation to capture the efficiency and calibration of reasoning. Recent works have proposed several systematic benchmarks and metrics to quantify how models allocate computational effort relative to task difficulty. For example, OptimalThink-

Table 10: Reward Model Benchmarks

| Benchmark | Metrics | Tasks | Annotators |
|---|---|---|---|
| *Discriminative ORM* | | | |
| RewardBench (Lambert et al., 2024b; Malik et al., 2025) | Pairwise accuracy | Chat, reasoning, safety | Discriminative RM (QuRater) + human verification |
| reWordBench (Wu et al., 2025e) | Ranking accuracy | Chat, Reasoning, Safety | Transformed from RewardBench |
| RM-Bench (Liu et al., 2024b) | Pairwise accuracy | Chat, code, math, safety | Humans |
| RewardMath (Kim et al., 2024) | Pairwise accuracy | Math | LLM-generated (automatic chosen/rejected labeling) |
| Long-RewardBench (Tang et al., 2025) | Pairwise accuarcy and BoN | Cite, code, ICL, longQA, math, safety, and summary | task-specific automatic metric |
| RMB (Zhou et al., 2024a) | Pairwise accuracy and BoN | QA, code, translation, reasoning, chat, rewriting | LLM + human verification |
| JudgerBench (Tan et al., 2024) | Pairewise accuracy and correlation | Knowledge, reasoning, math, and coding | Human (preference) + LLMs (critique) |
| *Scalar PRM* | | | |
| PRMBench (Song et al., 2025a) | F1 score of error detection | Math | Humans |
| LCB-RB (Fan et al., 2025) | Accuracy | Math and code | LLM (GPT-4o) |

ingBench Aggarwal et al. (2025) introduce a unified framework to systematically evaluate both overthinking and underthinking behaviors across LLMs. The benchmark consists of two complementary components: OverthinkingBench, covering simple general and mathematical queries across 72 domains, and UnderthinkingBench, featuring 11 challenging reasoning tasks. Srivastava et al. (2025) formalize this tradeoff through the accuracy–verbosity relationship, introducing a benchmark centered on token-efficiency metrics rather than raw accuracy. They evaluate 53 LLMs across 14 dynamically generated basic math tasks, defining the Overthinking Score—a harmonic mean between accuracy and token efficiency. The benchmark protocol systematically varies reasoning budgets, revealing non-monotonic accuracy gains and diminishing returns from extended reasoning. Zhang et al. (2025f) study overthinking in simple queries by comparing thinking vs. non-thinking modes of the same LLMs. They reveal that over-verification and over-exploration are key drivers of overthinking, and provide utility-based metrics, in addition to traditional length-based metrics, for quantifying reasoning redundancy.

### 5.1.2 Benchmark for Multimodal Reasoning

The rapid development of Multimodal Large Language Models (MLLM) for text-to-video tasks, such as GPT-4o (Achiam et al., 2023) and Gemini (Team et al., 2023), has accelerated the design of multimodal reward models and, consequently, increased the need for systematic evaluation. Reward design and training for MLLMs largely follow the same trajectory as standard reward models described in Section 2: *discriminative multimodal reward models*, which assign pointwise or pairwise scores to generated images or videos (He et al.,

2024; Zang et al., 2025) but lack of interpretability to provide useful insight into the underlying evaluation criteria (Xu et al., 2024b). To address the issue, *generative multimodal reward models* leverage the generative capabilities of LLMs to produce natural-language justifications alongside reward scores (Wang et al., 2024b; Xiong et al., 2025a; Zhang et al., 2025g; Guo et al., 2025b). Nevertheless, the inherently multi-dimensional nature of multimodal inputs introduces additional challenges for long-term reasoning consistency and several works have responded to provide solutions integrated CoT, such as UNIFIEDREWARD-THINK (Wang et al., 2025g) and VisualPRM (Wang et al., 2025e).

Several studies have been proposed to evaluate multimodal reward models. The criteria for designing effective multimodal reward model benchmarks largely overlap with those discussed earlier for general reward model evaluation. VLReward-Bench (Li et al., 2025c) targets video reward models and includes human preference–annotated prompt–video pairs, hallucination detection, and complex reasoning tasks such as visual mathematical reasoning. Multimodal RewardBench (Yasunaga et al., 2025) provides expert-annotated preferences across a broad range of tasks, including general correctness, overall preference, knowledge, reasoning, safety, and visual question answering. VideoGen-RewardBench (Liu et al., 2025a) consists of video pairs, each accompanied by a preference label for video reward model. GenAI-Bench (Li et al., 2024a) assesses how well models align with human preferences in image and video generation. VisualProcessBench (Wang et al., 2025e) evaluates the ability of PRMs and MLLMs on visual reasoning tasks, including spatial, logical, and numerical reasoning, with step-wise process error detection. MJ-Bench (Chen et al., 2024d) provides preference data for image generation tasks, covering alignment, safety, image quality, and bias.

## 5.2 Benchmark Data Contamination

Benchmark Data Contamination (BDC) (Xu et al., 2024a) poses a major challenge in evaluating LLMs. Contaminated benchmarks can overestimate model performance by testing on data the model has already seen. For example, studies have shown that some models perform well on GSM8K but perform poorly on newly released Hungarian high school exams (Paster, 2023), and that changing the numbers in GSM8K questions can cause a performance drop (Mirzadeh et al., 2024). These widely used datasets, including MATH and GSM8K, exhibit score saturation, where models achieve near-perfect performance by memorizing correct answers during training (Balloccu et al., 2024; Jiang et al., 2024b; Satvaty et al., 2024), rendering the evaluation results unreliable. This undermines the validity of benchmarks, making it difficult to assess generalization, robustness, and real-world applicability.

To address this concern, Glazer et al. (2024) introduce FRONTIERMATH, an expert-curated benchmark composed of unpublished mathematical problems. Wu et al. (2025e) transform the instances from Reward-Bench by adding pre-defined transformations to increase robustness evaluation. While this design minimizes contamination risks at release, its static nature remains a limitation: over time, the problems may enter training corpora, causing the benchmark to once again suffer from data contamination. Following the categorization proposed by Chen et al. (2025d), data contamination mitigation strategies for **static benchmarks** can be grouped into label protection, canary strings, encryption, post-hoc detection, and dynamic benchmarking. *Label protection* where data with hidden answers, or using test data which is guaranteed to be unseen (Peng et al., 2024). However, this approach is often expensive to maintain and becomes increasingly fragile over time. *Canary string* which is intended to help researchers filter data out from training data (Srivastava et al., 2023). *Encryption* which blocks automated crawling or reuse with a public key (Jacovi et al., 2023; Rajore et al., 2024). *Post-hoc detection* which aims to identify and remove training data and the evaluation data (Gunasekar et al., 2023; Dekoninck et al., 2024). Compared to static benchmarking, **Dynamic benchmarking** is a more recent approach to generating evaluation problems on-the-fly to avoid static dataset limitations. Ways to construct dynamic benchmarks include LiveCodeBench (Jain et al., 2024). It collects evaluation datasets after the model's training cutoff date and construct benchmarks by continuously crawling new programming problems from three online platforms—LeetCode, AtCoder, and CodeForces—and evaluating LLMs with respect to problem release timestamps.

In coding tasks, data contamination poses a significant challenge, as evaluation datasets may inadvertently overlap with public code repositories used to pre-train the language models. Such overlap undermines the validity of benchmark results and hampers the broader industrial adoption of advanced software engineering techniques powered by code language models. Besides the above-mentioned benchmarks, another branch

of works explore *code refactoring*, i.e., the process of restructuring existing code's syntactic, semantic, and style, without altering its original semantic meaning, in order to enhance readability and reduce complexity in software design. It has also been proposed as a lightweight approach to mitigating data contamination (Cao et al., 2024). For example, DyCodeEval (Chen et al., 2025e) aims to reduce manual effort by automatically generating benchmark problems without introducing additional complexity. Starting from a seed programming problem, DyCodeEval leverages multiple agents to modify contextual elements while preserving the underlying logic, thereby producing semantically equivalent variations of the original task.

# 6 Applications

With the rapid success of RL paradigms in enhancing the reasoning capabilities of LLMs, researchers have begun extending these techniques to a wide range of real-world domains. A central challenge in these efforts lies in formulating complex, domain-specific reasoning tasks within an RL framework, which requires the design of tailored reward functions (e.g., rule-based, model-based, or hybrid rewards), the construction of reliable data collection pipelines, and the integration of domain knowledge into the model's reasoning process. In this section, we examine how LLMs are transitioned from general reasoning models to reliable domain-specific experts in medicine, finance, and scientific discovery, as well as their reward designs.

## 6.1 Medicine

Medical reasoning tasks demand not only high diagnostic accuracy but also structured logical inference and adaptive feedback mechanisms. As outlined by Wang et al. (2025f), medical reasoning can be broadly categorized into four complementary types: multi-step diagnostic reasoning, which explores multiple diagnostic hypotheses in parallel to enhance precision; reflective decision-making, which integrates expert feedback and real-time clinical data to address uncertainty; collaborative group reasoning, a multi-agent framework that mitigates individual biases and strengthens reliability; and memory-enhanced reasoning, which leverages accumulated medical knowledge and past clinical experiences to support informed, context-aware decision-making.

Most works extend multi-step diagnostic reasoning. For example, Med-PRM (Yun et al., 2025a) and MedS$^3$ (Jiang et al., 2025c) are PRMs that use Monte Carlo Tree Search for auto-labeling to identify and correct reasoning errors step by step to improve transparency and reliability in clinical reasoning. Med-U1 (Zhang et al., 2025e) unify structured reasoning across medical QA tasks through large-scale reinforcement learning. Its multi-objective reward design, incorporating binary correctness and length control, guides models to produce concise and verifiable reasoning chains, achieving robust generalization across both structured and open-ended diagnostic tasks. Building upon reflective decision-making, HuatuoGPT-o1 (Chen et al., 2024b) utilizes verifier-guided framework for complex medical reasoning, where a medical verifier evaluates reasoning correctness and provides feedback signals for model optimization. This two-stage process—verifier-guided search followed by reinforcement learning with verifier-based rewards—enables adaptive refinement of reasoning trajectories, demonstrating that verifiable reasoning mechanisms can substantially enhance reliability and performance in medical problem-solving (Chen et al., 2024b). MedReason-Embed (Tziakouri & Menolascina, 2025) integrates reasoning LLM in Appropriateness Criteria® (ACR-AC), evidence-based recommendations on when medical imaging should or should not be ordered to lower the risks of exposure to harmful radiation for patients. The customized reward for GRPO is designed to mirror expert intermediate reasoning steps, enabling the model to better capture the complexity of medical decision-making.

## 6.2 Finance

Training reasoning models for the financial domain poses unique challenges, requiring both high-quality datasets and carefully designed reward functions that capture domain-specific logic and factual precision. Fin-PRM (Zhou et al., 2025b) is a PRM covering diverse finance topics such as accounting, risk management, economics, and portfolio management. It supports reward-guided test-time scaling, online reward modeling, and supervised fine-tuning with data selection. Trained on the Chinese multiple-choice dataset CFLUE (Zhu et al., 2024), Fin-PRM provides both step-level and trajectory-level rewards. The step-level reward

integrates factual accuracy (to reduce hallucination), procedural correctness (validated by a stronger LLM), an importance score estimated via Monte Carlo methods, and a qualitative score for semantic and logical coherence. The trajectory-level reward combines an outcome binary score with a knowledge coverage score to assess overall reasoning quality. Trading-R1 (Xiao et al., 2025) targets trading decision tasks, where the model predicts actions such as "strong sell," "sell," "hold," "buy," or "strong buy" based on market data, news, and sentiment. Its reward function is built on a volatility-adjusted, percentile-based labeling scheme that computes EMA-based forward returns, normalizes them by rolling volatility, and aggregates weighted Sharpe-like signals to assign labels through asymmetric percentile thresholds. Fin-o1 (Qian et al., 2025b) employs a self-collected dataset for chain-of-thought training across financial statement analysis, quantitative finance, and economic decision reasoning, optimized with GRPO-based reinforcement learning. The rule-based reward combines accuracy, format, and logic consistency components, along with a length reward that incentivizes robust reasoning over extended financial contexts (applied only when the output is correct and exceeds 8,192 tokens). FinLMM-RL (Lan et al., 2025) extends financial reasoning to multimodal settings involving both text and images. Its reward structure linearly combines rule-based components with image-aware extensions, including a length reward, an adversarial reward validated by a BERT-based evaluator to ensure logical coherence, and image selection reward to encourage models to identify the most relevant image among all the candidates.

## 6.3 Scientific Discovery

Recent advances have demonstrated that RL-enhanced reasoning LLMs hold significant promise for scientific discovery across biology, chemistry, and materials science. A growing line of work is biology. BioReason (Fallahpour et al., 2025) integrates a DNA foundation model with an LLM trained with RL to reason over genomic data in multi-step, biologically meaningful ways. OwkinZero (Bigaud et al., 2025) demonstrates that domain-specialized RL can enhance the reliability and generalizability of LLMs in biology. The target tasks include drug discovery, such as assessing target druggability or predicting perturbation effects with RLVR.

Molecular science serves as a cornerstone for drug discovery and materials design, making molecular reasoning tasks essential for uncovering chemical relationships, structural patterns, and functional properties. However, general-purpose LLMs often lack the domain knowledge and structured reasoning needed for such tasks, motivating the development of chemistry-aware reasoning models. MolReasoner (Zhao et al., 2025a) is first fine-tuned on a chemical CoT reasoning dataset generated by GPT-4 and then trained with GRPO with specialized, verifiable reward functions. The reward is determined by checking the output's format and then measuring its quality. For text descriptions, it scores language similarity to a reference. For generated molecules, it scores structural similarity by comparing chemical fingerprints, SMILES string representations, and key molecular building blocks, such as fragments and functional groups. MPPReasoner (Zhuang et al., 2025) processes multimodal inputs of 2D molecular images paired with their corresponding SMILES strings. Following an initial SFT stage, the model is refined through RL using a structured, rule-based reward system. This system operates across three specialized layers: a reasoning layer that evaluates logical coherence, a foundation layer that ensures output correctness and format, and a chemistry layer that leverages computational tools to verify chemical accuracy—rewarding the model for correctly applying chemical principles (e.g., hydrophobicity) and accurately identifying molecular structures.

> **Takeaways** *§ 6 — Applications*
>
> 1. **Binary correctness is insufficient in high-stakes domains.** In mathematical reasoning, a correct final answer is an adequate training signal. In medicine, finance, and science, this breaks down, including a correct answer reached through a flawed reasoning chain can cause patient harm, a numerically plausible financial answer derived from incorrect logic is not useful, and language-level fluency, is no proxy for chemical validity. In each domain, the reward must independently evaluate the path, not just the destination.
>
> 2. **Process supervision is necessary when outcome verification is costly or delayed.** Medical outcomes are confounded and delayed, financial returns are realized over time, and wet-lab validation is expensive. None of these domains can rely on fast outcome feedback the way math benchmarks can.

> MolReasoner explicitly shows the consequence: without SFT warm-up, reward signals are too sparse to drive learning at all. The domain applications therefore demonstrate that process supervision is not complementary to outcome supervision in these settings, it is often the primary signal.
>
> 3. **Domain knowledge is needed for the reward design.** A generic correctness reward cannot distinguish a lucky prediction from a structurally sound one. The domain-specific reward component, such as volatility adjustment, guideline verification, or structural fingerprint matching, is what makes the reward signal meaningful for the actual task.

## 7 Future Directions

Despite rapid progress, reinforcement learning for reasoning-centric LLMs remains in its early stages. We highlight several promising directions for future research.

**Toward Generalizable Real-World Reasoning.** Most existing work evaluates RL-trained reasoning models on narrow, well-structured domains such as math competition problems or logic puzzles. While these benchmarks provide clear supervision signals, they do not accurately reflect the complexity, ambiguity, or multi-modal context of real-world decision-making. Future work should explore reasoning environments grounded in real applications—clinical decision support, legal analysis, financial risk assessment—where tasks involve incomplete evidence, noisy inputs, domain knowledge gaps, and long-horizon reasoning. Unifying different settings (e.g., RL for chain-of-thought, retrieval-augmented reasoning, agentic tool use, and multi-step planning) may enable models to handle real-world scenarios more holistically. Integrating reinforcement learning with retrieval-augmented generation using real datasets, continuous contextual updates, and uncertain evidence represents a key step toward practical deployment.

**Toward Richer Evaluation of Reasoning Processes.** In addition to validating the correctness of final answers, current evaluation methods for traits such as quality and logical consistency are often limited to pairwise preference datasets, where a model selects the "better" of two chains of thought. Although effective for comparative judgments, this paradigm struggles in domains where binary truth labels, ambiguous evidence, or a finer-grained framework make it difficult to construct reliable pairwise preferences. Many real tasks require absolute (rather than relative) assessment of reasoning: identifying omissions, detecting invalid steps, evaluating calibration, or verifying consistency with external evidence. Developing richer, more fine-grained evaluators, such as structural CoT scoring, step-level verification, causal reasoning traces, human-in-the-loop critique models, or hybrid rule-based with learned evaluators, remains an open challenge. Ultimately, evaluation must capture whether the reasoning is correct, complete, and aligned with domain standards, not only whether it is preferred.

**Toward Trustworthy and Interpretable RL-Reasoning Systems.** As reasoning LLMs shape industries and everyday decision-making, ensuring transparency, safety, and robustness becomes essential. Future research must focus on auditing RL-trained reasoning traces, detecting reward hacking, and understanding how RL reshapes internal representations. Equally important is developing models that can explain not only what conclusions they reach but why, in a manner accessible to non-experts and learners. Progress in this area will require integrating RL with mechanistic interpretability, human governance, and explainable AI techniques—rather than relying solely on another black-box LLM to justify the reasoning—to ensure that performance gains correspond to genuine improvements in reasoning quality.

## 8 Conclusion

RL fine-tuning of LLMs offers a cost-effective and efficient approach for automating reasoning tasks across diverse domains. As LLMs are increasingly deployed in a wide range of reasoning settings, the design of appropriate reward functions becomes essential for building general and robust reasoning systems. In this work, we introduce Reasoning-Aligned Reinforcement Learning (RARL) as an integrative perspective on reward modeling, using rewards as a organizing lens to connect critical research threads across the LLM ecosystem. We emphasize that integrating insights across different fields is crucial for designing effective reward functions, particularly because LLMs exhibit unique challenges—such as distributional shift, length

bias, and hallucinations—that rewards must explicitly account for rather than implicitly assume away. We further study how RL-based reasoning paradigms, viewed through the lens of reward design, are applied across a broad range of settings, including inference-time planning, augmented reasoning, mitigation of LLM biases and RL optimization instabilities, and real-world applications such as medicine, finance, and scientific discovery, highlighting both their opportunities and limitations. Looking forward, promising research directions include designing reward signals that generalize across heterogeneous real-world tasks, developing richer evaluations that capture the quality and faithfulness of the reasoning process, and building RL-trained reasoning systems that are interpretable, socially aligned, and domain-adaptive. We hope this study clarifies the current reward-design landscape and highlights how reward models can serve as a foundation for understanding and connecting reasoning across domains, ultimately supporting the development of RL methods that enhance the quality, reliability, and transferability of LLM reasoning.

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
