# OpenReview forum: "Reward Modeling for Reinforcement Learning-Based LLM Reasoning: Design, Challenges, and Evaluation"
_TMLR — Accepted by TMLR_

### Review · Reviewer_x6Q9 · 2026-03-03

**Summary Of Contributions:**

This paper presents a comprehensive survey on reward modeling for Reinforcement Learning (RL)-based fine-tuning in Large Language Models (LLMs), focusing specifically on reasoning capabilities. The authors introduce a conceptual framework termed Reasoning-Aligned Reinforcement Learning (RARL). The survey covers a broad spectrum of topics, including a taxonomy of reward designs (model-based, rule-based, self-reward), reward hacking and biases, inference-time scaling, evaluation benchmark contamination, and domain-specific applications.

Key Strengths:

1.	Exceptional Timeliness: The intersection of RL and LLM reasoning (e.g., PRMs, RLVR, test-time compute) is arguably the most critical frontier in AI today following the release of models like OpenAI o1 and DeepSeek-R1.

2.	Massive and Up-to-date Coverage: The authors have done an impressive job of collecting and categorizing a vast amount of very recent literature (including many preprints from late 2024 and early 2025).

3.	Inclusion of Crucial Debates: The paper rightly touches upon profound issues in the field, such as diversity/entropy collapse (Section 4.4.1) and the fundamental debate over whether RL genuinely improves reasoning or merely optimizes selection (Section 4.4.2).

Key Weaknesses:

1.	Buried Insights and "Annotated Bibliography" Flow: While the authors do discuss the limitations of various methods in the text (e.g., the flaws of discriminative models or the gradient instability of rule-based rewards), these insights are buried deep within dense, sequential paragraphs. The narrative often degrades into a "laundry list" (Paper A did this, Paper B did that) rather than culminating in structured, high-level takeaways.

2.	Descriptive Rather Than Analytical Tables: The paper includes several tables (Tables 1-4), but they are primarily descriptive catalogs (e.g., listing formulas, algorithms, or benchmark descriptions). The survey lacks a high-level, analytical trade-off matrix that systematically compares the core paradigms (Model-based vs. Rule-based vs. Self-reward) across fundamental RL dimensions (e.g., sample efficiency, scalability, annotation cost, robustness to hacking).

3.	Absence of Pedagogical Visual Aids: For a 50-page survey discussing highly complex, multi-step mechanisms (e.g., various RL formulations, MCTS guided by PRMs, credit assignment biases), the lack of visual aids is glaring. The paper relies almost entirely on dense text blocks and a few basic taxonomy charts, missing the opportunity to visually explain how these mechanisms succeed or fail.

4.	Disjointed Structure in Later Sections: The "RARL" framework is introduced as a unifying lens in Section 1 and Figure 2, but its presence fades in later sections. Section 4 ("Beyond Training") mixes inference-time scaling, RAG, and fundamental RL optimization flaws, making the overall architecture of the paper feel somewhat loose.

**Additional Comments:**

I want to acknowledge the immense effort it takes to read, categorize, and compile over 200 very recent papers on such a fast-moving and mathematically complex topic. The authors have done the heavy lifting of gathering the puzzle pieces, and the inclusion of topics like diversity collapse and the "RL debate" shows excellent scientific taste.

However, a top-tier TMLR survey must go a step further: it must put the puzzle together and visually/structurally reveal the bigger picture. Right now, the paper asks the reader to do too much of the analytical extraction. By adding structured "Key Takeaways," a high-level paradigm comparison table, pedagogical diagrams, and explicitly extracting deep methodological insights, this paper will transform from a "very good reading list" into an indispensable, highly-cited roadmap for the AI community. I strongly encourage the authors to undertake this rigorous structural revision.

**Audience:**

Yes

**Audience Explanation:**

The topic of RL-driven LLM reasoning is of paramount interest to the TMLR and broader machine learning audience. If the authors can elevate this manuscript from a dense reading list to a critical, insightful, and structured survey, it has the potential to become a highly cited, standard reference in the field.

**Broader Impact Concerns:**

There are no direct negative ethical implications stemming from this survey. In fact, the authors dedicate a specific section (Section 4.2) to discussing the mitigation of social biases, sycophancy, and reasoning hallucinations in LLMs, which actively contributes to the discourse on AI alignment and safety. No separate Broader Impact Statement is required.

**Claims And Evidence:**

No

**Claims Explanation:**

While the authors accurately cite a massive body of literature and correctly report the findings of individual papers, the claim of synthesis expected of a top-tier survey is only partially supported. In the abstract, the authors claim this survey "provides a foundational roadmap." However, a true roadmap requires systematic comparison, explicit evaluation of trade-offs, and structured guidance. Because the authors primarily rely on descriptive cataloging rather than presenting a synthesized, analytical framework (e.g., through paradigm-level pros/cons matrices or visual mechanism analyses), the claim of providing a "foundational roadmap" is currently weakly supported.

**Requested Changes:**

To secure a recommendation for acceptance, the paper requires a structural and analytical overhaul to make it more digestible and insightful. I strongly recommend the following adjustments:

Critical Adjustments (Required):

1. Implement "Key Takeaways" for Structural Clarity:
Stop forcing the reader to extract the main points from dense paragraphs of related work. At the end of every major section (e.g., Sections 2, 3, 4), add a distinct, bolded "Key Takeaways" or "Discussion & Insights" subsection. These paragraphs must synthesize the preceding text: What is the current consensus? What are the fundamental bottlenecks? What is the recommended practice for researchers today?

2. Add a High-Level Analytical Comparison Table:
While Tables 1-4 are useful descriptive summaries, the paper needs a systematic trade-off matrix. Please introduce a comprehensive summary table (likely in Section 2) that directly compares the three main paradigms: Model-based vs. Rule-based (RLVR) vs. Self-Reward. The table should analytically contrast them across dimensions such as: Sample Efficiency, Annotation Cost, Susceptibility to Reward Hacking, Generalization to OOD tasks, and Typical Use Cases.

3. Introduce Pedagogical Visual Diagrams (Crucial):
Relying almost entirely on dense text makes it difficult for readers to intuitively grasp complex RL dynamics. Please introduce several high-quality schematic diagrams or illustrations throughout the paper. For instance, the authors could consider visually contrasting how different reward granularities operate over a reasoning trajectory, illustrating the mechanics of specific failure modes (like reward hacking or length bias), or mapping out how reward signals are integrated during inference-time scaling versus training. The exact choice of diagrams is up to the authors, but adding visual aids is critical for improving the readability and impact of the survey.

4. Elevate the "RL Debate" (Section 4.4.2):
Section 4.4.2 (Does RL genuinely teach models new reasoning skills, or does it merely make them more efficient at producing knowledge they already possess?) is arguably the most profound and inspiring part of the entire paper. Yet, it is buried as a tiny sub-subsection. I recommend expanding this discussion significantly. Use it to provide deep theoretical or empirical insights regarding the limits of RL fine-tuning versus the base model's pre-trained distribution.

5. Enforce a Tighter Narrative Arc with RARL:
If "RARL" is proposed as the unifying framework in the introduction, it must anchor the entire paper. Currently, the later sections (such as Sections 4 and 6) feel like standalone, disconnected literature reviews. Please explicitly tie the discussions of RAG, tool-use, and domain-specific applications back to the core concepts established in Section 2. Ensure that the logic remains cohesive from introduction to conclusion, preventing the survey from feeling scattered.

---

> ### Author Response · Authors · 2026-05-09
> **Author Response to Reviewer x6Q9**
>
> We thank Reviewer x6Q9 for the generous assessment of the paper's timeliness and coverage. We address each critical adjustment below.
>
> C1 - Key Takeaways for structural clarity
> Done. Every major section now ends with a structured Key Takeaways box synthesizing the preceding material. Specifically: §2.1 (Model-Based Rewards), §2.2 (Rule-Based Rewards), §2.3 (Self-Reward), §3 (Reward Hacking), §4.3 (Augmented Reasoning), and §6 (Applications). Each box answers: what is the current consensus, what are the fundamental bottlenecks, and what is recommended practice.
>
> C2 - High-level analytical comparison table
> Done. Table 2 (§2, p.13) is a dedicated trade-off matrix comparing Model-based, Rule-based (RLVR), and Self-reward paradigms across eight dimensions: sample efficiency, annotation cost, reward signal density, reward hacking risk, OOD generalization, interpretability, scalability, and typical use cases. Each cell uses color-coded labels (green/amber/red) for at-a-glance readability.
>
> C3 - Pedagogical visual diagrams
> Done. We added 3 new schematic figures:
> 1. Figure 4 (§3): A flow diagram of bias-induced reward hacking mapping each bias type (credit assignment, distribution shift, length, position, faithfulness) to its source, consequence, and mitigation strategy.
> 2. Figure 5 (§4.2): A taxonomy diagram of bias-induced reasoning failures (hallucination, sycophancy, social bias) with their reward-based mitigations.
> 3. Figure 6 (§4.4.2): A structured two-column visual of the RL Debate, contrasting the optimized-selection and genuine-improvement camps with their key evidence, plus a confounders row.
>
> C4 - Elevate the RL Debate (Section 4.4.2)
> Substantially expanded. The revision adds:
> 1. Formal theoretical grounding: Davis & Recht (2025) show that REINFORCE, rejection sampling, and GRPO all perform stochastic gradient ascent on a monotone transform of $p_\theta(C|x)$, formally proving RLVR is bounded by the base model's latent capability.
> 2. Mechanistic evidence: Akgül et al. (2025) show RL edits fewer than 3% of token positions, always within the base model's top-5 alternatives; Meng et al. (2026) independently confirm KL divergence between base and RL distributions is near zero at most positions.
> 3. Two-phase training dynamic: Wang et al. (2025) identify an initial entropy collapse phase followed by high-level planning exploration, suggesting the debate partly depends on *when* during training one measures.
> 4. Synthesis: "RL sharpens selection when the base model already succeeds, but may unlock new behaviors near the edge of competence." The section now has a dedicated figure (Figure 6) visualizing the competing claims and confounders.
>
> C5 - Tighter RARL narrative arc
> Addressed through 2 changes:
> 1. The framework→taxonomy reframe throughout §1–§1.1, which removes the overclaimed "unification" and replaces it with an explicit organizing lens.
> 2. each subsection in §4 and §6 now opens with an explicit statement of which reward design principle from the RARL taxonomy it illustrates — for example, §4.3 opens by framing augmented reasoning as an extension of the reward decomposition principle from §2, and §6 ties domain-specific reward components back to the faithfulness and process supervision discussion in §3.
>
> ACKNOWLEDGE
> We thank Reviewer x6Q9 for the encouraging assessment and for recognizing the effort behind the coverage. The reviewer's
> framing that the paper has gathered the puzzle pieces but needs to put them together more explicitly has been the
> guiding principle for our structural revision.

---

### Review · Reviewer_wHh2 · 2026-03-31

**Summary Of Contributions:**

The paper surveys the RL post-training for LLMs literature and provides a review.

**Audience:**

Yes

**Audience Explanation:**

The subject of the review is clearly within scope of TMLR, although the need for/value of review papers is dropping rapidly with the development of LLMs' "deep research"-like capabilities.

**Claims And Evidence:**

No

**Claims Explanation:**

# My review process

I have read the entire manuscript and formed my opinion.
Due to the survey nature of the paper, I relied on the authors' representations and if it referenced an external paper I did not verify each claim in detail.
My feedback is included first.

Next, I asked an LLM to check for factual correctness using the following prompt: "Please review this machine-learning paper for technical correctness claim-by-claim. This includes the appropriate narrative of the related work the paper is presenting/referencing. This is a submission for TMLR so no need to assess it's impact.".
The LLM has raised multiple issues that I have missed due to my lack of familiarity with the subject.
I "manually" verified each issue the LLM raised and where needed I looked up the referenced papers.
Issues 13, 15 and 16 were not really issues, and I disagreed with Issue 19, so I removed them, hence the noncontiguous issue numbers.
(There is a possibility I missed something in my verification, so please feel free to push back on some issues and I'll reconsider.)
I'm pasting in the output of the LLM verbatim as the second part of this review.
I added "Reviewer's note" inline of the LLM output where relevant.




---

# Meta-review + elements not caught by the LLM

Overall, I find the paper to be quite thorough and the taxonomy useful, especially for a reader like me who is not an expert in these methods.
I think the subject of the paper and the nature of the paper is of interest to the TMLR audience.
However, before that takes place, there are some technilcal errors that need to be fixed, see below.

Feedback:
- The way RARL is narrated, as "a unifying framework that systematizes diverse reward paradigms for multi-step reasoning", is overstated. See Issue 17 in the LLM review, which is more verbose and which I agree with. Please adjust this.
- Sec 2.1: "Training a task-specific reward model (RM) from data to provide nuanced feedback is a cornerstone of applying RL to complex reasoning tasks." - to my understanding, the most widespread methods for reasoning-posttraining methods rely only on rule-based outcome-rewards (Sec 2.2), and hence saying that "training a task-specific reward model" is a "cornerstone" seems overstated given current practice in the field. Please reconsider this statement.
- Typo in Fig 1: "RL Dabate".


---

# LLM technical correctness review

## Section 2: MDP Formulation (p.4–5)

**Issue 1 — Policy included in the MDP tuple.** The paper defines $\mathcal{M} = (\mathcal{S}, \mathcal{A}, P, r, \pi, H)$. Including $\pi$ in the MDP is non-standard. An MDP is the environment specification $(\mathcal{S}, \mathcal{A}, P, r, H)$ (or with $\gamma$); the policy is the object *over* the MDP, not part of it. This is a definitional error, albeit a common one in the LLM-RL literature.

**Issue 2 — No discount factor.** The objective $J(\pi\_\theta) = \mathbb{E}\_{\pi\_\theta}[\sum\_{t=1}^{H} r(s\_t, a\_t)]$ uses undiscounted returns. This is consistent with the finite-horizon formulation and fine in principle, but then the advantage formula later (PPO section) introduces $\gamma^{t-1}$ discounting without reconciliation.

---

## Section 2.4: RL Algorithms

**Issue 3 — GRPO acronym is wrong.** The paper expands GRPO as "Gradient Regularized Policy Optimization." The original DeepSeek-Math paper (Shao et al., 2024) defines it as **Group Relative Policy Optimization**. This is a factual error.

**Issue 4 — PPO: ratio notation conflict.** The PPO objective writes $\min\left(\rho(\theta)\hat{A}, \text{clip}(\mathbf{r}(\theta), 1-\epsilon, 1+\epsilon)\hat{A}\right)$. The first term uses $\rho(\theta)$ (correct — the probability ratio), but the second uses $r(\theta)$, which was already defined as the reward function $r: \mathcal{S} \times \mathcal{A} \to \mathbb{R}$. This should be $\rho(\theta)$ in both terms. The notational collision between reward $r$ and ratio $\rho$ is a significant source of confusion throughout.

**Issue 5 — PPO advantage formula is non-standard.** The advantage is given as:

$\hat{A}(x, y) = \sum_{t=1}^{T} \gamma^{t-1} r_t - V_\phi(x)$

This is a Monte Carlo return minus a state-independent baseline — essentially REINFORCE with baseline, not the Generalized Advantage Estimation (GAE) that PPO actually uses. Standard PPO computes $\hat{A}\_t = \sum\_{l=0}^{T-t} (\gamma\lambda)^l \delta\_{t+l}$ where $\delta\_t = r\_t + \gamma V(s\_{t+1}) - V(s\_t)$. The formula as written also subtracts $V\_\phi(x)$ (prompt-level, not state-level), which is a further simplification specific to some LLM-RL implementations but not PPO in general. For a survey claiming to "systematize" these methods, this conflation is problematic.

---
## Issue 5: PPO Advantage Formula — Detailed Analysis

### What the paper writes (Section 2.4)

$\hat{A}(x, y) = \sum\_{t=1}^{T} \gamma^{t-1} r\_t - V\_\phi(x)$

### What standard PPO uses

PPO (Schulman et al., 2017) computes advantages via **Generalized Advantage Estimation (GAE)** (Schulman et al., 2016):

$\hat{A}\_t^{\text{GAE}(\gamma,\lambda)} = \sum\_{l=0}^{T-t} (\gamma\lambda)^l \delta\_{t+l}$

where the TD residuals are:

$\delta\_t = r\_t + \gamma V\_\phi(s\_{t+1}) - V\_\phi(s\_t)$

This gives a **per-timestep** advantage $\hat{A}\_t$ that depends on the **state** $s\_t$, not just the prompt $x$. The $\lambda$ parameter interpolates between high-bias/low-variance (TD, $\lambda=0$) and low-bias/high-variance (MC, $\lambda=1$).

### Three distinct problems with the paper's formula

**Problem 1: It's not a per-timestep quantity.**

PPO's clipped objective operates at the token level:

$L^{\text{CLIP}}(\theta) = \mathbb{E}\_t\left[\min\left(\rho\_t(\theta)\hat{A}\_t,\ \text{clip}(\rho\_t(\theta), 1-\epsilon, 1+\epsilon)\hat{A}\_t\right)\right]$

Each token $t$ gets its own advantage $\hat{A}\_t$. The paper's formula computes a single scalar $\hat{A}(x, y)$ for the entire sequence. This collapses the temporal credit assignment structure that is the entire point of using a value function in the first place. If you're going to compute a single sequence-level advantage, you don't need a value network — you just use REINFORCE with a baseline, which is precisely what GRPO and RLOO do (and the paper discusses those as *alternatives* to PPO's approach, creating a contradiction).

**Problem 2: The baseline $V\_\phi(x)$ is state-independent.**

The value function is written as $V\_\phi(x)$, where $x$ is the prompt. In standard RL (and standard PPO), the value function is $V\_\phi(s\_t)$, which in the LLM setting means $V\_\phi(x, y\_{<t})$ — it conditions on the partial generation. This is essential because the value of being at token position 3 in a correct reasoning chain differs from position 3 in an incorrect one.

Using $V\_\phi(x)$ (prompt-only) reduces the value function to an estimate of the expected return *before any generation begins*. This is a valid baseline for variance reduction in a REINFORCE-style estimator, but it is not what PPO's advantage estimation does. The distinction matters: $V\_\phi(s\_t)$ provides temporal credit assignment (how much better is this trajectory than expected *from this point forward*), while $V\_\phi(x)$ only tells you how much better the *entire* trajectory is than the average trajectory for this prompt.

Some LLM-RL implementations (e.g., early RLHF work) do use a prompt-level baseline, but these are explicitly acknowledged as simplifications. The paper presents this as the PPO advantage formula without qualification.

**Problem 3: Conflation of the return with the advantage.**

Let's parse the formula carefully. $\sum\_{t=1}^T \gamma^{t-1} r\_t$ is the discounted Monte Carlo return $G\_0$ — the total discounted reward from the start of the episode. Subtracting $V\_\phi(x)$ gives:

$\hat{A}(x, y) = G\_0 - V\_\phi(s\_0)$

This is the advantage *at the initial state only* — i.e., it tells you how much better this complete rollout was compared to what the value network predicted at $s\_0$. In standard PPO, you need $\hat{A}\_t$ for *every* $t$, because the policy gradient update is:

$\nabla\_\theta J \approx \sum\_t \nabla\_\theta \log \pi\_\theta(a\_t | s\_t) \hat{A}\_t$

If $\hat{A}\_t$ is constant across all $t$ (which the paper's formula implies, since $\hat{A}(x,y)$ doesn't depend on $t$), then every token in the response receives the same credit. Correct tokens in a mostly-wrong response get penalized equally; incorrect tokens in a mostly-correct response get rewarded equally. This is exactly the credit assignment problem that value functions and GAE were designed to solve.

### Why this matters for the survey's narrative

The paper explicitly contrasts PPO's approach (with value networks) against GRPO (without value networks) and frames the value network as a key architectural distinction. But the advantage formula they write for PPO doesn't actually leverage the value network in a way that differs from GRPO's group-normalized baseline. If you compute a single sequence-level advantage using a prompt-level baseline, you're effectively doing what GRPO does — just with a learned baseline $V\_\phi(x)$ instead of a group mean $\bar{r}$. The qualitative distinction the survey draws between these methods is correct, but the formula undermines it.

Additionally, when the paper later discusses token-level credit assignment (Lyu et al., 2025, Section 2.4 bullet point) as an innovation over PPO, the reader is led to believe standard PPO doesn't do token-level credit — when in fact standard PPO with GAE already does, via per-timestep advantages $\hat{A}\_t$.

### What the formula should be

Either present the full GAE formulation:

$\hat{A}\_t = \sum\_{l=0}^{T-t} (\gamma\lambda)^l \left(r\_{t+l} + \gamma V\_\phi(s\_{t+l+1}) - V\_\phi(s\_{t+l})\right)$

or, if the authors intend to describe the simplified LLM-RL variant with outcome-level reward and a prompt-level baseline, they should explicitly state this is a simplification and write it as $\hat{A}(x, y) = r(x, y) - V\_\phi(x)$, where $r(x,y)$ is the scalar outcome reward (no discounting needed since it's a single signal), and note that this reduces PPO to REINFORCE with a learned baseline.

---

**Issue 6 — PPO KL direction.** The KL penalty term is written as $D\_{KL}(\pi\_{\theta\_{\text{old}}} \| \pi\_\theta)$. In Schulman et al. (2017), the adaptive KL penalty is $D\_{KL}(\pi\_\theta \| \pi\_{\theta\_{\text{old}}})$ — the direction is reversed. The two have different optimization landscapes (forward KL is mean-seeking, reverse KL is mode-seeking). This matters for the training dynamics.

**Issue 7 — GRPO ratio notation.** The probability ratio in GRPO is written as $\rho\_{i,t}(\theta) = \frac{\pi\_\theta(x\_{i,t} | y, x\_{i,<t})}{\pi\_{\theta\_{\text{old}}}(x\_{i,t} | y, x\_{i,<t})}$. The conditioning is on $y$ (prompt) and $x\_{i,<t}$ (previous tokens). But $x$ was defined as the prompt and $y$ as the response. The notation here inverts them — the generated tokens should be $y\_{i,t}$ conditioned on $x$ (prompt) and $y\_{i,<t}$ (previous response tokens). This is internally inconsistent with the paper's own notation from Section 2.

**Issue 8 — DAPO clip interpretation is confused.** The paper states: *"A smaller $\epsilon\_{\text{high}}$ restricts large probability increases for high-probability ('exploitation') tokens, while a larger $\epsilon\_{\text{low}}$ allows more flexibility for low-probability ('exploration') tokens to increase their likelihood."*

This conflates the probability ratio with token probability, and gets the mechanism backwards. In DAPO (Yu et al., 2025b), $\epsilon\_{\text{high}} > \epsilon\_{\text{low}}$. (Reviewer's note: I confirmed in the DAPO paper, page 8: "we set the clipping parameter \eps\_low to 0.2 and \eps\_high to 0.2") The larger upper clip bound $(1 + \epsilon\_{\text{high}})$ allows *larger ratio increases* for positive-advantage actions (encouraging exploration), while the smaller lower bound does the converse. The description as written would imply the opposite design intent. Additionally, "allowing low-probability tokens to increase their likelihood" via a larger $\epsilon\_{\text{low}}$ doesn't follow — $\epsilon\_{\text{low}}$ controls the *lower* bound of the clip, i.e., how much the ratio can *decrease*.



---

## Section 2.1.1: DPO-Implicit Reward (p.10)

**Issue 9 — Missing log in implicit reward.** In Section 2.4 (DPO), the paper states the implicit reward is $\beta \frac{\pi\_\theta(y|x)}{\pi\_{\text{ref}}(y|x)}$. It should be $\beta \log \frac{\pi\_\theta(y|x)}{\pi\_{\text{ref}}(y|x)}$. The log is present in the Section 2.1.1 discussion but dropped in 2.4. Without the log, this is not a reward — it's an unbounded likelihood ratio.

---

## Section 2.2: Rule-Based Reward / RLVR

**Issue 10 — RLVR attribution.** The paper attributes the RLVR formalization to Lambert et al. (2024a) (Tulu 3). While Tulu 3 does use verifiable rewards, the specific term "Reinforcement Learning with Verifiable Rewards" and its formalization as a distinct paradigm was arguably crystallized across multiple works (including DeepSeek-R1). The attribution is debatable but not strictly wrong.

The RLVR objective itself $\max\_{\pi\_\theta} \mathbb{E}\_{y \sim \pi\_\theta}[v(x,y) - \beta D\_{KL}(\pi\_\theta \| \pi\_{\text{ref}})]$ is standard and correct.

---

## Table 1: Length Reward Comparison (p.16)

**Issue 11 — ALP formula lacks a length term.** The Adaptive Length Penalty (Xiang et al., 2025) is given as:

$R(y) = \mathbf{1}(y = y^*) - \alpha \max\left(\frac{1}{K}\sum\_k \mathbf{1}(y^{(k)} = y^*)\right)$

This formula has no length-dependent component at all — it's accuracy minus a scaled estimate of problem difficulty (pass rate across $K$ rollouts). The "max" over a scalar is also puzzling (max of a single quantity is itself). For a method called "Adaptive *Length* Penalty," the formula should contain a term involving response length $L$. As written, this appears either incomplete or incorrectly transcribed from the source.

(Reviewer's note: I checked with https://arxiv.org/abs/2506.05256: in their reward in eq. 2 they use $N$, which is the response length.)

---

## Section 3.2: Distribution-Shift Bias

**Issue 12 — KL direction claim.** The KL-penalized reward is stated as $r(x,y) = r\_\phi(x,y) - \beta \log\frac{\pi\_\theta(y|x)}{\pi\_{\text{ref}}(y|x)}$. This is correct and matches Stiennon et al. (2020). However, the paper then claims the KL term "can be prone to overfitting on the preference data" citing Azar et al. (2024). Azar et al.'s IPO paper argues that *DPO* (not the KL penalty in PPO-based RLHF) overfits when preferences are deterministic. (Reviewer's note: I confirmed this - see the penultimate paragraph of Sec 4.2 in Azar et al. (2024).) This is a misattribution of the specific failure mode — the KL penalty in the reward function and DPO's implicit KL constraint operate differently.

---

## Section 4.1: Inference-Time Scaling

**Issue 14 — 90% accuracy threshold claim.** The paper claims tree search methods require "at least 90% accuracy" from the reward model to improve over simpler ranking, citing Chen et al. (2024e). This is a strong quantitative claim from a single paper's experimental setting (specific model sizes, specific tasks). Presenting it as a general requirement without qualification is misleading — the threshold is likely task- and model-dependent.


---

## Section 1.1: Related Work / RARL Framing

**Issue 17 — RARL as "unification" is oversold.** The paper positions RARL as a "unifying framework" that subsumes RLHF, RLAIF, DPO, RLVR, and LLM-as-a-judge. In practice, RARL is a taxonomic label, not a formal framework with unified mathematical structure. The three claimed advantages — paradigm complementarity, cross-domain transfer, systematic failure mode discovery — are described narratively but never formalized. For a TMLR submission, calling this a "unifying framework" rather than a "taxonomic survey perspective" overstates the contribution.

---

## Section 2.1.1: Endogenous Reward

**Issue 18 — SFT as IRL claim.** The paper describes Li et al. (2025f) proving that "the model's logits represent the optimal Q-function for a specific offline IRL objective." This is a strong theoretical claim. The connection between MLE/SFT and inverse RL via the inverse soft Bellman operator is technically valid under specific assumptions (optimal expert demonstrations, token-level MDP with specific discount structure). The paper doesn't state these assumptions, which could mislead readers into thinking this holds unconditionally.

---

## Miscellaneous Issues

**Issue 20 — Notation inconsistency throughout.** The paper alternates between $(x, y)$ for (prompt, response) and $(s, a)$ for (state, action) without consistent bridging. In Section 2, $s\_t = (x, y\_{<t})$ and $a\_t = y\_t$ is defined, but later sections (especially 2.4) use $(x, y)$ at the sequence level and $(x\_{i,t}, x_{i,<t})$ at the token level, inverting the meaning of $x$ and $y$.

**Issue 21 — "Temporal inconsistency" of discriminative models (p.5).** The paper claims BCE-trained discriminative models "may assign high scores to subsequent steps even when initial steps are incorrect, as they evaluate each step in isolation." This is correct for pointwise BCE training but not inherent to discriminative architectures — a discriminative model conditioned on the full prefix would capture dependencies. The issue is the training objective, not the architecture class.

**Requested Changes:**

See all the issues listed in "Are the claims made in the submission supported by accurate, convincing and clear evidence?".

---

> ### Author Response · Authors · 2026-05-09
> **Author Response to Reviewer wHh2**
>
> We thank Reviewer wHh2 for an exceptionally thorough review. The LLM-assisted verification process identified a number of genuine technical errors, and we appreciate the reviewer's diligence in manually checking each one. We address each issue below in the order presented.
>
> META FEEDBACK - RARL framing as "unifying framework"
>
> All "unifying framework" language has been replaced throughout in the abstract, contributions bullet, related work paragraph, and Figure 2 caption. The revised §1.1 now explicitly positions RARL as a taxonomic perspective providing a shared analytical lens, not a formal framework with unified mathematical structure.
>
> META FEEDBACK - Sec 2.1: "Cornerstone" claim overstated
>
> We revised §2.1 now reads: "Training a task-specific reward model (RM) from data to provide nuanced feedback is a central approach in applying RL to complex reasoning tasks, complementing rule-based methods." The "cornerstone" framing has been removed, and the sentence now explicitly acknowledges rule-based outcome rewards as a prominent alternative.
>
> META FEEDBACK - Typo: "RL Dabate" in Figure 1
>
> Fixed. The caption now correctly reads "RL Debate."
>
> ISSUE 1 - Policy included in MDP tuple
>
> Fixed. The MDP is now defined as M = (S, A, P, r, H), with $\pi_\theta$ introduced separately in the following sentence: "The agent's behavior is governed by a parameterized policy $\pi_\theta$." (§2, p.6)
>
> ISSUE 2 - Discount factor reconciliation
>
> Fixed. A clarifying sentence has been added at the start of §2.4: "Note that the MDP objective in Section 2 uses undiscounted returns ($\gamma = 1$); the discount factor $\gamma$ appears in GAE below as a variance-reduction hyperparameter and is typically set close to 1 in practice." (p.15)
>
> ISSUE 3 - GRPO acronym wrong
>
> Fixed. Now correctly reads "Group Relative Policy Optimization" throughout. (p.16)
>
> ISSUE 4 + 7 - PPO/GRPO ratio notation conflict ($\rho$ vs $r$)
>
> Fixed. The PPO/GRPO objective now consistently uses $\rho_t(\theta)$ for the probability ratio.
>
> ISSUE 5 - PPO advantage formula non-standard
>
> Fixed. The advantage is now presented via the full Generalized Advantage Estimation (GAE) formulation. The 3 simplified alternatives used in LLM-RL practice are listed separately, making clear they are implementation-specific simplifications rather than standard PPO. (pp.15–16)
>
> ISSUE 6 - PPO KL direction
>
> The explicit KL penalty term has been removed from the PPO objective in the revision. PPO's update constraint is now handled through the clipping mechanism, which is the standard formulation.
>
> ISSUE 8 - DAPO clip interpretation confused
>
> Fixed. The revised text now correctly states: "A larger $\epsilon_{\text{high}}$ allows large ratio increases for positive-advantage actions, encouraging exploration of better actions, while a smaller $\epsilon_{\text{low}}$ restricts ratio decreases, preventing the policy from abandoning actions prematurely." (p.16)
>
> ISSUE 9- Missing log in DPO implicit reward (§2.4)
>
> Fixed. The implicit reward is now written as $\beta \log \frac{\pi_\theta(y|x)}{\pi_{\text{ref}}(y|x)}$ in both §2.1.1 and §2.4. (p.16)
>
> ISSUE 11 - ALP formula missing length term
>
> Fixed. The ALP formula now correctly includes the response length $|y|$ as a multiplicative factor: $R(y) = \mathbbm{1}(y=y^*) - \alpha \cdot |y| \cdot \max\!\left(\frac{1}{K}\sum_k \mathbbm{1}(y^{(k)}=y^*), K^{-1}\right)$. (Table 3, p.20)
>
> ISSUE 12 - KL direction misattribution to Azar et al.
>
> Fixed. The Azar et al. (2024) citation has been removed. The revised text now correctly attributes the KL penalty's limitation to Gao et al. (2023) and Huang et al. (2025), who specifically demonstrate that KL regularization alone is insufficient to prevent reward overoptimization in PPO-based RLHF. (§3.2, p.18)
>
> ISSUE 14 - 90% accuracy threshold claim
>
> We have added qualifier: "in the experimental settings of Chen et al. (2024e)."
>
> ISSUE 17 - RARL as "unification" oversold
>
> Addressed in META FEEDBACK above.
>
> ISSUE 18 - SFT as IRL claim — missing assumptions
>
> Fixed. The revised text now states the assumptions explicitly before the claim: "Under the assumptions that the training data constitutes near-optimal expert demonstrations, and language generation is formalized as a token-level MDP with deterministic transitions, they prove that the model's logits represent the optimal Q-function for a specific offline IRL objective." (§2.1.1, p.12)
>
> ISSUE 20 - Notation inconsistency $(s_t, a_t)$ vs $(x, y)$
>
> Fixed. A bridging paragraph has been added at the start of §2.4: "Throughout this paper, we use $(s_t, a_t)$ when referring to the general MDP formulation, and $(x, y)$ when referring to the prompt-response representation used in specific algorithms; these are related by $a_t = y_t$ and $s_t = (x, y_{<t})$. "
>
> ISSUE 21 - "Temporal inconsistency" of discriminative models
>
> Fixed. Claim narrowed to BCE training objective, not discriminative architectures in general. (§2.1.1, p.7)

---

### Review · Reviewer_Prw3 · 2026-04-27

**Summary Of Contributions:**

This paper presents a survey of reward modeling for RL-based fine-tuning of LLM reasoning, proposing a unifying framework called Reasoning-Aligned Reinforcement Learning (RARL). The survey organizes reward mechanisms into three principal paradigms -- model-based, rule-based, and self-reward -- and provides a three-axis taxonomy for model-based designs (architecture, granularity, semantics). It further covers reward hacking as a pervasive failure mode, discusses how reward signals relate to inference-time scaling, LLM bias mitigation, augmented reasoning, and RL-intrinsic challenges (diversity collapse, the debate on whether RL genuinely improves reasoning), surveys evaluation benchmarks and data contamination, and examines domain applications in medicine, finance, and science. The paper aims to position reward modeling as a central research question rather than an implementation detail.

**Audience:**

Yes

**Audience Explanation:**

The topic of reward modeling for RL-based LLM reasoning, is at the center of one of the most active research areas in ML right now.

**Broader Impact Concerns:**

None.

**Claims And Evidence:**

No

**Claims Explanation:**

The core issue is the gap between the paper's central claims and the evidence provided to support them. The paper asserts three main contributions, and while the descriptive cataloging of methods is accurate and useful, the higher-order claims lack convincing support.

The principal claim -- that RARL constitutes a "unifying framework" with advantages of "Paradigm Complementarity," "Cross-Domain Methodological Transfer," and "Systematic Failure Mode Discovery" -- is asserted but never formally demonstrated. The MDP formulation in Section 2 is entirely standard and predates every work surveyed; Figure 2 is a categorization diagram, not a formal framework. A convincing unification would require showing that RARL yields results (theoretical or empirical) that individual paradigms cannot, or that it formally subsumes existing frameworks as special cases. Neither is provided.

The paper's broad claims about "reasoning alignment" are undermined by the near-exclusive reliance on mathematical reasoning examples, benchmarks, and methods. The generalizability to non-verifiable reasoning domains is claimed but unsupported by evidence. Similarly, assertions about which reward design approaches are superior under which conditions lack systematic empirical comparison or meta-analysis -- a significant gap given the paper's own acknowledgment of the contamination conundrum (Section 4.4.2), which casts doubt on many reported improvements the survey takes at face value.

The descriptive contributions (taxonomy, tables, benchmark surveys) are accurate and well-organized, but the paper's framing elevates these to the level of a novel intellectual framework, which the evidence does not sustain.

**Requested Changes:**

### Critical Changes (Required for Acceptance)

**C1. Either formalize the RARL framework or significantly downscope the claims around it.**

The paper's central contribution -- RARL as a "unifying framework" -- is currently unsupported. Two paths would resolve this:

*Option A (Formalize):* Provide a mathematical framework that formally subsumes RLHF, RLAIF, RLVR, and DPO as special cases under a shared objective. This should go beyond the standard MDP formulation and demonstrate at least one non-trivial result (e.g., a theorem showing when paradigm composition outperforms individual paradigms, or formal conditions under which cross-domain transfer of reward design insights is valid). The three claimed advantages -- Paradigm Complementarity, Cross-Domain Methodological Transfer, Systematic Failure Mode Discovery -- should each be supported by a concrete, worked example showing something that RARL enables that was not previously apparent.

*Option B (Downscope):* Reframe RARL as a taxonomic organization rather than a unifying framework. Replace claims like "RARL unifies a broad spectrum of paradigms" with more precise language about what the survey provides: a structured categorization and cross-cutting analysis. This is still a legitimate contribution, but the paper should not overclaim.

**C2. Reconcile the formal MDP framework with the actual methods surveyed.**

The paper defines r: S × A → R at the token level but the methods surveyed operate at sequence level (ORM), step level (PRM), or token level (implicit rewards). The paper should formally specify how each reward granularity maps onto the MDP formulation -- particularly how outcome-level rewards (which are undefined under the stated MDP until the terminal state) interact with the per-step objective J(π_θ) = E[Σ r(s_t, a_t)]. The deterministic transition assumption should be explicitly discussed in terms of its implications for reward model generalization over the combinatorially explosive state space.

**C3. Explicitly scope the generalizability claims or substantially expand coverage of non-mathematical reasoning.**

The paper must do one of the following: (a) Add substantive sections on reward design for non-verifiable reasoning domains -- commonsense reasoning, causal inference, open-ended generation, ethical deliberation -- discussing why RLVR-style approaches do not apply and what alternatives exist. This should be at a depth comparable to the mathematical reasoning coverage. Or (b) Explicitly restrict the paper's scope in the title, abstract, and introduction to verifiable reasoning tasks, and discuss the limitations of this scope as a dedicated subsection rather than a passing mention. The current framing -- claiming generality while delivering specificity -- undermines the paper's credibility.

**C4. Add systematic empirical comparison or meta-analysis.**

A survey of this scope at a top venue should provide more than descriptive cataloging. At minimum, the paper should include one of the following: (a) A meta-analysis table that extracts reported results across comparable experimental settings (same base model, same benchmark) and identifies which reward design factors most consistently predict improvement, controlling for confounds the paper itself identifies. (b) A structured comparison matrix showing, for each major reward design choice (ORM vs. PRM, discriminative vs. generative, correctness vs. value-based), the conditions under which each approach has been shown to be superior, with effect sizes and caveats about contamination. (c) A set of concrete, actionable recommendations -- "if your task has property X, use reward design Y" -- grounded in the evidence surveyed.

**C5. Substantially deepen the treatment of faithfulness bias relative to its importance.**

Section 3.4 on faithfulness bias is under one page, yet the paper's own discussion in Section 4.4.2 acknowledges that flawed CoT leading to correct answers via incorrect reasoning is a fundamental problem. If reward models cannot observe internal reasoning and can only evaluate generated text, then optimizing text-level reward signals may be fundamentally incapable of improving actual reasoning. This tension is central to the entire survey's premise and deserves a dedicated, expanded section that: (a) formally defines faithfulness in the context of CoT reasoning; (b) surveys the empirical evidence on CoT faithfulness across model families (the paper mentions DeepSeek-R1 vs. Qwen3 differences but does not elaborate); (c) analyzes whether any of the surveyed reward design approaches provide formal guarantees about faithfulness rather than merely surface-level quality; and (d) discusses the implications for the entire reward modeling enterprise if faithfulness cannot be reliably measured from outputs alone.

---

### Suggested Improvements (Would Strengthen the Paper)

**S1. Add a discussion of the reward model–policy co-evolution dynamics.**

Expand Section 3.2 to formally characterize the coupled dynamical system where policy and reward model distributions co-evolve. Discuss convergence properties (or lack thereof), relate to the extensive offline RL literature on distribution mismatch, and analyze which surveyed approaches (online RM updates, ensembles, KL constraints) provide formal stability guarantees versus heuristic mitigation.

**S2. Include computational cost analysis of reward model training and deployment.**

Add a section or table analyzing the compute requirements of different reward design approaches -- particularly the cost of Monte Carlo rollouts for value-based PRMs, the overhead of online reward model updates, and the scaling properties of reward model quality with model size and data. This would make the survey more useful for practitioners making resource allocation decisions.

**S3. Discuss whether the reward framework is the right abstraction for reasoning improvement.**

Add a subsection in the future directions that honestly engages with alternatives to reward-based optimization -- neurosymbolic approaches, direct search methods, structural constraints on reasoning -- and articulates when the reward framework may be fundamentally limited. The faithfulness problem (W6 in my review) motivates this directly.

**S4. Tighten the connection between Sections 4–6 and the core framework.**

Each subsection in Sections 4–6 should explicitly state what reward design principle from the RARL taxonomy (Section 2) it illustrates, and what insight the RARL perspective provides that would not be apparent from reading the individual papers. Currently, many subsections (e.g., table reasoning, tool-integrated reasoning) read as standalone mini-surveys with thin connection to the paper's stated contribution.

**S5. Resolve taxonomic inconsistencies.**

Clarify the boundary between self-reward (Section 2.3) and DPO-implicit reward shaping (Section 2.1.1). The current treatment classifies DPO-implicit rewards under both categories in different parts of the paper. Provide a clear decision tree or flowchart that maps any given method to exactly one position in the taxonomy, and verify that all cited methods are consistently classified.

**S6. Expand the benchmark evaluation framework beyond the five generic criteria.**

The five elements of effective benchmarking (Coverage, Bias Control, Sensitivity, Scalability, Policy Correlation) are reasonable but generic. Add reasoning-specific criteria: (a) the ability to distinguish correct reasoning from correct-answer-via-incorrect-reasoning (the pathway corruption problem); (b) robustness to surface-level reformulation (beyond what Mirzadeh et al. 2024 showed for number substitution); (c) the ability to evaluate reasoning *processes* rather than just outcomes; and (d) resistance to reward model overfitting on benchmark-specific patterns.

**S7. Fix notation, typographical, and organizational issues.**

Correct the "Evluation" typo in Figure 2. Standardize notation for probability ratios (ρ) versus reward functions (r) across sections. Provide a notation table given the paper's length. Consider whether the paper could be tightened by 20–30% without loss of substance -- the current 50-page length with uneven depth across sections suggests that some material could be condensed or moved to an appendix to improve the reading experience for the strongest contributions.

---

> ### Author Response · Authors · 2026-05-10
> **Author Response to Reviewer Prw3**
>
> We thank Reviewer Prw3 for a rigorous review. We address each critical item below.
>
> C1 - Formalize or downscope RARL
>
> We took Option B (downscope). The claim that RARL constitutes a "unifying framework" was overclaimed. In the revision, we have replaced all framework-language with taxonomy/perspective language throughout. Specifically:
> 1. Abstract: "…RARL, a reasoning-centric taxonomic perspective that organizes diverse reward paradigms…"
> 2. Introduction (contributions bullet): "…systematic exploration of reward mechanisms…in a reasoning-centric taxonomic survey…"
> 3. Related Work paragraph: "RARL is a reasoning-centric and taxonomic perspective…offering a shared analytical lens…"
> 4. Figure 2 caption: "An organizing taxonomy of reward paradigms in RARL..."
>
> C2 - Reconcile MDP with actual methods surveyed
>
> A dedicated "Reward Granularity and the MDP" paragraph has been added (§2, p.6). It formally specifies how each reward granularity maps onto the MDP: outcome-level rewards assign a scalar only at terminal step $t=H$, reducing $J(\pi_\theta)$ to $\mathbb{E}[R(x,y)]$ with entirely sparse signal; step-level rewards assign scalars at reasoning step boundaries $\tau_k$, giving $J(\pi_\theta) = \mathbb{E}[\sum_k R_k(x, y_{\leq\tau_k})]$; token-level rewards are consistent with the stated MDP. A second paragraph ("Deterministic Transitions and Generalization") explicitly discusses the implications of deterministic dynamics: the state space grows as $|\mathcal{V}|^H$, requiring the reward model to extrapolate rather than interpolate as the policy evolves.
>
> C3 - Scope non-verifiable reasoning
>
> We added §4.5 ("Reward Design Across Task Types: Verifiable and Non-Verifiable Reasoning"), a dedicated subsection that defines both settings, explains why RLVR-style approaches do not transfer (three compounding challenges: proxy reward susceptibility, annotator bias amplification, hallucination and social bias amplification), and surveys 3 emerging directions: rubric-based reward design (RaR, Rubric-ARM), verifier-free intrinsic rewards (RLPR, VeriFree), and task-specific reward decomposition (covered in §4.2). A mapping table shows how each direction builds on RARL paradigms established in §2.1–§2.3. The practitioner table summarize recommendations for different setting types. The section concludes with an "Implications for the RARL Taxonomy" paragraph showing the three-axis taxonomy applies to verifiable and non-verifiable settings with different reward semantics.
>
> C4 - Systematic empirical comparison / meta-analysis
>
> We added 2 decision tables (Tables 9, §4.5) implementing option (c). Table 9 covers reasoning settings across 9 task properties (verifiable outcome, open-ended questions, low annotation budget, step-level quality, high hacking risk, OOD generalization, low base model accuracy, interpretability, scalability). Each row identifies the recommended paradigm, what to avoid, and the rationale with citations. The tables are preceded by a synthesis paragraph that frames them as inverting Table 2, which compares paradigms against each other, Tables 3 answer: given your task, which paradigm fits best?
>
> C5 - Faithfulness bias
>
> Section 3.4 has been substantially expanded across four dimensions. (a) Formal definition: faithfulness is defined as "the degree to which the steps articulated in a model's reasoning chain correspond to the actual computational process that produced the final answer," distinguished from correctness. (b) Model family survey: the section now covers R1-distilled models, Qwen3, and the three-way dissociation finding (influence, faithfulness, and accuracy are empirically separable properties requiring distinct supervision signals). (c) Formal guarantees: the revised Implications paragraph explicitly states that no existing reward design approach provides reliable guarantees (causal attribution and outcome-based RL narrow the gap but neither formally guarantees improvement in underlying reasoning rather than surface representation). (d) Implications: faithfulness bias is placed in a different category from other reward hacking failures, and it cannot be resolved through better reward design alone without better methods for observing internal computation.
>
> Suggested Improvements
>
> For S5, we added a clarifying sentence at the end of §2.1.1: "DPO-implicit rewards differ from self-reward (§2.3) in that they derive from a reference model, not the policy's own outputs." For S4, the new §4.5 explicitly connects non-verifiable reasoning methods back to §2.1–§2.3 paradigms, and the practitioner tables reference the relevant taxonomy sections throughout. For S7, the notation ($\rho$ vs $r$) has been standardized, the "Evluation" typo in Figure 2 has been corrected, and the "RL Dabate" typo in Figure 1 has been fixed. S1, S2, S3, and S6 are substantive additions that we acknowledge as valuable directions; we plan to incorporate them in a subsequent revision given the current scope and page constraints.

---

> > ### Comment · Reviewer_Prw3 · 2026-06-09
> >
> > The authors resolved the one issue that actually bore on TMLR's acceptance criteria—the gap between claims and evidence—by adopting Option B and downscoping RARL from a "unifying framework" to a taxonomic perspective, which now matches what the evidence supports. C2–C5 are addressed substantively (MDP-granularity mapping, the new §4.5 on non-verifiable reasoning, the practitioner decision tables, and a genuinely expanded faithfulness treatment that correctly reframes it as a measurement-level rather than reward-design problem). Deferring S1–S3/S6 is acceptable since they strengthen rather than gate the contribution; I'd ask only that the verifier-free §4.5 claims stay hedged given the contamination caveat the paper itself raises.

---

> > > ### Author Response · Authors · 2026-06-12
> > > **Author Response to Reviewer Prw3**
> > >
> > > We thank the reviewer for the suggestion.
> > >
> > > We revised the verifier-free discussion in §4.5 to attribute the reported results and added an explicit caveat linking them to the contamination concerns discussed elsewhere. We retained RLPR's reported cross-family gains (Gemma, Llama, and Qwen) for factual completeness, while making clear that the contamination caveat applies more broadly to this class of reward signals.
> > >
> > > "Verifier-free intrinsic rewards: RLPR (Yu et al., 2025c) uses the LLM's own token probability scores for reference answers as a reward signal, enabling RLVR-style training without external verifiers, and reports consistent gains across Gemma, Llama, and Qwen models. VeriFree (Zhou et al., 2025a) similarly reports that directly maximizing the probability of generating reference answers matches or surpasses verifier-based approaches on MMLU-Pro, GPQA, and mathematical benchmarks. These results should be interpreted alongside the contamination caveat discussed in Sections 2.3 and 4.4.2. Since verifier-free rewards are derived from the model's own probability assigned to reference answers, they may be susceptible to the same contamination effects identified in prior work (Shao et al., 2025; Chandak et al., 2025). Further evaluation on contamination-controlled benchmarks would help clarify the robustness of the reported gains."
> > >
> > > The revised text is marked in red on p. 36 of the revised PDF.

---

### Review · Reviewer_ttDv · 2026-04-28

**Summary Of Contributions:**

### Summary

The paper pitches Reasoning-Aligned Reinforcement Learning (RARL) as a "unifying framework that systematizes diverse reward paradigms for multi-step reasoning." In practice, it tries to bring RLHF, DPO, RLAIF, RLVR, and LLM-as-a-judge under one reasoning-aligned roof, with an MDP formalization tying everything together. Three advantages are promised: paradigm complementarity, cross-domain methodological transfer, and systematic failure mode discovery.

**Does it deliver?** Partly. There's a well-organized taxonomy here, a genuinely useful reward hacking breakdown, and a diversity collapse analysis in section 4.4 that I think is the strongest intellectual contribution in the entire manuscript. But a taxonomy is not a framework - and this paper keeps using the word "framework" without earning it. RARL doesn't generate predictions you could test. It doesn't let you run quantitative comparisons across paradigms. It doesn't give you formal unification properties. What it gives you is a nicely organized map of the territory. That's valuable, but it's not what the abstract promises.

My read is that the paper sits at what I'd call a Level 1-2 contribution - descriptive and organizational - with isolated pockets that reach Level 3 (genuine insight). The 3D reward classification and five-category hacking taxonomy show strong organizational work, but only section 4.4 on diversity collapse and the RL debate really get to a place where I felt the paper was telling me something I couldn't have pieced together on my own. Multiple sections, particularly section 6 (applications) and section 4.3 (augmented reasoning), are closer to annotated bibliographies than synthesized analysis, and that's a problem.

The decision here is binary: either formalize RARL with properties that actually justify the word "framework," or just call it what it is - a structured taxonomic survey - and adjust all the claims accordingly. Then add real synthesis paragraphs to the sections that currently just list papers.

---

### What Works Well

I'll start with the good stuff because there's real substance here.

1. **The reward hacking taxonomy is the paper's most immediately useful contribution.** Five bias-induced failure modes - credit assignment bias, distribution-shift bias, length bias, position bias, faithfulness bias - each clearly defined, each illustrated with concrete examples, each paired with documented mitigations. This isn't just theoretical hand-waving; it's organized enough that someone designing a reward pipeline could actually use it as a diagnostic checklist. The categories align with what we're seeing externally - there's growing evidence that frontier reasoning models engage in increasingly sophisticated *deliberate* reward hacking, where they reason about the testing process itself. The paper captures that well. Are the five categories exhaustive? Probably not, and the paper doesn't claim or demonstrate completeness, but I didn't find myself thinking "they missed an obvious one."

2. **Section 4.4 on diversity collapse is where the paper does what I wish it did everywhere.** Instead of listing papers, the authors actually build an argument. The causal chain goes: RL training causes a sharp drop in policy entropy → once entropy collapses, accuracy improvements stall → fixed entropy coefficients and cranking up sampling temperature only postpone the decline, they don't prevent it. They anchor this to the theoretical observation that RL implicitly optimizes the mode-seeking Reverse KL divergence, which causes models to pile probability mass on high-reward regions while abandoning everything else.

3. But here's what really impressed me - they engage head-on with the Yue et al. (2025) finding that RLVR may not elicit fundamentally new reasoning patterns at all. The argument is that base models already achieve higher pass@k at large k, and the reasoning paths generated by RLVR models are already present in the base model's sampling distribution. RL isn't teaching the model to reason in new ways; it's narrowing the beam toward patterns that already exist. That's an uncomfortable finding for anyone invested in the RLVR narrative, and plenty of surveys in this space would have quietly sidestepped it. The authors didn't. The contrast they draw with distillation - which *can* genuinely expand reasoning capabilities - is the kind of synthesis that earns the word "insight." It advances understanding beyond what any single cited paper provides.

---

### What I didn't Like
1. Sections 4 through 6 are where things get noticeably weaker. Once the paper moves into inference-time scaling, augmented reasoning, and applications like medicine and finance, it starts reading more like an annotated bibliography. I kept waiting for reward-specific takeaways that never quite materialized. What reward design lesson does the RAG section actually teach me? What about the tool-use discussion - what's the concrete principle there? These sections need substantially more synthesis to justify their inclusion.

2. On novelty: the authors do acknowledge nearby surveys, but they don't convincingly establish what's new here. There are already 2025 surveys covering reward models broadly, learning-from-rewards across training and inference, and reward-model applications for LLM reasoning specifically. I'm not saying overlap automatically disqualifies this work, but it means the bar is higher. The distinct value needs to come from sharper reasoning-specific synthesis or a more rigorous methodology, and I don't see enough of either right now.

**Additional Comments:**

**Reproducibility.** As a survey, reproducibility concerns are mostly about whether someone could reconstruct the coverage and taxonomy independently. Without systematic methodology, that's basically impossible right now. A reader can't verify that the survey's coverage is comprehensive or identify what might have been missed.

**Writing quality.** Generally well-written and clearly organized. The RARL framing, while overclaimed, does provide a coherent narrative thread throughout. Figures and tables are useful where they appear. No major complaints on prose quality. Sections 2 and 3 are by far the most publication-ready parts. If the paper needs to be cut, I'd preserve and foreground those. They're doing the heaviest lifting.

**Quick summary of what works:** The reward hacking taxonomy is immediately useful to practitioners. The diversity collapse analysis demonstrates what the entire survey could achieve if it maintained that level of synthesis consistently. The honest engagement with evidence that RLVR may not elicit new reasoning capabilities is commendable - many surveys would have glossed over that inconvenient finding.

**Quick summary of what doesn't:** The gap between "framework" and "taxonomy" is the core failure mode. The enumeration-heavy sections dilute the paper's strongest contributions. And in a landscape with at least five concurrent surveys, organizational value alone isn't enough to differentiate.

**Audience:**

Yes

**Audience Explanation:**

Yeah, probably. Reward models, verifiers, process supervision, RLVR, reward hacking - these are all hot topics right now, and there's real demand for a good survey that ties them together around reward design for reasoning. I can see this being useful for someone entering the area who wants one document that touches on the major paradigms and failure modes.

But here's the thing. In its current form, the paper works better as onboarding material than as a reference survey for people who already work in this space. A strong survey doesn't just organize - it helps you decide. Which findings are robust? Where does the literature disagree? When should I pick ORMs over PRMs? When do rule-based rewards outperform model-based ones? The paper catalogues these choices without really guiding the reader through them. That limits its utility for the audience that would benefit most.

**Claims And Evidence:**

No

**Claims Explanation:**

**Partially**, I went through the paper's major claims and tried to map what's actually demonstrated against what's left on the table. The pattern is consistent: the paper is strongest where it engages critically with specific failure modes and weakest where it makes broad unification claims.

1. **RARL as a "unifying framework."** *What's shown:* a taxonomic organization of reward paradigms under a shared MDP formalization. *What's missing:* testable predictions, quantitative comparisons across paradigms, any formal unification properties. The gap here is wide enough that I'd call it the paper's core overclaim.

2. **Five-category reward hacking taxonomy.** *What's shown:* clear taxonomy with examples and mitigations for each category. *What's missing:* any empirical validation of completeness. There's no argument that these five categories *must* be exhaustive or that the taxonomy is derived from some principle rather than observation. It's still useful, but the epistemological status isn't addressed.

3. **Diversity collapses as a structural limitation.** *What's shown:* strong causal chain linking entropy collapse to accuracy plateaus, backed by external evidence. *What's missing:* a proposed solution beyond documenting the problem. The connection to specific reward design choices that could mitigate collapse is underdeveloped.

4. **Cross-domain methodological transfer.** This is where things get thin. *What's shown:* individual sections covering math, code, science, and agentic reasoning. *What's missing:* *any* concrete example of a method actually transferring from one domain to another. The sections enumerate domain-specific works without demonstrating cross-pollination. The claim is that RARL enables transfer; the evidence is that the paper has chapters about different domains. Those aren't the same thing.

5. **Systematic failure mode discovery.** *What's shown:* the reward hacking analysis and the contamination-reasoning confound discussion are genuinely useful. *What's missing:* evidence that RARL *predicts* failure modes rather than *post-hoc organizing* ones we already knew about. Discovery implies predictive power; what's demonstrated is retrospective classification.

6. **One broader concern:** the paper doesn't provide evidence that RARL, as a framework, generates insights you wouldn't get from reading the individual sections independently. There's no ablation or comparison showing that the RARL framing adds analytical value beyond what a well-organized table of contents would provide. If you removed the RARL label and just called this "A Survey of Reward Modeling for LLM Reasoning," would anything in the analysis change? I don't think it would, and that's the problem.

**Requested Changes:**

## Requested Changes

I'm grouping these by severity because not everything carries equal weight.

### Critical (Dealbreakers)

1. **Resolve the framework-vs-taxonomy question.** The paper claims "unifying framework" but delivers a taxonomy. The authors need to choose one path: either formalize RARL with testable predictions, quantitative comparison methodology, or formal unification properties that justify "framework" - or honestly reframe RARL as a "reasoning-centric taxonomic survey" and adjust all claims throughout. Every instance of the word "framework" should either be backed by formal properties or replaced with "taxonomy" or "structured survey." This is the single most important revision.

2. **Add synthesis paragraphs to enumeration-heavy sections (section 4.3, section 6).** Each application domain subsection currently lists papers without extracting cross-cutting principles. The authors need to add a synthesis paragraph at the end of each subsection answering: "What does this body of work collectively reveal about reward design for this domain?" Without this, those sections are annotated bibliographies masquerading as survey analysis. Each application subsection must contain at least one paragraph synthesizing findings across cited works rather than summarizing them individually.

### Major

3. **Sharpen differentiation from concurrent surveys.** With five concurrent surveys in the same space, the paper needs to explicitly demonstrate - not just claim - what RARL provides that others don't. A comparison table showing coverage gaps across at least three concurrent surveys, with specific topics, analyses, or insights unique to RARL, is the minimum. Prose assertions about scope differences won't suffice.

4. **Add cross-method comparative analysis.** The paper claims "paradigm complementarity" but provides no evidence of it. Either add a comparative analysis - even qualitative - showing how different reward paradigms perform on shared benchmarks or reasoning tasks, or remove the complementarity claim entirely. A comparison table or analysis, or deletion of the claim.

### Minor

5. **Report survey methodology.** Add a brief methodology section (at least 200 words) describing the search strategy, time period covered, sources consulted, and any selection criteria applied. This doesn't need to be PRISMA-compliant, but it should let a reader assess whether coverage is comprehensive. Right now it's a black box.

6. **Distinguish peer-reviewed from preprint sources.** Flag which key findings rely on non-peer-reviewed preprints and note where conclusions should be considered preliminary. At minimum, a footnote or appendix noting the peer-review status of heavily-cited works.

---

> ### Author Response · Authors · 2026-05-08
> **Author Response to Reviewer ttDv**
>
> We thank Reviewer ttDv for a constructive reading. We address each requested change below.
>
> CRITICAL · C1 - Framework vs. taxonomy
>
> The claim that RARL constitutes a "unifying framework" was overclaimed. In the revision, we have replaced all framework-language with taxonomy/perspective language throughout. Specifically:
> 1. Abstract: "…RARL, a reasoning-centric taxonomic perspective that organizes diverse reward paradigms…"
> 2. Introduction (contributions bullet): "…systematic exploration of reward mechanisms…in a reasoning-centric taxonomic survey…"
> 3. Related Work paragraph: "RARL is a reasoning-centric and taxonomic perspective…offering a shared analytical lens…"
> 4. Figure 2 caption: "An organizing taxonomy of reward paradigms in RARL..."
>
> We also added a dedicated paragraph (§1.1, "What RARL adds beyond a reading list") that makes the analytical value of the taxonomy concrete: the shared MDP formulation reveals that RLHF, RLVR, and RLAIF differ primarily in reward semantics rather than algorithmic structure — a non-obvious observation that motivates the three-axis taxonomy. We no longer claim RARL generates testable predictions.
>
> CRITICAL · C2 - Synthesis paragraphs in Sections 4.3 and 6
> Both sections now have structured Key Takeaways boxes that synthesize cross-cutting principles rather than summarizing individual papers.
>
> 1. Section 4.3 (Augmented Reasoning) takeaway box identifies two principles that hold across RAG, table reasoning, and TIR: (1) reward decomposition is structurally necessary once external components enter the loop; (2) outcome sparsity in augmented settings makes process supervision a structural necessity rather than an optional refinement.
> 2. Section 6 (Applications) takeaway box identifies three principles across medicine, finance, and science: (1) binary correctness rewards break down when flawed reasoning chains can cause real-world harm; (2) process supervision becomes the primary signal when outcome verification is costly or delayed; (3) domain-specific reward components (volatility adjustment, guideline verification, structural fingerprint matching) are what distinguish meaningful feedback from generic correctness.
>
> MAJOR - Differentiation from concurrent surveys
>
> We added Table 1 (§1.1) comparing RARL against four concurrent surveys across eight coverage dimensions. The table uses a three-level rating (covered / partial / absent) and demonstrates that RARL is the only survey covering all of: reward hacking with 5 bias types and mitigations, hallucination and diversity collapse, augmented reasoning (RAG/TIR/Tables), RL training challenges (entropy collapse), and domain applications with synthesis. No concurrent survey covers more than three of these dimensions.
>
> MAJOR - Paradigm complementarity claim
>
> We have softened this claim. The revised §1.1 no longer asserts that complementarity is demonstrated — it now describes RARL as providing an "organizing lens that enables categorization of reward semantics and facilitates the creation of more diverse reward annotations," with a concrete example (RLAIF's automation at the cost of self-preference bias vs. DPO's stability at the cost of adaptivity). We agree with the reviewer that a full comparative analysis across paradigms on shared benchmarks would be needed to substantiate the stronger claim, and we have removed the stronger version.
>
> MINOR - Survey methodology
>
> Done. A methodology section (§1.2) has been added (~200 words) covering: search sources (arXiv cs.LG/CL/AI, Semantic Scholar, NeurIPS/ICML/ICLR/ACL/EMNLP proceedings), time period (January 2022 – April 2025), inclusion criteria (reward signal design, reward model training/evaluation, RL fine-tuning with explicit reasoning objective), and exclusion criteria (non-language RL unless directly transferable). The corpus comprises over 250 papers.
>
> MINOR - Peer-review status of preprint sources
>
> Done. The methodology section now explicitly flags that a significant share of late 2024–early 2025 citations are non-peer-reviewed preprints, and notes that "conclusions from those works should be considered preliminary." This applies particularly to DeepSeek-R1 and several RLVR papers in Section 4.4.
>
> ACKNOWLEDGED
> On what the reviewer found valuable
>
> We appreciate the reviewer's characterization of Section 4.4 (diversity collapse and the RL debate) as the paper's strongest contribution, and the reward hacking taxonomy as immediately useful to practitioners. The reviewer's framing that Section 4.4 is what the whole paper could achieve if it maintained that level of synthesis has been guiding our revision of the weaker sections. We also note the reviewer's comment that Sections 2 and 3 are "by far the most publication-ready parts" and have preserved those without structural changes.

---

> > ### Comment · Reviewer_ttDv · 2026-05-09
> > **Thank you for the responsive revision. The paper has improved substantially and several additions go beyond what was requested.**
> >
> > ## What Works Well
> >
> > **The framework-to-taxonomy reframing is effective.** The abstract, contributions bullet, and Figure 2 caption now accurately describe what the paper delivers. The "What RARL adds beyond a reading list" paragraph makes a compelling case for the taxonomy's value without inflating it. The concession that "This perspective does not generate formal predictions, but it does generate a diagnostic checklist with demonstrated utility" builds reader trust.
> >
> > **The Takeaway boxes are the revision's standout contribution.** They answer my core question: "What does this body of work collectively reveal?" The §4.3 box (faithfulness enforcement + process supervision across RAG/tables/TIR) and §6 box (binary correctness insufficiency, process supervision necessity, domain-specific components) extract genuine cross-cutting principles. The §3 box — "Reward hacking is structural, not incidental... biases share a common root" — was unrequested and genuinely insightful.
> >
> > **Table 1 differentiates effectively from concurrent surveys.** The eight-dimension comparison credibly demonstrates RARL's unique coverage of augmented reasoning, RL training challenges, and domain applications with synthesis.
> >
> > **Proactive improvements:** Section 3.4 on Faithfulness Bias grew into a rigorous treatment with a novel conclusion — all existing faithfulness measures are behavioral proxies, making the problem fundamentally hard to address through reward design alone. The RL Debate (§4.4.2) now has formal grounding via Davis & Recht (2025) and mechanistic evidence from Akgül et al. (2025), with a consensus specific enough to be actionable. The MDP Granularity formalization adds formal depth absent before.
> >
> > ## What Remains Unresolved
> >
> > **The "Analytical Advantages" were not actually addressed (C4).** The authors state they "softened this claim," but the actual change is from "This unification offers several key advantages" to "This shared perspective offers several analytical advantages" — one word. The three advantages themselves (Paradigm Complementarity, Cross-Domain Methodological Transfer, Systematic Failure Mode Discovery) are prose-identical to the original.
> >
> > This matters because these claims set expectations the paper does not meet. "Cross-Domain Methodological Transfer" asserts that RARL "enables the transfer of insights across traditionally separate research areas," yet the domain sections treat medicine, finance, and science independently with no cross-references. "Systematic Failure Mode Discovery" implies prediction, but the taxonomy organizes already-known failures — it does not discover new ones.
> >
> > The irony is that the paper contains an implicit cross-domain transfer example — MCTS for step-level reward estimation originates in math PRM literature and appears in ReasonRAG — but never identifies this as transfer. The evidence exists; it is just not articulated.
> >
> > **The Conclusion was not updated.** The abstract says "taxonomic perspective" but §8 still says "using rewards as a unifying lens." This inconsistency is noticeable reading start-to-finish.
> >
> > **Body text of §4.3 and §6 is unchanged.** The Takeaway boxes are appended to the same paper-by-paper enumeration I originally flagged. The boxes compensate for acceptance purposes, but this is why I cannot recommend Survey Certification — the body prose would need to match the analytical quality of §4.4.
> >
> > ## What I Need Before Recommending Accept
> >
> > One issue requires resolution. The three "analytical advantages" need to be qualified as structural affordances rather than demonstrated properties:
> >
> > 1. Reframe the introductory sentence to signal intent rather than established fact — e.g., "This shared perspective is designed to afford several analytical advantages" rather than presenting them as demonstrated.
> >
> > 2. For "Cross-Domain Transfer" — either qualify it as a potential benefit the structure enables but does not yet demonstrate, or identify the MCTS-from-math-to-RAG example already in the paper as a concrete instance. The evidence is there; it just needs naming.
> >
> > 3. For "Systematic Failure Mode Discovery" — replace "discovery" with "organization" or "characterization." Organizing known failure modes into an actionable structure is valuable. It is not discovery.
> >
> > 4. Fix the Conclusion's "unifying lens" to match the "organizing/taxonomic" language elsewhere.
> >
> > These are sentence-level edits requiring no new research.

---

> > > ### Author Response · Authors · 2026-05-10
> > > **Follow-up Reply to Reviewer ttDv**
> > >
> > > We thank Reviewer ttDv for the follow-up. All the 4 issues are addressed.
> > >
> > > FIX 1
> > > Analytical advantages
> > >
> > > The introductory sentence now reads: "This shared perspective is designed to afford several analytical advantages.", as requested.
> > >
> > > FIX 2
> > > Cross-Domain Transfer — concrete MCTS example added
> > >
> > > We added the instance: Monte Carlo tree search for step-level reward estimation, originating in the math PRM literature, reappears in ReasonRAG for augmented reasoning, a transfer made visible by the shared process supervision principle in the RARL taxonomy. The reward hacking sentence is retained as a second, broader structural benefit.
> > >
> > > FIX 3
> > > "Discovery" → "Characterization" in the third advantage.
> > >
> > > FIX 4
> > > Conclusion "unifying lens" → "organizing lens" in Conclusion (§8)

---

### Decision · Action_Editor_T3vW · 2026-06-26

**Recommendation:** Accept as is

**Audience:**

Yes

**Audience Explanation:**

The reviewers agree that the paper provides a valuable organization of ideas in the literature surrounding reasoning, which provides worthwhile insights into e.g. different types of reward hacking problems and how they can be addressed.

**Claims And Evidence:**

Yes

**Claims Explanation:**

The reviewers agreed that the original manuscript overclaimed on several topics, most centrally when it positioned RARL as a unifying framework/theory for reasoning in LLMs. There was a productive discussion period and the authors provided a revised manuscript to right-size the claims of the paper to the evidence presented; the reviewers all agree that the new manuscript properly positions itself as providing a post-hoc categorization of existing approaches rather than an explanatory/predictive theory. Claims of insights in other areas such as cross-domain transfer and failure mode discovery have also been appropriately tempered.